# Analyzing Sharpness along GD Trajectory: Progressive Sharpening and Edge of Stability

**Zhouzi Li**[*]
IIIS, Tsinghua University
zhouzi188763@gmail.com

**Zixuan Wang**[*]
IIIS, Tsinghua University
wangzx2019012326@gmail.com

**Jian Li**[†]
IIIS, Tsinghua University
lapordge@gmail.com

## Abstract

Recent findings demonstrate that modern neural networks trained by full-batch gradient descent typically enter a regime called Edge of Stability (EOS). In this regime, the sharpness, i.e., the maximum Hessian eigenvalue, first increases to the value 2/(step size) (the progressive sharpening phase) and then oscillates around this value (the EOS phase). This paper aims to analyze the GD dynamics and the sharpness along the optimization trajectory. Our analysis naturally divides the GD trajectory into four phases depending on the change in the sharpness value. We empirically identify the norm of output layer weight as an interesting indicator of the sharpness dynamics. Based on this empirical observation, we attempt to theoretically and empirically explain the dynamics of various key quantities that lead to the change of the sharpness in each phase of EOS. Moreover, based on certain assumptions, we provide a theoretical proof of the sharpness behavior in the EOS regime in two-layer fully-connected linear neural networks. We also discuss some other empirical findings and the limitation of our theoretical results.

## 1 Introduction

Deep learning has achieved great success in a variety of machine learning applications, and gradient-based algorithms are the prevailing optimization methods for training deep neural networks. However, mathematically understanding the behavior of the optimization methods for deep learning is highly challenging, due to non-convexity, over-parameterization, and complicated architectures. In particular, some recent empirical findings in deep networks contradict the traditional understandings of gradient methods. For example, Wu et al. [30] observed that the solution found by gradient descent has sharpness approximately equal to $2/\eta$ instead of just being smaller than $2/\eta$. Also, Jastrzebski et al. [14] observed that there is a break-even point in the SGD trajectory, and after this point, there is a regularization effect on the loss curvature.

One recent well-known example is the phenomenon called "Edge of Stability" (EOS) (Cohen et al. [6]). Based on the classical optimization theory, the learning rate $\eta$ of gradient-based method should be smaller than $2/\lambda$ so that the loss can decrease, where $\lambda$ is the largest eigenvalue of the Hessian of the objective, also called "sharpness" in the literature. Otherwise, the loss diverges (even for simple quadratic functions). However, the empirical findings in Cohen et al. [6] show that under various

---

[*]Contributed equally, listed in alphabetical order.

[†]The authors are supported in part by the National Natural Science Foundation of China Grant 62161146004, Turing AI Institute of Nanjing and Xi'an Institute for Interdisciplinary Information Core Technology.

36th Conference on Neural Information Processing Systems (NeurIPS 2022).

network settings, the EOS phenomena typically occurs along the gradient descent trajectory: (1) the sharpness first increases until it reaches $2/\eta$ (called "progressive sharpening") (2) the sharpness starts hovering around $2/\eta$ (the EOS regime) and (3) the loss non-monotonically decreases without diverging.

Although (1) seems to be consistent with the traditional beliefs about optimization, a rigorous mathematical explanation for it is still open. Moreover, phenomena (2) and (3) are more mysterious because they violate the $\eta < 2/\lambda$ "rule" in traditional optimization theory, yet the training loss does not completely diverge. Instead, the loss may oscillate but still decrease in the long run, while the sharpness seems to be restrained from further increasing.

In this paper, we aim to provide a theoretical and empirical explanation for the mystery of EOS. Towards the goal, we focus on the dynamics of these key quantities when EOS happens and attempt to find out the main driving force to explain these phenomena along the gradient descent trajectory from both theoretical and empirical perspectives.

## 1.1 Our Contributions

Our contributions can be summarized as follows.

(Section 3.1) We analyze the typical sharpness behavior along the gradient descent trajectory when EOS happens, and propose a four-phase division of GD trajectory, based on the dynamics of some key quantities such as the loss and the sharpness, for further understanding this phenomenon.

(Section 3.2) We empirically identify the weight norm of the output layer as an effective indicator of the sharpness dynamics. We show that analyzing the dynamics of this surrogate can qualitatively explain the dynamics of sharpness. By assuming this relation, together with some additional simplifying assumptions and approximations, we can explain the dynamics of the sharpness, the loss, and the output layer norm in each phase of EOS (Section 3.3). In this context, we also offers an interesting explanation for the non-monotonic loss decrement (also observed in Cohen et al. [6], Xing et al. [32]) (Section 3.4).

(Section 4) Following similar ideas, we provide a more rigorous proof for the progressive sharpening and EOS phenomena in a two-layer fully-connected linear neural network setting based on certain assumptions. The assumptions made here are either weaker or arguably less restrictive.

## 1.2 Related work

**The structure of Hessian** The Hessian matrix carries the second order information of the loss landscape. Several prior works have empirically found that the spectrum of Hessian has several "outliers" and a continuous "bulk" (Sagun et al. [28, 29], Papyan [25, 26]). Typically, each outlier corresponds to one class in multi-class classification. As we consider the binary classification setting, there is typically one outlier (i.e., the largest eigenvalue) that is much larger than other eigenvalues. It is consistent with our Assumption 4.1. The Gauss-Newton decomposition of the Hessian was used in several prior works (Martens [23], Bottou et al. [4], Papyan [25, 26]). Papyan [25] empirically showed that the outliers of Hessian can be attributed to a "G component", which is also known as Fisher Information Matrix (FIM) in Karakida et al. [15, 16]. Also, Wu et al. [31] analyzed the leading Hessian eigenspace by approximating the Hessian with Kronecker factorization and theoretically proved the outliers structure under some random setting assumption.

**Neural Tangent Kernel** A recent line of work studied the learning of over-parameterized neural networks in the so-called. "neural tangent kernel (NTK) regime or the lazy training regime (Jacot et al. [13], Lee et al. [18], Du et al. [8, 7], Arora et al. [2], Chizat et al. [5]). A main result in this regime is that if the neural network is wide enough, gradient flow can find the global optimal empirical minimizer very close to the initialization. Moreover, the Hessian does not change much in the NTK regime. Our findings go beyond NTK setting to analyze the change of sharpness.

**Edge of Stability regime** The Edge of Stability phenomena was first formalized by Cohen et al. [6]. Similar phenomena were also identified in Jastrzebski et al. [14] as the existence of the "break-even" point on SGD trajectory after which loss curvature gets regularized. Xing et al. [32] observed that gradient descent eventually enters a regime where the iterates oscillate on the leading curvature direction and the loss drops non-monotonically. Recently Ahn et al. [1] studied the non-monotonic

decreasing behavior of GD which they called unstable convergence, and discussed the possible causes of this phenomenon. Ma et al. [22] proposed a special subquadratic landscape property and proved that EOS occurs based on this assumption. Arora et al. [3] studied the implicit bias on the sharpness of deterministic gradient descent in the EOS regime. They proved in some specific settings with a varying learning rate (called normalized GD) or with a modified loss $\sqrt{L}$, gradient descent enters EOS and further reduces sharpness. They mainly focus on the analysis near the manifold of minimum loss, but our analysis also applies to the early stage of the training when the loss is not close to the minimum. In particular, our analysis provides an explanation of non-monotonic loss decrease that cannot be explained by their theory. Another difference is that they consider $\sqrt{L}$ (for constant learning rate) where $L$ is a fairly general MSE loss independent of any neural network structure, while our analysis is strongly tied with the MSE loss of a neural network. Very recently, Lyu et al. [21] explained how GD enters EOS for normalized loss (e.g., neural networks with normalization layers), and analyzed the sharpness reduction effect along the training trajectory. The notion of sharpness in their work is somewhat different due to normalization. In particular, they consider the so-called *spherical sharpness*, that is the sharpness of the normalized weight vector. They also mainly studied the regime where the parameter is close to the manifold of minimum loss as in [3] and proved that GD approximately tracks a continuous sharpness-reduction flow. Lewkowycz et al. [19] proposed a similar regime called "catapult phase" where loss does not diverge even if the largest Hessian eigenvalue is larger than $2/\eta$. Our work mainly considers training in this regime and assumes that the training is not in the "divergent phase" in Lewkowycz et al. [19]. Compared with Lewkowycz et al. [19], we provide a more detailed analysis in more general settings along gradient descent trajectory.

## 2 Preliminaries

**Notations:** We denote the training dataset as $\{\mathbf{x}_i, y_i\}_{i=1}^n \subset \mathbb{R}^d \times \{1, -1\}$ and the neural network as $f : \mathbb{R}^d \times \mathbb{R}^p \to \mathbb{R}$. The network $f(\boldsymbol{\theta}, \mathbf{x})$ maps the input $\mathbf{x} \in \mathbb{R}^d$ and parameter $\boldsymbol{\theta} \in \mathbb{R}^p$ to an output in $\mathbb{R}$. In this paper, we mainly consider the case of binary classification with mean square error (MSE) loss $\ell(z, y) = (z - y)^2$.

Denote the input matrix as $\mathbf{X} = (\mathbf{x}_1, \mathbf{x}_2, ..., \mathbf{x}_n) \in \mathbb{R}^{d \times n}$ and the label vector as $\boldsymbol{Y} = (y_1, y_2, ..., y_n) \in \mathbb{R}^n$. We let $\boldsymbol{F}(t) = (f(\boldsymbol{\theta}(t), \mathbf{x}_1), f(\boldsymbol{\theta}(t), \mathbf{x}_2), ..., f(\boldsymbol{\theta}(t), \mathbf{x}_n)) \in \mathbb{R}^n$ and $\boldsymbol{D}(t) = \boldsymbol{F}(t) - \boldsymbol{Y}$ be the (output) prediction vector, and the residual vector respectively at time $t$. The training objective is: $\mathcal{L}(f(\boldsymbol{\theta})) = \frac{1}{n} \sum_{i=1}^n \ell(f(\boldsymbol{\theta}, \mathbf{x}_i), y_i) = \frac{1}{n} \sum_{i=1}^n (f(\boldsymbol{\theta}, \mathbf{x}_i), y_i)^2$.

**Hessian, Fisher information matrix and NTK:** In this part, we apply previous works to show that the largest eigenvalue of Hessian is almost the same as the largest eigenvalue of NTK. We use the latter as the definition of **the sharpness** in this paper. Further details can be found in Appendix F.

As shown in Papyan [26], Martens [23], Bottou et al. [4], the Hessian can be decomposed into two components, where the term known as "Gauss-Newton matrix", G-term or Fisher information matrix (FIM), dominates the second term in terms of the largest eigenvalue. Meanwhile, Karakida et al. [16] pointed out the duality between the FIM and a Gram matrix $\boldsymbol{M}$, defined as $\boldsymbol{M} = \frac{2}{n} \frac{\partial \boldsymbol{F}(\boldsymbol{\theta})}{\partial \boldsymbol{\theta}} \frac{\partial \boldsymbol{F}(\boldsymbol{\theta})}{\partial \boldsymbol{\theta}}^\top$. It is also known as the neural tangent kernel NTK (Karakida et al. [16, 15]), which has been studied extensively in recent years (see e.g., [13],[8],[2],[5]). Note that in this paper, we do not assume the training is in NTK regime, in which the Hessian does not change much during training. It is not hard to see that $\boldsymbol{M}$ and FIM share the same non-zero eigenvalues: if $\boldsymbol{G}\boldsymbol{u} = \lambda\boldsymbol{u}$ for some eigenvector $\boldsymbol{u} \in \mathbb{R}^p$, $\boldsymbol{M}\frac{\partial \boldsymbol{F}(\boldsymbol{\theta})}{\partial \boldsymbol{\theta}}\boldsymbol{u} = \frac{\partial \boldsymbol{F}(\boldsymbol{\theta})}{\partial \boldsymbol{\theta}}\boldsymbol{G}\boldsymbol{u} = \lambda\frac{\partial \boldsymbol{F}(\boldsymbol{\theta})}{\partial \boldsymbol{\theta}}\boldsymbol{u}$, i.e., $\lambda$ is also an eigenvalue of $\boldsymbol{M}$.

In this paper, we use $\boldsymbol{\theta}(t)$ to denote the parameter at iteration $t$ (or time $t$) and the sharpness at time $t$ as $\Lambda(t) = \Lambda(\boldsymbol{\theta}(t))$. We similarly define $\boldsymbol{M}(t), \boldsymbol{F}(t), \boldsymbol{D}(t), \mathcal{L}(t)$.

Here we show the gradient flow dynamics of the residual vector $\boldsymbol{D}(t)$:

$$\frac{\mathrm{d}\boldsymbol{D}(t)}{\mathrm{d}t} = \frac{\partial \boldsymbol{D}(t)}{\partial \boldsymbol{\theta}} \frac{\mathrm{d}\boldsymbol{\theta}(t)}{\mathrm{d}t} = -\frac{\partial \boldsymbol{F}(t)}{\partial \boldsymbol{\theta}} \frac{\partial \mathcal{L}(t)}{\partial \boldsymbol{\theta}} = -\frac{2}{n} \frac{\partial \boldsymbol{F}(t)}{\partial \boldsymbol{\theta}} \frac{\partial \boldsymbol{F}(t)}{\partial \boldsymbol{\theta}}^\top \boldsymbol{D}(t) = -\boldsymbol{M}(t)\boldsymbol{D}(t) \quad (1)$$

## 3 A Four-phase Analysis of GD Dynamics

In this section, we study the dynamics of gradient descent and the change of sharpness along the optimization trajectory. We divide the whole training process into four phases, occurring repeatedly

in the EOS regime. In Section 3.1, we introduce the four phases. In Section 3.2, we show empirically that the change of the norm of the output layer weight vector almost coincides with the change of the sharpness. In Section 3.3, using this observation, we attempt to explain the dynamics of each phase and provide a mathematical explanation for the changes in the sharpness. In Section 3.4, we explain why the loss decreases but non-monotonically. We admit that a completely rigorous theoretical explanation is still beyond our reach and much of our argument is based on various simplifying assumptions and is somewhat heuristic at some points. Due to space limits, we defer all the proofs in this section to Appendix E.1.

### 3.1 A Four-phase Division

To further understand the properties along the trajectory when EOS happens, we study the behaviors of the loss and the sharpness during the training process. As illustrated in Figure 1, we train a shallow neural network by gradient descent on a subset of 1,000 samples from CIFAR-10 (Krizhevsky et al. [17]), using the MSE loss as the objective. Notice that the sharpness keeps increasing while the loss decreases until the sharpness reaches $2/\eta$. Then the sharpness begins to oscillate around $2/\eta$ while the loss decreases non-monotonically. This is a typical sharpness behavior in the EOS regime, and consistent with the experiments in [6].

We divide the training process into four phases according to the evolution of the loss, the sharpness, and their correlation, as shown in Figure 1. The four phases happen cyclically along the training trajectory. We first briefly describe the properties of each phase and explain the dynamics in more detail in Section 3.3.

**Phase I:** Sharpness $\Lambda < 2/\eta$. In this stage, all the eigenvalues of Gram matrix $\boldsymbol{M}$ are below the threshold $2/\eta$. In particular, using standard initialization, the training typically starts from this phase, and during this phase the loss keeps decreasing and the sharpness keeps growing along the trajectory. This initial phase is called *progressive sharpening (PS)* in prior work Cohen et al. [6]. Empirically, the behavior of GD trajectory (as well as the loss and the sharpness) is very similar to that of gradient flow, until the sharpness reaches $2/\eta$ (this phenomena is also observed in Cohen et al. [6]. See Figure 5 or Appendix J.1 in their paper). We note that GD may come back to this phase from Phase IV later.

**Phase II:** Sharpness $\Lambda > 2/\eta$. In this phase, the sharpness exceeds $2/\eta$ and may keep increasing. We will show shortly that the fact that $\Lambda > 2/\eta$ causes $|\boldsymbol{D}^\top \boldsymbol{v}_1|$ (where $\boldsymbol{v}_1$ the first eigenvector of $\boldsymbol{M}$) to increase exponentially (Lemma 3.2). This would quickly lead $\|\boldsymbol{D}\|$ to exceed $\|\boldsymbol{Y}\|$ in a few iterations, which leads the sharpness to start decreasing by Proposition 3.1, hence the training process enters Phase III.

**Phase III:** Sharpness $\Lambda > 2/\eta$ yet begins to gradually drop. Before $\Lambda$ drops below $2/\eta$, Lemma 3.2 still holds, so $|\boldsymbol{D}^\top \boldsymbol{v}_1|$ keeps increasing. Proposition 3.1 still holds and thus the sharpness keeps decreasing until it is below $2/\eta$, at which point we enter Phase IV. A distinctive feature of this phase is that the loss may increase due to the exponential increase of $|\boldsymbol{D}^\top \boldsymbol{v}_1|$.

**Phase IV:** Sharpness $\Lambda < 2/\eta$. When the sharpness is below $2/\eta$, $|\boldsymbol{D}^\top \boldsymbol{v}_1|$ begins to decrease quickly, leading the loss to decrease quickly. At the same time, the sharpness keeps oscillating and gradually decreasing for some iterations. This lasts until the loss decrease to a level that is around its value right before Phase III. The sharpness is still below $2/\eta$ and our training process gets back to Phase I.

### 3.2 The Norm of the Output Layer Weight

It is difficult to rigorously analyze the dynamics of the sharpness $\Lambda(t)$. In this subsection, we make an interesting observation, that the change of the norm of the output layer of the network (usually a fully-connected linear layer) is consistent with the change of the sharpness most of the time.

In particular, for a general neural network $f(\mathbf{x}) = \boldsymbol{A}^\top \boldsymbol{h}(\boldsymbol{W}, \mathbf{x})$, where $\boldsymbol{A} \in \mathbb{R}^m$ is the output layer weight and the feature extractor $\boldsymbol{h} : \mathbb{R}^p \times \mathbb{R}^d \to \mathbb{R}^m$ outputs a $m$-dimensional feature vector ($\boldsymbol{h}$ corresponds to all but the last layers). $\boldsymbol{W} \in \mathbb{R}^p$ is the parameter vector of the extractor $\boldsymbol{h}$.

Note that $\boldsymbol{M} = (\frac{\partial \boldsymbol{F}}{\partial \boldsymbol{\theta}})^\top (\frac{\partial \boldsymbol{F}}{\partial \boldsymbol{\theta}})$ can be decomposed as follows:

$$\boldsymbol{M} = \left(\frac{\partial \boldsymbol{F}}{\partial \boldsymbol{\theta}}\right)\left(\frac{\partial \boldsymbol{F}}{\partial \boldsymbol{\theta}}\right)^\top = \left(\frac{\partial \boldsymbol{F}}{\partial \boldsymbol{A}}\right)\left(\frac{\partial \boldsymbol{F}}{\partial \boldsymbol{A}}\right)^\top + \left(\frac{\partial \boldsymbol{F}}{\partial \boldsymbol{W}}\right)\left(\frac{\partial \boldsymbol{F}}{\partial \boldsymbol{W}}\right)^\top := \boldsymbol{M_A} + \boldsymbol{M_W}.$$

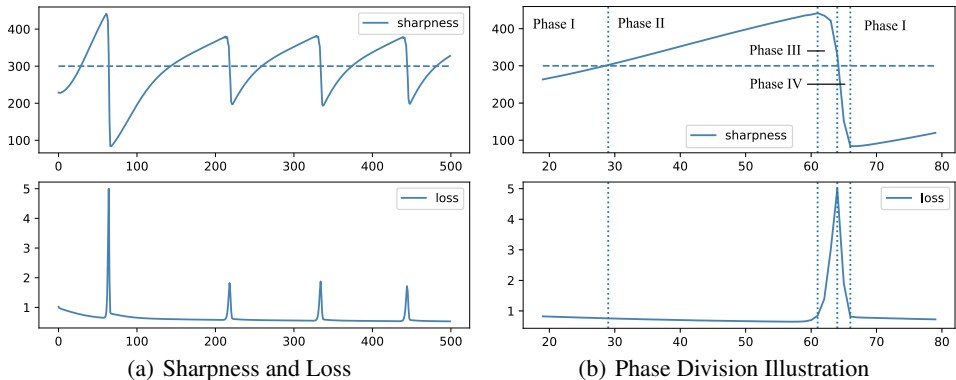

(a) Sharpness and Loss  (b) Phase Division Illustration

Figure 1: In Figure (a), we show the sharpness and the loss when training on a two-hidden-layer linear activated network by gradient descent. Shortly after the sharpness crosses $2/\eta$, it drops quickly back to some value below $2/\eta$. In Figure (b) we illustrate how the four phases are divided according to the evolution of the sharpness.

where the $(i,j)$−entry of $M_{\boldsymbol{W}}$ is $(M_{\boldsymbol{W}})_{ij} = \left\langle \frac{\partial f(\mathbf{x}_i)}{\partial \boldsymbol{W}}, \frac{\partial f(\mathbf{x}_j)}{\partial \boldsymbol{W}} \right\rangle = \boldsymbol{A}^\top \frac{\partial \boldsymbol{h}(\boldsymbol{W}, \mathbf{x}_i)}{\partial \boldsymbol{W}} \frac{\partial \boldsymbol{h}(\boldsymbol{W}, \mathbf{x}_j)}{\partial \boldsymbol{W}}^\top \boldsymbol{A}$.

In this expression, intuitively $\|\boldsymbol{A}\|$ should be positively related to $\|M_{\boldsymbol{W}}\|$. We empirically observe that the part $M_{\boldsymbol{A}} = (\frac{\partial \boldsymbol{F}}{\partial \boldsymbol{A}})(\frac{\partial \boldsymbol{F}}{\partial \boldsymbol{A}})^\top$ has a much smaller spectral norm compared to the whole Gram matrix $M$ (see Figure 3(a) and Appendix D), which means $\|M_{\boldsymbol{W}}\|$ dominates $\|M_{\boldsymbol{A}}\|$. Therefore, $\|\boldsymbol{A}\|$ should be positively correlated with $\|M\|$.

The benefit of analyzing $\|\boldsymbol{A}\|^2$ is that the gradient flow of $\|\boldsymbol{A}\|^2$ enjoys the following clean formula:

$$\frac{\mathrm{d}\|\boldsymbol{A}\|^2}{\mathrm{d}t} = -2\left(\frac{\partial \mathcal{L}}{\partial \boldsymbol{A}}\right)^\top \boldsymbol{A} = -\frac{4}{n}\boldsymbol{D}^\top \left(\frac{\partial \boldsymbol{F}}{\partial \boldsymbol{A}}\right)\boldsymbol{A} = -\frac{4}{n}\boldsymbol{D}^\top \boldsymbol{F}. \tag{2}$$

In this work, we do experiments on two-layer linear networks, fully connected deep neural networks, and Resnet18, and all of them have such output layer structures. From Figure 3(a), we can observe that **the output layer norm $\|\boldsymbol{A}\|^2$ and the sharpness $\Lambda$ change in the same direction** most of the time along the gradient descent trajectory, i.e., they both increase or decrease at the same time. We note that they may change in different directions very occasionally around the time when $\|\boldsymbol{A}(t+1)\|^2 - \|\boldsymbol{A}(t)\|^2$ changes its sign (see the experiments in Figure 2).

### 3.3 Detailed Analysis of Each Phase

In this section, we explain the dynamics of each phase in more detail. For clarity, we first list the assumptions we need in this section. For different phases, we may need some different assumptions to simplify the arguments. Most of the assumptions are consistent with the experiments or the findings in the literature. Some of them are somewhat stronger, and we also discuss how to relax them.

#### 3.3.1 Assumptions Used in Section 3.3

**Assumption 3.1.** *(A-norm and sharpness) Along the gradient descent training trajectory, for all time $t$, the norm $\|\boldsymbol{A}(t)\|$ of the output layer and the sharpness $\Lambda(t)$ moves in the same directions, i.e., $\mathrm{sign}(\Lambda(t+1) - \Lambda(t)) = \mathrm{sign}(\|\boldsymbol{A}(t+1)\| - \|\boldsymbol{A}(t)\|)$.*

It is the key observation that we have discussed in Section 3.2. The following are two assumptions about the gradient descent trajectory. The first one assumes that $\boldsymbol{D}(t)$ and $\|\boldsymbol{A}\|^2$ are updated according their first order approximations. Empirical justification of this approximation can be found in Appendix D.1.3.

**Assumption 3.2.** *(First Order Approximation of GD) Along the gradient descent trajectory, the update rule is assumed as the first order approximation*

$$\boldsymbol{D}(t+1) - \boldsymbol{D}(t) = -\eta M(t)\boldsymbol{D}(t), \quad \|\boldsymbol{A}(t+1)\|^2 - \|\boldsymbol{A}(t)\|^2 = -\frac{4\eta}{n}\boldsymbol{D}(t)^\top \boldsymbol{F}(t) \tag{3}$$

**Assumption 3.3.** *(Gradient flow for the PS phase) When $\Lambda(t) < 2/\eta$, $\boldsymbol{D}(t)$ follows the gradient flow trajectory:* $\frac{\mathrm{d}\boldsymbol{D}(t)}{\mathrm{d}t} = -M(t)\boldsymbol{D}(t)$.

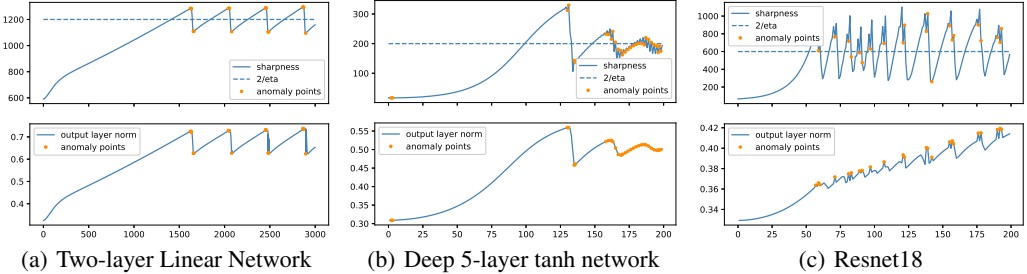

(a) Two-layer Linear Network    (b) Deep 5-layer tanh network    (c) Resnet18

Figure 2: Sharpness and A norm are highly correlated; *Anomaly points* (orange points) in the figure correspond to the iterations when $\text{sign}(\Lambda(t+1) - \Lambda(t)) \neq \text{sign}(\|\boldsymbol{A}(t+1)\| - \|\boldsymbol{A}(t)\|)$. More discussions can be found in Section 5 and Appendix B.

Assumption 3.3 holds empirically, especially in the progressive sharpening phase (see Figure 5 or Appendix J.1 in Cohen et al. [6]) when the networks are continuously differentiable. We include these experimental details in Appendix D. See also (Theorems 4.3 and 4.5) in Arora et al. [3] for further theoretical justification. We need this assumption for the proof in the progressive sharpening phase.

Then we state an assumption on the upper bound of the sharpness to restrict the regime we discuss:

**Assumption 3.4.** *(Sharpness upper bound) If the training does not diverge, there exists some constant $B_\Lambda$, such that $0 < \Lambda(t) \leq \frac{B_\Lambda}{\eta}$ for all $t$.*

This assumption states that there is an upper bound of the sharpness throughout the optimization process. Actually, in Lewkowycz et al. [19], they proved that $4/\eta$ is an upper bound of the sharpness in a two-layer linear network with one datapoint, otherwise the training process (loss) would diverge. They empirically found that similar upper bounds exist also for nonlinear activations, albeit with somewhat larger constant $B_\Lambda$. In the work, We focus on the case when the loss does not diverge and hence we make Assumption 3.4.

The main set of assumptions we need is about the change of $\boldsymbol{M}$'s eigendirections.

**Assumption 3.5.** *Denote $\{\boldsymbol{v}_i\}_{i=1}^n$ to be the set of eigenvectors of $\boldsymbol{M}(t)$. We have three levels of assumptions on $\boldsymbol{M}$'s eigenspace.*

    *(i) (fixed eigendirections) the set $\{\boldsymbol{v}_i\}_{i=1}^n$ is fixed throughout the phase under consideration;*

    *(ii) (eigendirections move slowly) at all time $t$ and for any $i$, $\boldsymbol{F}(t)^\top \frac{\mathrm{d}\boldsymbol{v}_i(t)}{\mathrm{d}t} < \lambda_i(t)\boldsymbol{D}(t)^\top \boldsymbol{v}_i(t)$;*

    *(iii) (principal directions moves slowly) at all time $t$, there is a small constant $\epsilon_2 \geq 0$ such that $\langle \boldsymbol{v}_1(t), \boldsymbol{v}_1(t+1) \rangle \geq 1 - \epsilon_2$.*

Clearly, these three assumptions are increasingly weaker from (i) to (iii). Assumption 3.5 (i) on the eigenvectors is somewhat strong, and the eigenvectors corresponding to small eigenvalues may change notably in our experiments. We use it to illustrate a basic proof idea of the progressive sharpening phase, but later we relax this assumption to Assumption 3.5 (ii).

Moreover, for the proof in Phase II and III, Assumption 3.5 (iii), which only assume that the main direction changes slightly, is sufficient for our proof. Actually, we note that $\boldsymbol{v}_1(t)$ (the eigenvector corresponding to the largest eigenvalue) changes slowly and the inner product of its initial direction and its direction at the end of the phase is also large (see Appendix D for the empirical verification).

For the proof in Phase II, we need another small technical assumption:

**Assumption 3.6.** *Assume $\boldsymbol{D}(t)^\top \boldsymbol{v}_1(t) \geq c\epsilon_2 \|\boldsymbol{D}(t)\|$ for some $c > 1$ for some $t = t_0$ at the beginning of this phase. Here $\epsilon_2$ is defined in Assumption 3.5 (iii).*

Assumption 3.6 says that $\boldsymbol{D}(t)$ has a non-negligible component in the direction of $\boldsymbol{v}_1$. Since $\epsilon_2 > 0$ is a small constant, this is not a strong assumption as some small perturbation (due to discrete updates) would make the assumption hold for some $c > 1$.

### 3.3.2 Detailed Analysis

In each phase, we attempt to explain the main driving force of the change of the sharpness and the loss.

**Phase I:** In this phase, we show that $\boldsymbol{D}(t)^\top \boldsymbol{F}(t) < 0$ under certain assumptions (detailed shortly) on the spectral properties of $\boldsymbol{M}(t)$ (see Lemma 3.1 below). By Assumption 3.2, we have $\|\boldsymbol{A}(t+1)\|^2 - \|\boldsymbol{A}(t)\|^2 > 0$, implying that the sharpness $\Lambda(t)$ also increases based on Assumption 3.1. This phase stops if $\Lambda(t)$ grows larger than $2/\eta$.

We assume the output vector $\boldsymbol{F}(t)$ is initialized to be small (this is true if we use very small initial weights). For simplicity, we assume $\boldsymbol{F}(0) = 0$ in the following argument.

**Lemma 3.1.** *For all $t$ in Phase I, under Assumption 3.5 (i) and 3.3, it holds that $\boldsymbol{D}(t)^\top \boldsymbol{F}(t) < 0$.*

From this lemma, $\|\boldsymbol{A}\|$ keeps increasing by Assumption 3.2; hence the sharpness keeps increasing by Assumption 3.1 until it reaches $2/\eta$ or the loss converges to 0. In the former case, the training process enters Phase II, while the latter case is also possible when $\eta$ is very small (e.g., even the largest possible sharpness value is less than $2/\eta$). We admit that Assumption 3.5 (i) is somewhat strong. In fact, the assumption can be relaxed significantly to Assumption 3.5 (ii). We show in Appendix E.2 that under Assumption 3.5 (ii) and Assumption 3.3, we can still guarantee $\boldsymbol{D}(t)^\top \boldsymbol{F}(t) < 0$. Moreover, we provide a dynamical system view of the dynamics of $\boldsymbol{D}(t)^\top \boldsymbol{F}(t)$ in that Appendix.

**Phase II:** When the training process just enters Phase II, the sharpness keeps increasing. We show shortly that $\boldsymbol{D}(t)^\top \boldsymbol{v}_1(t)$ starts to increase geometrically, and this causes the sharpness to stop increasing at some point, thus entering Phase III.

In this phase, we adopt a weaker assumption on the sharpness direction $\boldsymbol{v}_1$: Assumption 3.5 (iii). This assumption holds in our experiments (See Figure 17). Also, Assumption 3.6 is necessary.

**Lemma 3.2.** *Suppose Assumption 3.5 (iii) and 3.6 hold during this phase (with constants $\epsilon_2 > 0$ and $c > 1$). If $\Lambda(t) = (2 + \tau)/\eta$ and $\tau > \frac{1}{1 - \epsilon_2 - 1/c} - 1$, then $\boldsymbol{D}(t)^\top \boldsymbol{v}_1(t)$ increases geometrically with factor $(1 + \tau)(1 - \epsilon_2 - 1/c) > 1$ for $t \geq t_0$ in this phase.*

Since $\boldsymbol{D}(t)^\top \boldsymbol{v}_1(t)$ increases geometrically, $\|\boldsymbol{D}\| \geq \boldsymbol{D}(t)^\top \boldsymbol{v}_1(t)$ will exceed $\|\boldsymbol{Y}\|$ eventually. Next, the following proposition states that when this happens, $\boldsymbol{D}(t)^\top \boldsymbol{F}(t) > 0$. Consequently, $\|\boldsymbol{A}\|$ decreases by Assumption 3.2, leading to the decrement of the sharpness based on our Assumption 3.1.

**Proposition 3.1.** *If $\|\boldsymbol{D}(t)\| > \|\boldsymbol{Y}\|$, then $\boldsymbol{D}(t)^\top \boldsymbol{F}(t) > 0$.*

**Phase III:** The sharpness is still larger than $2/\eta$, but it starts decreasing. Meanwhile, the loss continues to increase rapidly due to Lemma 3.2. Eventually, the sharpness will fall below $2/\eta$ and then the training process enters phase IV.

By Lemma 3.2, if the sharpness stays above $2/\eta$, then we can have an arbitrarily large loss. According to Proposition 3.1, if the loss is large enough, the sharpness keeps decreasing.

Now we show that if the sharpness stays above $2/\eta$, $\|\boldsymbol{A}(t)\|^2$ will decrease by a significant amount. This partially explains that the sharpness should also decrease significantly until it drops below $2/\eta$ (instead of decreasingly converging to a value above $2/\eta$ without ever entering the next phase).

**Proposition 3.2.** *Under Assumption 3.2, if $\|\boldsymbol{D}(t)\| > \|\boldsymbol{Y}\|$, then $\|\boldsymbol{A}(t+1)\|^2 - \|\boldsymbol{A}(t)\|^2 < -\frac{4\eta}{n}(\|\boldsymbol{D}(t)\| - \|\boldsymbol{Y}\|)^2$.*

From the above argument, we can see that if $\boldsymbol{D}(t)^\top \boldsymbol{v}_1(t)$ is larger than $\|\boldsymbol{Y}\|$, then $\boldsymbol{D}(t)^\top \boldsymbol{v}_1(t)$ does not decrease in Phase III, and according to Proposition 3.2, $\|\boldsymbol{A}(t)\|^2$ decreases significantly, implying the sharpness drops below $2/\eta$ eventually.

**Remark:** The fact that the sharpness can provably drop below $2/\eta$ in this phase can be proved more rigorously in Section 4 for the two-layer linear setting. See Theorem 2.

**Phase IV:** First, since the training process has just left phase III, $\boldsymbol{D}(t)^\top \boldsymbol{F}(t)$ is still positive and large, hence $\|\boldsymbol{A}(t)\|^2$ keeps decreasing and the sharpness decreases as well. Since the sharpness stays below $2/\eta$, the loss decreases due to the following descent lemma (with $\boldsymbol{u}$ replaced by $\boldsymbol{D}(t)$).

**Lemma 3.3.** *If $\Lambda(t) < 2/\eta$, then for any vector $\boldsymbol{u} \in \mathbb{R}^n$, $\|\boldsymbol{u}^\top(\boldsymbol{I} - \eta\boldsymbol{M}(t))\| \leq (1 - \eta\alpha)\|\boldsymbol{u}\|$, where $\alpha = \min\{2/\eta - \Lambda(t), \lambda_{\min}(\boldsymbol{M}(t))\}$. In particular, replacing $\boldsymbol{u}$ with $\boldsymbol{D}(t)$, we can see $\|\boldsymbol{D}(t+1)\| \leq (1 - \eta\alpha)^2\|\boldsymbol{D}(t)\|$.*

Next we argue that $\boldsymbol{D}(t)^\top \boldsymbol{F}(t)$ will become negative eventually, which indicates that $\|\boldsymbol{A}(t)\|^2$ and hence the sharpness will grow again. Since the sharpness is below $2/\eta$, $\boldsymbol{D}(t)^\top \boldsymbol{v}_1(t)$ decreases geometrically due to Lemma 3.3 (replacing $\boldsymbol{u}$ with $\boldsymbol{D}(t)\boldsymbol{v}_1(t)\boldsymbol{v}_1(t)^\top$).

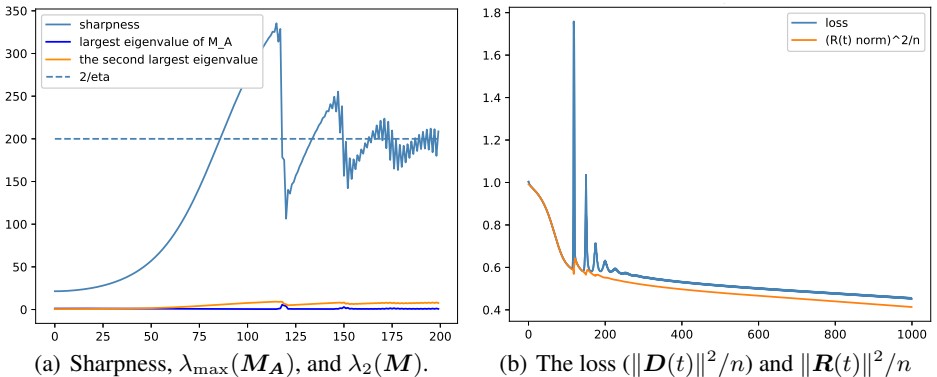

(a) Sharpness, $\lambda_{\max}(\boldsymbol{M_A})$, and $\lambda_2(\boldsymbol{M})$.     (b) The loss ($\|\boldsymbol{D}(t)\|^2/n$) and $\|\boldsymbol{R}(t)\|^2/n$

Figure 3: In Figure (a), we verify the assumptions in Section 3.2 and 3.3 empirically. The steel blue line represents the largest eigenvalue of $\boldsymbol{M}$, i.e. the sharpness $\lambda_{\max}(\boldsymbol{M})$. The orange line: the second largest eigenvalue is far below $1/\eta$, illustrating Assumption 3.7. The blue line: the largest eigenvalue of $\boldsymbol{M_A}$ is negligible compared to the sharpness. In Figure (b), we show that even though the total loss $\mathcal{L}(t) = \|\boldsymbol{D}(t)\|^2/n$ oscillates considerably, $\|\boldsymbol{R}(t)\|^2/n$ decreases more steadily.

In fact, $\boldsymbol{D}$ can be decomposed into the $\boldsymbol{v}_1$-component $\boldsymbol{v}_1\boldsymbol{v}_1^\top\boldsymbol{D}$ and the remaining part $\boldsymbol{R}$ defined as $\boldsymbol{R}(t) := (\boldsymbol{I} - \boldsymbol{v}_1(t)\boldsymbol{v}_1(t)^\top)\boldsymbol{D}(t)$. Then we have

$$\boldsymbol{D}(t)^\top\boldsymbol{F}(t) = (\boldsymbol{v}_1(t)\boldsymbol{v}_1(t)^\top\boldsymbol{D}(t))^\top(\boldsymbol{v}_1(t)\boldsymbol{v}_1(t)^\top\boldsymbol{D}(t) + \boldsymbol{Y}) + \boldsymbol{R}(t)^\top(\boldsymbol{R}(t) + \boldsymbol{Y}). \qquad (4)$$

As shown in the next subsection, $\boldsymbol{R}(t)$ almost follows a similar gradient descent trajectory $\boldsymbol{R}'(t)$ (Lemma 3.5). More precisely, $\boldsymbol{R}'(t)$ is defined as $\boldsymbol{R}'(t+1) = (\boldsymbol{I} - \eta\boldsymbol{M}(t)(\boldsymbol{I} - \boldsymbol{v}_1(t)\boldsymbol{v}_1(t)^\top))\boldsymbol{R}'(t)$ (Lemma 3.4). While $\boldsymbol{D}$'s dynamics is $\boldsymbol{D}(t+1) = (\boldsymbol{I} - \eta\boldsymbol{M}(t))\boldsymbol{D}(t)$, $\boldsymbol{R}'$ follows a similar dynamics $\boldsymbol{R}(t+1) = (\boldsymbol{I} - \eta\boldsymbol{M}'(t))\boldsymbol{R}'(t)$, where $\boldsymbol{M}'(t) = \boldsymbol{M}(t)(\boldsymbol{I} - \boldsymbol{v}_1(t)\boldsymbol{v}_1(t)^\top)$. Note that $\boldsymbol{M}'(t)$ has eigenvalues smaller than $1/\eta$ for any time $t$ (by Assumption 3.7), hence with an assumption similar to Assumption 3.5 (i) (or a similar version of our relaxed assumption in Appendix E.2 for $\boldsymbol{M}'$), we can prove that $\boldsymbol{R}'(t)^\top(\boldsymbol{R}'(t) + \boldsymbol{Y}) < 0$ for any time $t$ (See Appendix E.1 for the rigorous proof).

Since $\boldsymbol{R}(t) \approx \boldsymbol{R}'(t)$, the second term in the decomposition (4) is always negative and the first term ($\boldsymbol{v}_1$ direction term) is decreasing geometrically. Therefore, there are only two possible cases. The first possibility is that the first term decreases to a small value near 0 and the second term remains largely negative. Then their sum will be negative, which is $\boldsymbol{D}(t)^\top\boldsymbol{F}(t) < 0$, thus implying the training enters Phase I. The second possibility is that when the first term decreases to a small value near 0, the second term is also a small negative value. In this case both $\boldsymbol{R}(t)$ and $\boldsymbol{D}(t)^\top\boldsymbol{v}_1(t)$ are small, implying the loss is almost 0, which is indeed the end of the training.

### 3.4 Explaining Non-monotonic Loss Decrement

In this subsection, we attempt to explain the non-monotonic decrement of the loss during the entire GD trajectory. See Figure 3(b). As defined in the last section, we decompose $\boldsymbol{D}$ into the $\boldsymbol{v}_1$-component $\boldsymbol{v}_1\boldsymbol{v}_1^\top\boldsymbol{D}$ and the remaining part $\boldsymbol{R}$. Below, we prove that $\boldsymbol{R}(t)$ is not affected much by the exponential growth of the loss (Proposition 3.5) in Phase II and III, and almost follows a converging trajectory (which is defined as $\boldsymbol{R}'(t)$ later in this section).

The arguments in this subsection need Assumption 3.4 and Assumption 3.5 (iii), both very consistent with the experiments. We need an additional assumption on the spectrum of $\boldsymbol{M}$.

**Assumption 3.7.** *All $\boldsymbol{M}(t)$'s eigenvalues except $\Lambda(t) = \lambda_{\max}(\boldsymbol{M}(t))$ are smaller than $1/\eta$ for all $t$.*

Recall that the largest eigenvalue is at most $\frac{B_\Lambda}{\eta}$ by Assumption 3.4. Empirically, the largest eigenvalue is an outlier in the spectrum, i.e., it is much larger than the other eigenvalues. Hence, we make Assumption 3.7 which states that all other eigenvalues are at most $1/\eta$, which is consistent with our experiments. See Figure 3(b). Similar fact is also mentioned in [28, 29].

First, we let $B_D$ be an upper bound of $\boldsymbol{D}(t)$, i.e., for all $t$, $\|\boldsymbol{D}(t)\| \le B_D$. In the two-layer linear network case, we can have an explicit form of $B_D$. (see Lemma C.9 in Appendix C.) Recall that in Assumption 3.4, $B_\Lambda$ is the upper bound of $\eta\Lambda$.

**Lemma 3.4.** *Suppose Assumption 3.5 (iii) holds. $\boldsymbol{R}(t)$ satisfies the following:*

$$\boldsymbol{R}(t+1) = (I - \eta\boldsymbol{M}(t))\boldsymbol{R}(t) + \boldsymbol{e}_1(t), \quad where \quad \|\boldsymbol{e}_1(t)\| \le 6\sqrt{\epsilon_2}\|\boldsymbol{D}(t)\|(B_\Lambda - 1)$$

**Lemma 3.5.** *Define an auxiliary sequence $\mathbf{R}'(t)$ by $\mathbf{R}'(0) = \mathbf{R}(0)$, and $\mathbf{R}'(t+1) = (\mathbf{I} - \eta \mathbf{M}(t)(\mathbf{I} - \mathbf{v}_1(t)\mathbf{v}_1(t)^\top))\mathbf{R}'(t)$. If Assumption 3.4, Assumption 3.5 (iii), Assumption 3.7 hold, and for any time $t$ there exists a quantity $\lambda_r > 0$, such that the smallest eigenvalue of $\mathbf{M}(t)$, i.e. $\lambda_{\min}(\mathbf{M}(t)) > \lambda_r$, then there exists a constant $c_r > 0$ such that $\|\mathbf{R}(t) - \mathbf{R}'(t)\| \leq c_r \frac{B_D(B_\Lambda - 1)\sqrt{\epsilon_2}}{\eta \lambda_r}$.*

Now, in light of Lemma 3.2 and Lemma 3.5, we arrive at an interesting explanation of the phenomena of non-monotonic decrease of the loss. Basically, $\mathbf{D}$ can be decomposed into the $\mathbf{v}_1$-component $\mathbf{v}_1\mathbf{v}_1^\top \mathbf{D}$ and the remaining part $\mathbf{R} = (\mathbf{I} - \mathbf{v}_1\mathbf{v}_1^\top)\mathbf{D}$. The $\mathbf{v}_1$-component may increase geometrically during the EOS (Lemma 3.2), but the behavior of the remaining part $\mathbf{R}(t)$ is close to $\mathbf{R}'(t)$, which follows the simple updating rule $\mathbf{R}'(t+1) = (\mathbf{I} - \eta \mathbf{M}(t))\mathbf{R}'(t)$, so Lemma 3.3 implies that the $\mathbf{R}$ part almost keeps decreasing during the entire trajectory (here Lemma 3.3 applies with $\mathbf{u}$ replaced by $\mathbf{R}'(t)$, noticing that the eigenvalues except the first are well below $2/\eta$). Hence, the non-monotonicity of the loss is mainly due to the $\mathbf{v}_1$-component of $\mathbf{D}$, and the rest part $\mathbf{R}$ is optimized in the classical regime (step size well below 2/(the operator norm)) and hence steadily decreases. See Figure 3(b).

# 4 A Theoretical Analysis for 2-Layer Linear NN

In this section, we aim to provide a more rigorous explanation of the EOS phenomenon in two-layer linear networks. The proof ideas follow similar high-level intuition as the proofs in Section 3.3. In particular, we can remove or replace the assumptions in Section 3.3 with arguably weaker assumptions. Due to space limit, we state our main theoretical results and elaborate their relation with the proofs in Section 3.3. The detailed settings and proof are more tedious and can be found in Appendix C.

## 4.1 Setting and basic notations

**Model:** In this section, we study a two-layer neural network with linear activation, i.e. $f(x) = \sum_{q=1}^m \frac{1}{\sqrt{m}} a_q \mathbf{w}_q x = \frac{1}{\sqrt{m}} \mathbf{A}^\top \mathbf{W} x$ where $\mathbf{W} = [\mathbf{w}_1, ..., \mathbf{w}_m]^\top \in \mathbb{R}^{m \times d}$, $\mathbf{A} = [a_1, ..., a_m] \in \mathbb{R}^m$.

**Dataset:** For simplicity, we assume $y_i = \pm 1$ for all $i \in [n]$, and $\|\mathbf{X}^\top \mathbf{X}\|_2 = \Theta(n)$. We assume $\mathbf{X}^\top \mathbf{X}$ has rank $r$, and we decompose $\mathbf{X}^\top \mathbf{X}$ and $\mathbf{Y}$ according to the orthonormal basis $\{\mathbf{v}_i\}$, the eigenvectors of $\mathbf{X}^\top \mathbf{X}$: $\mathbf{X}^\top \mathbf{X} = \sum_{i=1}^r \lambda_i \mathbf{v}_i \mathbf{v}_i^\top$, $\mathbf{Y} = \sum_{i=1}^r (\mathbf{Y}^\top \mathbf{v}_i)\mathbf{v}_i := \sum_{i=1}^r z_i \mathbf{v}_i$ where $\mathbf{v}_i$ is the eigenvector corresponding to the $i$-th largest eigenvalue $\lambda_i$ of $\mathbf{X}^\top \mathbf{X}$. $z_i = \mathbf{Y}^\top \mathbf{v}_i$ is the projection of $\mathbf{Y}$ onto the direction $\mathbf{v}_i$. Here we suppose $n \gg r$ and the global minimum $(\mathbf{A}^*, \mathbf{W}^*)$ exists.

**Update rule:** We write explicitly the GD dynamics of $\mathbf{D}(t)$: $\mathbf{D}(t+1) = (\mathbf{I} - \eta \mathbf{M}^*(t))\mathbf{D}(t)$, where $\mathbf{M}^*(t) = \frac{2}{mn}(\|\mathbf{A}(t)\|^2 \mathbf{X}^\top \mathbf{X} + \mathbf{X}^\top \mathbf{W}^\top(t)\mathbf{W}(t)\mathbf{X}) - \frac{4\eta}{n^2 m}(\mathbf{D}(t)^\top \mathbf{F}(t))\mathbf{X}^\top \mathbf{X}$ is the Gram matrix combined with second order terms.

## 4.2 Main Theorem and The Proof Sketch

**Phase I and Progressive Sharpening:**

**Assumption 4.1.** *There exists some constant $\chi > 1$, s.t. for all $i \in [r-1]$, $\lambda_i(\mathbf{X}^\top \mathbf{X}) \leq \chi \lambda_{i+1}(\mathbf{X}^\top \mathbf{X})$. Moreover, $\lambda_1(\mathbf{X}^\top \mathbf{X}) \geq 2\lambda_2(\mathbf{X}^\top \mathbf{X})$.*

**Assumption 4.2.** *There exists $\kappa = \Omega(r^{-1})$ such that $\min_{i \in [r]}\{z_i/\sqrt{n}\} \geq \kappa$.*

The first assumption is about the eigenvalue spectrum of $\mathbf{X}^\top \mathbf{X}$. [3] The second assumes that all component $z_i = \mathbf{Y}^\top \mathbf{v}_i$ are not too small.

**Theorem 1** (Informal). *Suppose Assumption 4.1, Assumption 4.2 hold, the smallest nonzero eigenvalue $\lambda_r = \lambda_r(\mathbf{X}^\top \mathbf{X}) > 0$ and $\lambda_1 = \lambda_{\max}(\mathbf{X}^\top \mathbf{X}) = c_1 n$. Then for any $\epsilon > 0$, if $m = \Omega(\frac{c_1 n^2}{\lambda_r^2})$, and $n = \Omega(\frac{\lambda_r^2}{\kappa^4 \epsilon^2})$, we have the progressive sharpening property: $\Lambda(t+1) - \Lambda(t) > 0$ for $t = 1, 2, ..., t_0 - 1$ where $t_0$ is the time when $\|\mathbf{D}(t)\|^2 \leq O(\epsilon^2)$ or $\lambda_{\max}(\mathbf{M}^*(t)) > 1/\eta$ for the first time.*

In the proof of this theorem, we show that the Gram matrix $\mathbf{M}(t) \approx \frac{2}{mn}(\|\mathbf{A}(t)\|^2 + \frac{m}{d})\mathbf{X}^\top \mathbf{X}$, which serves as a justification of Assumption 3.5 we made in Section 3.3. That shows all $\mathbf{M}(t)$

---

[3]It guarantees the gap between two adjacent eigenvalues is not very large, and there is a gap between the largest and the second largest eigenvalue. Note the second part of the assumption is a relaxed version of Assumption 3.7. In our CIFAR-10 1k-subset with samples' mean subtracted, $\lambda_1/\lambda_2 = \chi \approx 3$ (See Figure 19).

approximately share the same set of eigenvectors as $\mathbf{X}^\top \mathbf{X}$. In our proof, we also prove more rigorously that $\|\boldsymbol{A}(t)\|^2$ is an indicator of the sharpness in this simpler setting.

**Edge of Stability (Phase II - IV):**

**Assumption 4.3.** *There exists some constant $c_2 > 0$, such that $\|\boldsymbol{\Gamma}(t)\| \leq \frac{c_2}{m}$.*

This assumption is based on Theorem 1. In Theorem 1, we state that in the progressive sharpening phase, $\|\boldsymbol{\Gamma}(t)\|$ has an upper bound of $O(1/m)$. Now in the EOS phase, we assume that $\|\boldsymbol{\Gamma}(t)\|$ grows larger by at most a constant factor. Further discussions refer to Appendix D.2.2.

**Assumption 4.4.** *There exists some constant $\beta > 0$, such that $\Lambda \leq \frac{4}{\eta}(1 - \beta)$.*

This assumption is consistent with Assumption 3.4, which assumes an upper bound of the sharpness.

**Assumption 4.5.** *There exist some constant $c_3$ such that $|\boldsymbol{D}(t)^\top \boldsymbol{v}_1| > c_3 \sqrt{n}/m$ for some $t = t_0$ at the beginning of phase II.*

This assumption is in the same spirit of Assumption 3.6 with the only change of the bound in terms of $m$ and $n$. Now, we are ready to state our theorem in this stage.

**Theorem 2.** *Denote the smallest nonzero eigenvalue as $\lambda_r \triangleq \lambda_r(\mathbf{X}^\top \mathbf{X}) > 0$ and the largest eigenvalue as $\lambda_1 \triangleq \lambda_1(\mathbf{X}^\top \mathbf{X})$. Under Assumption 4.3, 4.4, 4.5, and $\lambda_1(\mathbf{X}^\top \mathbf{X}) \geq 2\lambda_2(\mathbf{X}^\top \mathbf{X})$ in Assumption 4.1, there exist constants $c_4, c_5, c_6$ such that if $n > c_6 \lambda_r \eta, m > \max\{\frac{c_4 d^2 n^2}{\lambda_r^2}, c_5 \eta\}$, then*

- *There exists $\rho = O(1)$ which depends on $c_3$ such that if $\Lambda(t_0) > \frac{2}{\eta}(1 + \rho)$ for some $t_0$, there must exist some $t_1 > t_0$ such that $\Lambda(t_1) < \frac{2}{\eta}(1 + \rho)$.*

- *If $\Lambda(t), \Lambda(t+1) > \frac{2}{\eta}(1 + \rho)$, then there is a constant $c_7 > 0$ (depending on $c_3$) such that $|\boldsymbol{D}(t+1)^\top \boldsymbol{v}_1| > |\boldsymbol{D}(t)^\top \boldsymbol{v}_1|(1 + c_7)$.*

- *Define $\boldsymbol{R}(t) := (\boldsymbol{I} - \boldsymbol{v}_1 \boldsymbol{v}_1^\top)\boldsymbol{D}(t)$, and $\boldsymbol{R}'(t) := (\boldsymbol{I} - \eta \boldsymbol{M}^*(t)(\boldsymbol{I} - \boldsymbol{v}_1 \boldsymbol{v}_1^\top))\boldsymbol{R}'(t-1)$. It holds that $\|\boldsymbol{R}(t) - \boldsymbol{R}'(t)\| = O(\frac{\sqrt{n^3} d}{\lambda_r \sqrt{m}})$.*

We can conclude the following from Theorem 2: (1) The first statement of the theorem states that if the progressive sharpening phase causes the sharpness to grow over $2/\eta$, then the sharpness eventually goes below $2/\eta$. This illustrates the regularization effect of gradient descent on the sharpness (this is consistent with the analysis of Phase III in Section 3.3). (2) The second states that $|\boldsymbol{D}(t)^\top \boldsymbol{v}_1|$ geometrically increases in Phase II and III. Note that we proved a similar Lemma 3.2 for Phase II in the more general setting in Section 3.3. (3) The third conclusion gives an upper bound for the distance between $\boldsymbol{R}(t)$'s trajectory and $\boldsymbol{R}'(t)$'s. This bound helps illustrate why $\boldsymbol{R}(t)$'s trajectory is similar with $\boldsymbol{R}'(t)$ in Phase IV of Section 3.3.

## 5 Discussions and Open Problems

In this section, we discuss the limitation of our theory and some related findings. First, our argument crucially relies on the assumption that $\|\boldsymbol{A}\|$ changes in the same direction as $\Lambda$ does most of the time. Here, we elaborate more on this point. Seeing from a longer time scale, $\|\boldsymbol{A}\|^2$ and the sharpness may have very different overall trends (See Figure (c) in 2), i.e., the sharpness oscillates around $2/\eta$ but $\|\boldsymbol{A}\|^2$ increases. Moreover, the sharpness may oscillate more frequently than $\|\boldsymbol{A}\|^2$, while the low-frequency trends seem to match well (See the late training phases in Figure (b) in 2). Currently, our theory cannot explain the high-frequency oscillation of the sharpness in Figure (b).

While we still believe the change of $\|\boldsymbol{A}\|$ is a major driving force of the change of the sharpness, other factors (such as other layers) must be taken into consideration for a complete understanding and explanation of the sharpness dynamics. We also carry out some experiments that reveal some interesting relation between the inner layers and the sharpness, which is not yet reflected in our theory. Due to space limit, we defer it to Appendix D.3.

We conclude with some open problems. It would be very interesting to remove some of our assumptions or replace them (especially those related to the spectrum of $\boldsymbol{M}$) by weaker or more natural assumptions on the data or architectures, or make some of the heuristic argument more rigorous (e.g., first order approximation of the dynamics (3)). Extending our results in Section 4 to deeper neural networks with nonlinear activation function is an intriguing and challenging open problem.

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
