# A  Experimental Setup

We provide detailed experimental setup in this section.

## A.1  Dataset

Under GPU memory constraints when computing Gram matrix of our models, we limit the size of dataset we use. The dataset is a 1,000-sample subset from CIFAR-10 (Krizhevsky et al. [17]) (`https://www.cs.toronto.edu/~kriz/cifar.html`). To make it a binary dataset, we constructed it by selecting the first 500 samples of class 0 and 1, respectively.[4] Then we label the samples by $\pm 1$. In the experiments in Appendix B and  D, we fix the objective as training on the 1000-example subset of CIFAR-10.

## A.2  Network Architecture

In general settings, we experiment with three architectures from simple models to more complicated models: one-hidden-layer linear neural network, four-hidden-layer fully-connected network, convolutional network and a ResNet18 (He et al. [11]) model. The initialization of each layer follows the default initialization of PyTorch (Paszke et al. [27]).

**Linear Network**  We first use a simple two-layer linear neural network. The hidden layer has 200 neurons. We empirically show that even a simple linear network can enter the EOS regime.

Table 1: Linear network

| Layer # | Name | Layer | In shape | Out shape |
|---|---|---|---|---|
| 1 | | `Flatten()` | $(3, 32, 32)$ | 3072 |
| 2 | fc1 | `nn.Linear(3072, 200)` | 3072 | 200 |
| 3 | fc2 | `nn.Linear(200, 1)` | 200 | 1 |

**Fully-connected Network**  We conduct further experiments on several different fully-connected networks with 4 hidden layers with various activation functions. We consider tanh, ReLU, ELU activations. For example, the structure of a fully-connected tanh network is shown in Table 2.

Table 2: Fully-connected network

| Layer # | Name | Layer | In shape | Out shape |
|---|---|---|---|---|
| 1 | | `Flatten()` | $(3, 32, 32)$ | 3072 |
| 2 | fc1 | `nn.Linear(3072,200,bias=False)` | 3072 | 200 |
| 3 | | `nn.tanh()` | 200 | 200 |
| 4 | fc2 | `nn.Linear(200,200,bias=False)` | 200 | 200 |
| 5 | | `nn.tanh()` | 200 | 200 |
| 6 | fc3 | `nn.Linear(200,200,bias=False)` | 200 | 200 |
| 7 | | `nn.tanh()` | 200 | 200 |
| 8 | fc4 | `nn.Linear(200,200,bias=False)` | 200 | 200 |
| 9 | | `nn.tanh()` | 200 | 200 |
| 10 | fc5 | `nn.Linear(200,1,bias=False)` | 200 | 1 |

**Convolutional Network**  We also conduct experiments on several different convolutional networks with two convolutional layers and two max-pooling layers. Like the fully-connected network experiments, we consider tanh, ReLU and ELU activations. For example, the structure of a convolutional tanh network is shown in Table 3.

---

[4]Cohen et al. [6] selects the first 5,000 examples from CIFAR-10. Our subset is smaller because of the computation limitation when calculating the Gram matrix. Experiments show that the properties along GD trajectory (e.g. the loss, the sharpness) is similar on both datasets.

Table 3: Convolutional network

| # | Name | Layer | In shape | Out shape |
|---|------|-------|----------|-----------|
| 1 | conv1 | nn.Conv2d(3,32,kernel_size=3,padding=1) | $(3,32,32)$ | (32,32,32) |
| 2 | | nn.tanh() | (32,32,32) | (32,32,32) |
| 3 | | nn.MaxPool2d(2) | (32,32,32) | (32,16,16) |
| 4 | conv2 | nn.Conv2d(32,32,kernel_size=3,padding=1) | (32,16,16) | (32,16,16) |
| 5 | | nn.tanh() | (32,16,16) | (32,16,16) |
| 6 | | nn.MaxPool2d(2) | (32,16,16) | (32,8,8) |
| 7 | | Flatten() | (32,8,8) | 2048 |
| 8 | fc1 | nn.Linear(2048,1,bias=False) | 2048 | 1 |

**ResNet18** We also conduct experiment on the ResNet18 architecture proposed by He et al. [11]. We use the default architecture implemented in PyTorch (Paszke et al. [27]). When calculating the sharpness, we use the numerical methods in the package (Golmant et al. [10]) to calculate the top eigenvalue of the Hessian matrix.

## B   Further Experiments and Discussions

In this appendix, we use the 1000-sized subset of CIFAR-10 introduced in Appendix A to conduct further experiments on various architectures. We verify our main observation about the correlation between the sharpness and the weight norm of the output layer (A-norm) of the neural network through the following experiments.

We consider simple linear networks, fully-connected networks, convolutional networks in this appendix. For nonlinear networks, we choose tanh, ReLU, ELU as activation functions. We train the networks with MSE loss. Here we exclude ResNet18 experiment since it is not feasible to compute the Gram matrix or the leading Gram matrix eigenvector due to GPU memory limitation. The sharpness and A-norm correlation of ResNet18 are included in Figure 2 in Section 3.2.

Here we run full-batch gradient descent with a selected step size $\eta$ such that the sharpness at initialization is smaller than $2/\eta$.

### B.1   Further Experiments

#### B.1.1   Linear Networks

We first verify our four-phase division of EOS phenomena in a simple linear network. This experiment shows that even linear networks can also enter EOS regime and the four-phase division of the gradient descent trajectory is quite apparent in this setting. The following Figure 4 illustrates the positive correlation between the sharpness and the A-norm, and the relationship between the loss $\|\boldsymbol{D}(t)\|^2$ and $\|\boldsymbol{R}(t)\|^2$ along the trajectory.

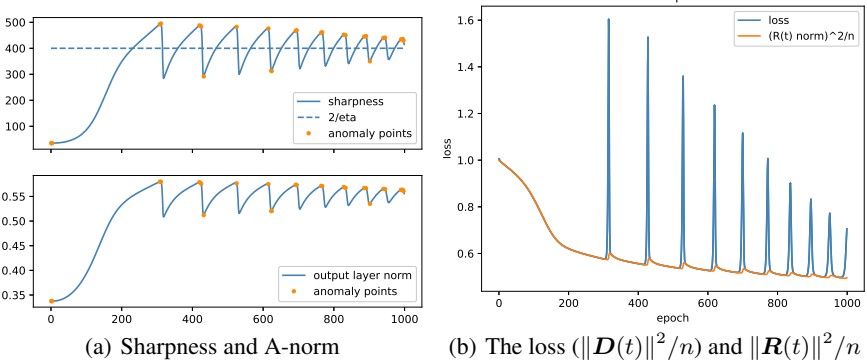

(a) Sharpness and A-norm     (b) The loss ($\|\boldsymbol{D}(t)\|^2/n$) and $\|\boldsymbol{R}(t)\|^2/n$

Figure 4: Fully-connected linear network. In Figure (a), we verify the correlation between sharpness and A-norm in convolutional network setting. In Figure (b), we show that in this case, even though the total loss $\mathcal{L}(t) = \|\boldsymbol{D}(t)\|^2/n$ oscillates considerably, $\|\boldsymbol{R}(t)\|^2/n$ decreases more steadily.

### B.1.2 Fully-connected networks

We train three 5-layer fully-connected networks with different activation functions: tanh, ReLU, and ELU activations. In Figure 5, 6, and 7, we verify that in these fully-connected networks, the sharpness is positively correlated to the dynamics output layer norm (A-norm) most of the time in the progressive sharpening stage and the first few oscillations.

Note that anomaly points appear much more frequently after a few oscillations. Meanwhile, the sharpness oscillates more frequently around $2/\eta$ in every few iterations. We further discuss the phenomenon in Appendix B.2, in which we elaborate the complicated relationship of the sharpness, A-norm and parameters of other layers.

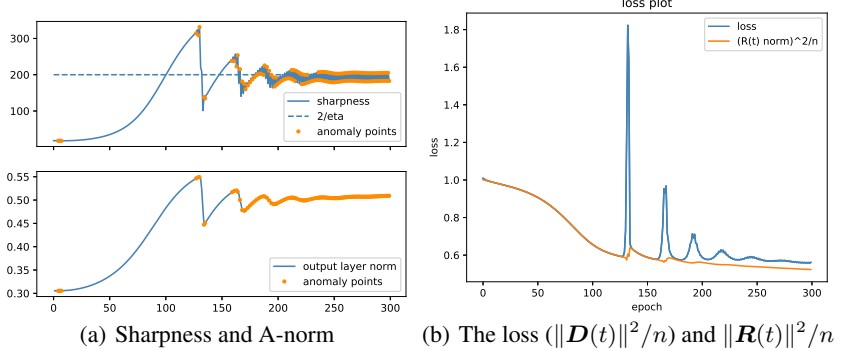

(a) Sharpness and A-norm      (b) The loss ($\|\boldsymbol{D}(t)\|^2/n$) and $\|\boldsymbol{R}(t)\|^2/n$

Figure 5: Fully-connected tanh Network. Refer to Figure 4 for more information.

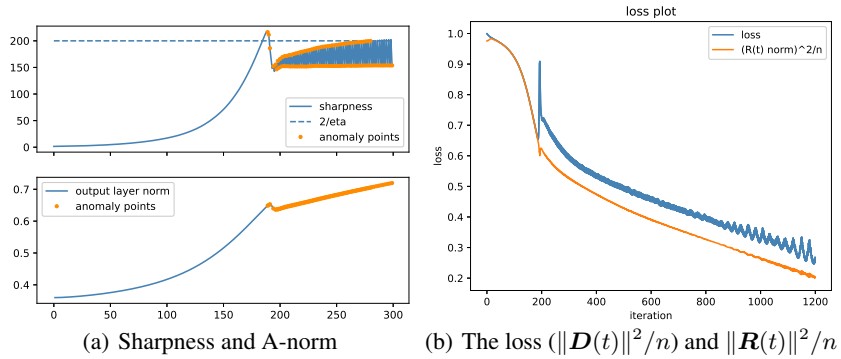

(a) Sharpness and A-norm      (b) The loss ($\|\boldsymbol{D}(t)\|^2/n$) and $\|\boldsymbol{R}(t)\|^2/n$

Figure 6: Fully-connected ReLU Network. Refer to Figure 4 for more information.

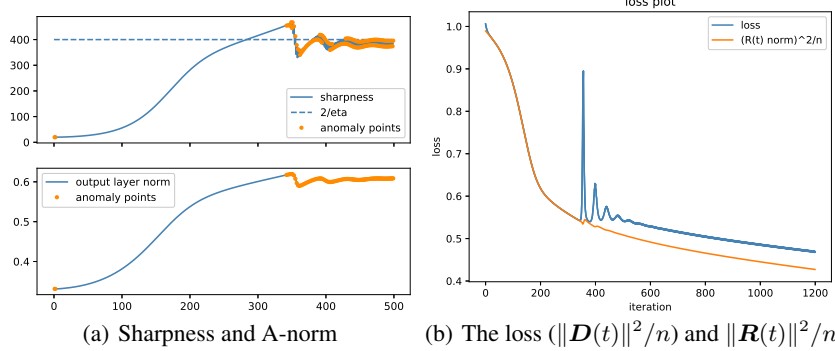

(a) Sharpness and A-norm      (b) The loss ($\|\boldsymbol{D}(t)\|^2/n$) and $\|\boldsymbol{R}(t)\|^2/n$

Figure 7: Fully-connected ELU Network. Refer to Figure 4 for more information.

### B.1.3 Convolutional Networks

We train three convolutional networks with different activation functions: ReLU, tanh, and ELU activations. In Figure 8, 9, and 10, we verify that in convolutional networks, the positive correlation between the sharpness and the output layer norm (A norm) is still correct in the training process. In the first few oscillations of the sharpness, the four-phase division is also valid. On the other hand, we notice that the same anomaly appears in this convolutional setting as in the fully-connected examples. We defer the discussion to Section B.2.

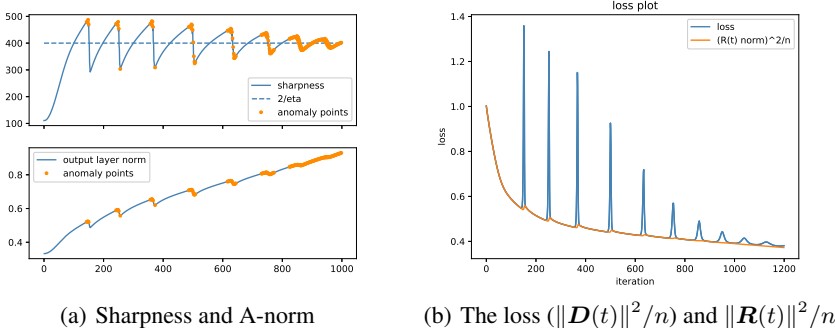

(a) Sharpness and A-norm      (b) The loss ($\|\boldsymbol{D}(t)\|^2/n$) and $\|\boldsymbol{R}(t)\|^2/n$

Figure 8: Conv. tanh Network. In Figure (a), we verify the correlation between sharpness and A-norm in convolutional network setting. In Figure (b), we show that in this case, even though the total loss $\mathcal{L}(t) = \|\boldsymbol{D}(t)\|^2/n$ oscillates considerably, $\|\boldsymbol{R}(t)\|^2/n$ decreases more steadily.

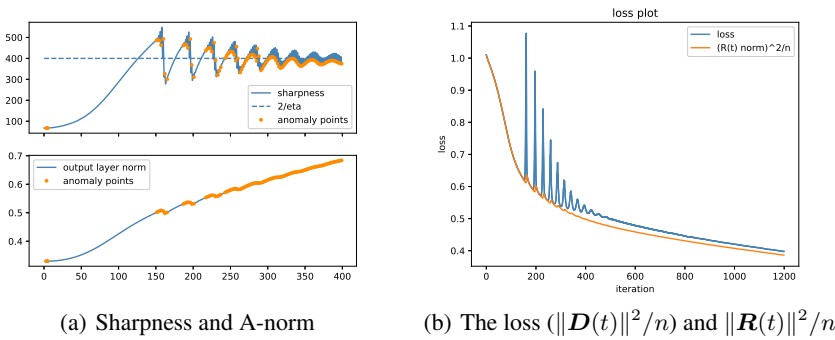

(a) Sharpness and A-norm      (b) The loss ($\|\boldsymbol{D}(t)\|^2/n$) and $\|\boldsymbol{R}(t)\|^2/n$

Figure 9: Conv. ReLU Network. Refer to Figure 8 for more information.

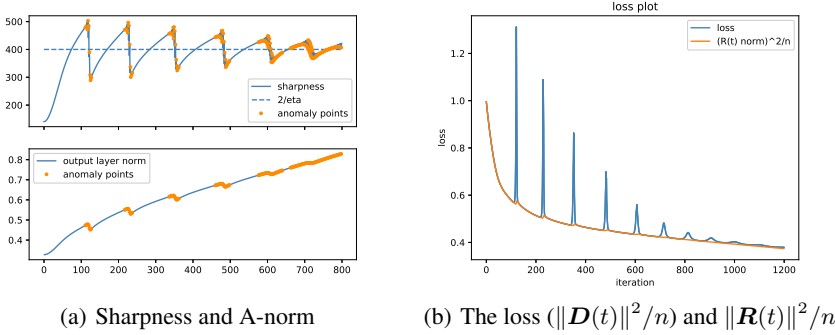

(a) Sharpness and A-norm      (b) The loss ($\|\boldsymbol{D}(t)\|^2/n$) and $\|\boldsymbol{R}(t)\|^2/n$

Figure 10: Conv. ELU Network. Refer to Figure 8 for more information.

### B.1.4 Gaussian Data on Linear Networks

In this section, we train two-layer fully-connected linear networks with datapoints sampled from Gaussian distribution and different label vectors.

In Figure 11, we train two-layer fully-connected linear networks with datapoints sampled from Gaussian distribution. In particular, the width of the network is 200. The data $\mathbf{X}$ is 1000 datapoints sampled from $\mathcal{N}(0, \mathbb{I}_{3072})$, where $\mathbb{I}_{3072} \in \mathbb{R}^{3072 \times 3072}$ is the identity matrix. The label $\mathbf{Y}$ is uniformly sampled from $\text{Unif}\{-1, 1\}^n$.

In Figure 11, we verify that even with simple two-layer linear network and Gaussian data, progressive sharpening and EOS can still be observed. However, because Gaussian data is easy for the network to learn, the convergence is so fast that within tens of epochs the training loss converges to zero. In this case, the EOS phenomenon is not quite typical.

To further explore the effect of different factors on the degree of progressive sharpening, we train the network with data points sampled from Gaussian distribution and different label vectors. In particular, the width of the network is 400. The input $\mathbf{X}$ consists of 500 data points sampled from $\mathcal{N}(\mu, \Sigma)$, where the mean $\mu \in \mathbb{R}^{3072}$ and the covariance $\Sigma \in \mathbb{R}^{3072 \times 3072}$ are the mean vector and the covariance matrix of 5000 CIFAR-10 data points, respectively. We note that to illustrate the degree of progressive sharpening, we choose a very small learning rate $\eta = 2/20000$ (so that the training converges before EOS can happen).

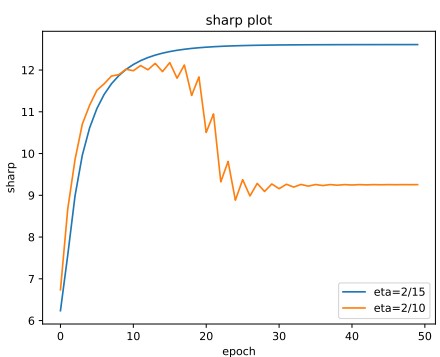

Figure 11: Two-layer Linear Network. Trained with data sampled from the standard Gaussian distribution. Both progressive sharpening and EOS are observed.

In Figure 12(a), the label $\mathbf{Y}$ is uniformly sampled from $\text{Unif}\{-1, 1\}^n$. Comparing the experimental results with those obtained using the same network trained with 500 CIFAR-10 data points, we find out that both have a similar degree of progressive sharpening. In Figure 12(b) and Figure 12(c), we let the label $\mathbf{Y}$ be $\sqrt{n}v_1$ and $\sqrt{n}v_{400}$ respectively. The result shows that when the label vector is aligned with the top eigenvector $v_1$, the degree of progressive sharpening is relatively small, and when the label vector is aligned with a bottom eigenvector, the degree of progressive sharpening is relatively large.

This phenomenon is consistent with our intuition. Empirically we found that a dataset that is easier to learn leads to faster convergence rate, which then leads to a smaller degree of progressive sharpening. For example, training standard Gaussian distributed dataset converges faster than that with CIFAR-10 dataset, hence the degree of progressive sharpening of the former is much smaller; as another example (Figure 12(b) and Figure 12(c)), if the label vector is aligned with the top eigenvector, the convergence in the first phase (the PS phase) is faster, which leads to a shorter first phase and thus a smaller degree of progressive sharpening compared to the other case.

## B.2    Further Discussions on the Relation between A-norm and Sharpness

In our paper, we use the observation that $\|\mathbf{A}\|^2$ shares the same trend as the sharpness to explain the dynamics of the sharpness. However, as shown in the experiments in this section (see Figure 5, 6, 7, 8, 9, 10), anomaly points exist during the training process. Here we briefly discussion these anomaly points further. We divide them into three kinds.

**At the time that $\|\mathbf{A}\|^2$ changes its trend:** This kind of anomaly points appear when $\|\mathbf{A}\|^2$ changes its trend. When $\|\mathbf{A}\|^2$ changes its trend, its changing rate (or differential, the $\mathbf{D}^\top \mathbf{F}$ term) changes its sign, hence the changing rate's absolute value is small. Now because the dynamics of the sharpness is effected by both $\|\mathbf{A}\|^2$ and the inner layers, when changing rate of $\|\mathbf{A}\|^2$ is small, the inner layers may play a larger role in the direction of sharpness, which may cause the anomaly.

**When the sharpness oscillates more frequently:** We notice that in some cases, $\|\mathbf{A}(t)\|^2$ and the sharpness $\Lambda(t)$ have very similar overall trend, but the sharpness oscillates more frequently but the magnitude is small (i.e., the sharpness curve has higher frequency oscillations. See Figure 6 or Figure 10). In this case, while we believe the change of $\|\mathbf{A}\|^2$ is a major driving force of the change

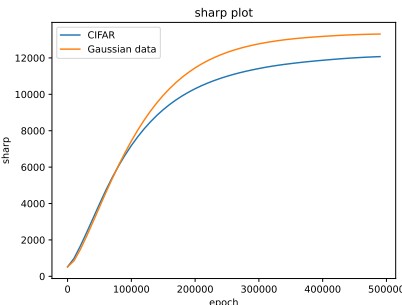 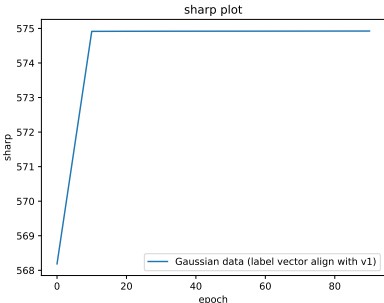

(a) Gaussian data with random label vector, com-(b) Gaussian data with label vector align with $v_1$
pared with CIFAR

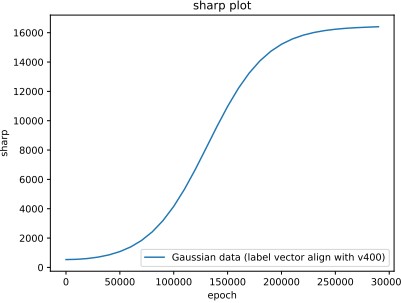

(c) Gaussian data with label vector align with $v_{400}$

Figure 12: progressive sharpening of two-layer fully-connected linear networks trained with data points sampled from anisotropic Gaussian distribution. In Figure (a), the label $Y$ is uniformly sampled from $\text{Unif}\{-1, 1\}^n$ and the result is compared with that trained with CIFAR data points. In Figure (b) and (c), the label $Y$ takes $\sqrt{n}v_1$ and $\sqrt{n}v_{400}$ respectively.

of the sharpness, other layers must be taken into consideration for understanding such small and frequent oscillations of sharpness.

**In late training phases:** We notice that in the late training phases in most settings, sharpness oscillates and crosses $2/\eta$ more frequently (it changes directions in a few iterations). In this case, our four-phase division does not strictly apply. At the same time, $\|A\|^2$ may also change direction more frequently, resulting more anomaly points during this period of time. Hence, understanding the behavior of the late stages is beyond our current analysis and requires new insights.

## C  Proof for Section 4

### C.1  Detailed Settings

**Model:** In this section, we study a two-layer neural network with linear activation, i.e.

$$f(\mathbf{x}) = \sum_{q=1}^{m} \frac{1}{\sqrt{m}} a_q \boldsymbol{w}_q \mathbf{x} = \frac{1}{\sqrt{m}} \boldsymbol{A}^\top \boldsymbol{W} \mathbf{x}$$

where $\boldsymbol{W} = [\boldsymbol{w}_1, \boldsymbol{w}_2, ..., \boldsymbol{w}_m]^\top \in \mathbb{R}^{m \times d}$ is the hidden layer's weight matrix, and $\boldsymbol{A} = [a_1, a_2, ..., a_m] \in \mathbb{R}^m$ is the weight vector of the output layer. $\mathbf{x} \in \mathbb{R}^d$ is the input vector.

**Input distribution:** Denote by $\mathbf{X} = [\mathbf{x}_1, \mathbf{x}_2, ..., \mathbf{x}_n] \in \mathbb{R}^{d \times n}$ the training data matrix and by $\boldsymbol{Y} = (y_1, y_2, ..., y_n) \in \mathbb{R}^n$ the label vector. We assume $y_i = \pm 1$ for all $i \in [n]$, and $\|\mathbf{X}^\top \mathbf{X}\| = \Theta(n)$.[5] We

---

[5]This property is mentioned in Hu et al. [12]. Empirically, for a randomly selected $k$-sample subset from CIFAR-10, $\|\mathbf{X}^\top \mathbf{X}\|/k$ is nearly constant.

assume $\mathbf{X}^\top\mathbf{X}$ has rank $r$, and we decompose $\mathbf{X}^\top\mathbf{X}$ and $\boldsymbol{Y}$ according to the orthonormal basis $\{\boldsymbol{v}_i\}$, the eigenvectors of $\mathbf{X}^\top\mathbf{X}$: $\mathbf{X}^\top\mathbf{X} = \sum_{i=1}^r \lambda_i \boldsymbol{v}_i \boldsymbol{v}_i^\top$, $\boldsymbol{Y} = \sum_{i=1}^r (\boldsymbol{Y}^\top \boldsymbol{v}_i)\boldsymbol{v}_i := \sum_{i=1}^r z_i \boldsymbol{v}_i$ where $\boldsymbol{v}_i$ is the eigenvector corresponding to the $i$-th largest eigenvalue $\lambda_i$ of $\mathbf{X}^\top\mathbf{X}$. $z_i = \boldsymbol{Y}^\top \boldsymbol{v}_i$ is the projection of $\boldsymbol{Y}$ onto the direction $\boldsymbol{v}_i$. Here we assume that $n \gg r$.

Also, we suppose there exists some parameter $\boldsymbol{A}^*, \boldsymbol{W}^*$, s.t.

$$\boldsymbol{Y}^\top = \boldsymbol{A}^{*\top}\boldsymbol{W}^*\mathbf{X}$$

Note when $n \gg d$, the matrix $\mathbf{X}^\top\mathbf{X}$ cannot be full rank. This condition guarantees that Gradient Descent (GD) only travels in the column space of $\mathbf{X}^\top\mathbf{X}$ according to Lemma E.10 at Appendix E.

Define the output vector as $\boldsymbol{F}$ and the residual vector as $\boldsymbol{D}$ in the training process.

$$\boldsymbol{F} = (f(\mathbf{x}_1), f(\mathbf{x}_2), ..., f(\mathbf{x}_n)) \in \mathbb{R}^n, \boldsymbol{D} = \boldsymbol{F} - \boldsymbol{Y}$$

(Note that they are functions of time $t$. So we use $\boldsymbol{D}(t), \boldsymbol{F}(t), \boldsymbol{A}(t), \boldsymbol{W}(t)$ to denote $\boldsymbol{D}, \boldsymbol{F}, \boldsymbol{A}, \boldsymbol{W}$ at time $t$.)

Consider the mean square error MSE loss during the training process:

$$\mathcal{L}(\boldsymbol{A}, \boldsymbol{W}) = \frac{1}{n}\sum_{i=1}^n (f(\mathbf{x}_i) - y_i)^2 = \frac{1}{n}\|\boldsymbol{D}\|^2$$

**Initialization:** We run GD on the loss and start from *symmetric initialization* for the weights, which guarantees $\boldsymbol{F}(0) = 0$.

$$a_q \sim \text{Unif}(\{-1, 1\}), \ a_{q+m/2} = -a_q, \ q = 1, 2, ..., m/2$$

$$\sum_{q=1}^{m/2} \boldsymbol{w}_q \boldsymbol{w}_q^\top = \frac{m}{2d}\boldsymbol{I}_d, \ \boldsymbol{w}_{q+m/2} = \boldsymbol{w}_q, \ q = 1, 2, ..., m/2$$

where $\boldsymbol{I}_d \in \mathbb{R}^{d\times d}$ is the identity matrix of $d$-dimension.

**Sharpness:** Recall the definition of the sharpness: $\Lambda(t) = \lambda_{\max}(\boldsymbol{M}(t))$, where $\boldsymbol{M}(t)$ is the Gram matrix (8).

**Learning rate selection:** We select a learning rate $\eta$ such that $\Lambda(0) < 2/\eta$. Specially, based on our method of initialization, $\Lambda(0) = \frac{m\lambda_1(d+1)}{d} \cdot \frac{2}{mn}$, hence we have

$$\eta < \frac{nd}{(d+1)\lambda_1} \tag{5}$$

**Gradient Descent Update Rule:** Following GD, the training dynamics is as follows:

$$a_q(t+1) - a_q(t) = -\eta\frac{\partial\mathcal{L}}{\partial a_q} = -\frac{2\eta}{n\sqrt{m}}\sum_{i=1}^n (f(\mathbf{x}_i) - y_i)\boldsymbol{w}_q\mathbf{x}_i, \ \forall q \in [m]$$

Similarly, we have

$$\boldsymbol{w}_q(t+1) - \boldsymbol{w}_q(t) = -\frac{2\eta}{n\sqrt{m}}\sum_{i=1}^n (f(\mathbf{x}_i) - y_i)a_q\mathbf{x}_i, \ \forall q \in [m]$$

Write them into matrix forms and we have:

$$\boldsymbol{A}(t+1) - \boldsymbol{A}(t) = -\frac{2\eta}{n\sqrt{m}}\boldsymbol{W}(t)\mathbf{X}\boldsymbol{D}(t)$$

$$\boldsymbol{W}(t+1) - \boldsymbol{W}(t) = -\frac{2\eta}{n\sqrt{m}}\boldsymbol{A}(t)\boldsymbol{D}(t)^\top\mathbf{X}^\top \tag{6}$$

Then we have the $\|\boldsymbol{A}(t)\|$'s dynamics according to (6):

$$\|\boldsymbol{A}(t+1)\|^2 - \|\boldsymbol{A}(t)\|^2 = -2\eta\boldsymbol{A}(t)^\top\frac{\partial\mathcal{L}}{\partial\boldsymbol{A}} + \eta^2\|\frac{\partial\mathcal{L}}{\partial\boldsymbol{A}}\|^2 = -\frac{4\eta}{n}\boldsymbol{F}(t)^\top\boldsymbol{D}(t) + \eta^2\|\frac{\partial\mathcal{L}}{\partial\boldsymbol{A}}\|^2 \tag{7}$$

where $\boldsymbol{F}(t)^\top \boldsymbol{D}(t) = \boldsymbol{D}(t)^\top \boldsymbol{F}(t) \in \mathbb{R}$ is a real number.

Define the Gram matrix as:

$$\boldsymbol{M}(t) = \frac{2}{mn}(\|\boldsymbol{A}(t)\|^2 \mathbf{X}^\top \mathbf{X} + \mathbf{X}^\top \boldsymbol{W}(t)^\top \boldsymbol{W}(t)\mathbf{X}) \tag{8}$$

We can also show the dynamics of $\boldsymbol{D}(t)$ according to (6) and (8). See Appendix E for a proof.

**Lemma C.1.** *(See Lemma E.8) The update rule of the residual vector of $\boldsymbol{D}(t)$:*

$$\boldsymbol{D}(t+1)^\top - \boldsymbol{D}(t)^\top = -\eta \boldsymbol{D}(t)^\top \boldsymbol{M}(t) + \frac{4\eta^2}{n^2 m} \boldsymbol{D}(t)^\top (\boldsymbol{F}(t)^\top \boldsymbol{D}(t)) \mathbf{X}^\top \mathbf{X} \tag{9}$$

We define some extra notations for preparation.

$$\begin{aligned}
\boldsymbol{M}^*(t) &= \boldsymbol{M}(t) - \frac{4\eta}{n^2 m}(\boldsymbol{D}(t)^\top \boldsymbol{F}(t)) \mathbf{X}^\top \mathbf{X} \\
\boldsymbol{\Gamma}(t) &= \frac{2}{mn}(\mathbf{X}^\top \boldsymbol{W}(t)^\top \boldsymbol{W}(t)\mathbf{X} - \frac{m}{d}\mathbf{X}^\top \mathbf{X}) \\
\widetilde{\boldsymbol{M}}(t) &= \boldsymbol{M}^*(t) - \boldsymbol{\Gamma}(t)
\end{aligned} \tag{10}$$

**Remark.** Here we explain the notations above and what they are for.

- $\boldsymbol{M}^*(t)$ is the Gram matrix $\boldsymbol{M}(t)$ plus a second order term $(-\frac{4\eta}{n^2 m}(\boldsymbol{D}(t)^\top \boldsymbol{F}(t))\mathbf{X}^\top \mathbf{X})$. Since the second order term is small, we have $\boldsymbol{M}(t) \approx \boldsymbol{M}^*(t)$. Also, in Section C.3 we prove $\boldsymbol{M}^*(t)$ is almost a linear interpolation of $\boldsymbol{M}(t)$ and $\boldsymbol{M}(t+1)$, which corroborates this argument.

- $\boldsymbol{\Gamma}(t)$ is the difference between $\frac{2}{mn}\mathbf{X}^\top \boldsymbol{W}(t)^\top \boldsymbol{W}(t)\mathbf{X}$ and $\frac{2}{mn}\mathbf{X}^\top \boldsymbol{W}(0)^\top \boldsymbol{W}(0)\mathbf{X}$. Note that $\boldsymbol{\Gamma}(0) = 0$ because of the initialization condition. In the following proof, we will show a small upper bound of the norm of $\boldsymbol{\Gamma}(t)$. Hence, we can show that the eigenvectors of $\boldsymbol{M}(t)$ are approximately aligned with the eigenvectors of $\mathbf{X}^\top \mathbf{X}$.

- $\widetilde{\boldsymbol{M}}(t)$ is the Gram matrix with the second order term, yet with $\boldsymbol{\Gamma}(t)$ excluded. $\widetilde{\boldsymbol{M}}(t)$ is used for bounding the main part of $\boldsymbol{\Gamma}(t)$ (i.e., the terms without the noise $\boldsymbol{e}(t)$, e.g. (A1),(A5)) which is defined in Theorem 3) in the proof of Theorem 3. Note that there exists some number $\gamma$ s.t. $\widetilde{\boldsymbol{M}}(t) = \gamma \mathbf{X}^\top \mathbf{X}$. This is because

$$\widetilde{\boldsymbol{M}}(t) = \boldsymbol{M}^*(t) - \boldsymbol{\Gamma}(t) = \frac{2}{mn}(\|\boldsymbol{A}(t)\|^2 + \frac{m}{d} - \frac{2\eta}{n}(\boldsymbol{D}(t)^\top \boldsymbol{F}(t)))\mathbf{X}^\top \mathbf{X} \tag{11}$$

Hence for any eigenvector $\boldsymbol{v}_i$ of $\mathbf{X}^\top \mathbf{X}$, $\lambda_i(\widetilde{\boldsymbol{M}}(t)) = \boldsymbol{v}_i^\top \widetilde{\boldsymbol{M}}(t)\boldsymbol{v}_i$.

Then the dynamics of $\boldsymbol{D}(t)$ can be written as:

$$\boldsymbol{D}(t+1) = (\boldsymbol{I}_n - \eta \boldsymbol{M}^*(t))\boldsymbol{D}(t) \tag{12}$$

**Update Rule of $\boldsymbol{M}(t)$:** Using the update rule of $\boldsymbol{A}(t)$ and $\boldsymbol{W}(t)$, we can also derive the update rule of the Gram matrix $\boldsymbol{M}(t)$ (see Appendix E for the proof detail):

**Lemma C.2.** *(See Lemma E.9) The update rule of the Gram matrix $\boldsymbol{M}(t)$ is,*

$$\begin{aligned}
\boldsymbol{M}(t+1) - \boldsymbol{M}(t) = &-\frac{4\eta}{n^2 m}\left(2(\boldsymbol{F}(t)^\top \boldsymbol{D}(t))\mathbf{X}^\top \mathbf{X} + \boldsymbol{F}(t)\boldsymbol{D}(t)^\top \mathbf{X}^\top \mathbf{X} + \mathbf{X}^\top \mathbf{X}\boldsymbol{D}(t)\boldsymbol{F}(t)^\top\right) \\
&+ \frac{8\eta^2}{n^3 m^2}(\boldsymbol{D}(t)^\top \mathbf{X}^\top \boldsymbol{W}(t)^\top \boldsymbol{W}(t)\mathbf{X}\boldsymbol{D}(t))\mathbf{X}^\top \mathbf{X} \\
&+ \frac{8\eta^2}{n^3 m^2}\|\boldsymbol{A}(t)\|^2 \mathbf{X}^\top \mathbf{X}\boldsymbol{D}(t)\boldsymbol{D}(t)^\top \mathbf{X}^\top \mathbf{X}
\end{aligned}$$

## C.2 Phase I and Progressive Sharpening

We suppose $\mathbf{X}^\top\mathbf{X}$ has rank $r$, and we decompose $\mathbf{X}^\top\mathbf{X}$ and $\mathbf{Y}$ into a basis $\{\boldsymbol{v}_i\}$ composed of orthonormal eigenvectors of $\mathbf{X}^\top\mathbf{X}$.

$$\mathbf{X}^\top\mathbf{X} = \sum_{i=1}^{r} \lambda_i \boldsymbol{v}_i \boldsymbol{v}_i^\top, \quad \mathbf{Y} = \sum_{i=1}^{r} (\mathbf{Y}^\top \boldsymbol{v}_i) \boldsymbol{v}_i := \sum_{i=1}^{r} z_i \boldsymbol{v}_i$$

where $\boldsymbol{v}_i$ is the eigenvector corresponding to the $i$-th largest eigenvalue $\lambda_i$ of $\mathbf{X}^\top\mathbf{X}$. $z_i = \mathbf{Y}^\top \boldsymbol{v}_i$ is the $\mathbf{Y}$'s projection onto the direction $\boldsymbol{v}_i$. In particular, $\boldsymbol{v}_1$ is the eigenvector corresponding to the largest eigenvalue of $\mathbf{X}^\top\mathbf{X}$.

Recall that the sharpness $\Lambda(t) = \lambda_{\max}(\boldsymbol{M}(t))$. Here we propose an approximation of sharpness $\Lambda$ by the eigenvector $\boldsymbol{v}_1$:

$$\Lambda^*(t) = \boldsymbol{v}_1^\top \boldsymbol{M}(t) \boldsymbol{v}_1 \tag{13}$$

The following corollary rationalizes this approximation. The proof is deferred to Corollary C.2 in Section C.3.

**Corollary C.1.** *There exists some constant $c_2, c_4$, s.t. for all time $t$ $\langle \boldsymbol{v}_1, \boldsymbol{u} \rangle^2 \geq 1 - c_4 \sqrt{\frac{1}{nm}}$ and*

$$\Lambda(t) \geq \Lambda^*(t) \geq \Lambda(t) - \frac{2c_2}{m}$$

*where $\boldsymbol{u}$ is the top eigenvector of $\boldsymbol{M}(t)$.*

By this corollary, we can see that the top eigenvector of $\boldsymbol{M}(t)$ is approximately $\boldsymbol{v}_1$, and the approximation $\Lambda^*(t)$ is very close to the real sharpness $\Lambda(t)$. Thus, we consider $\Lambda^*(t)$ as an approximation of the sharpness and analyze its dynamics.

We first show the main theorem of progressive sharpening under Assumption C.1 and C.2 below.

**Assumption C.1.** *There exists some constant $\chi > 1$, s.t. for all $i \in [r-1]$, $\lambda_i(\mathbf{X}^\top\mathbf{X}) \leq \chi\lambda_{i+1}(\mathbf{X}^\top\mathbf{X})$. Moreover, $\lambda_1(\mathbf{X}^\top\mathbf{X}) \geq 2\lambda_2(\mathbf{X}^\top\mathbf{X})$.*

**Assumption C.2.** *There exists $\kappa = \Omega(r^{-1})$ such that $\min_{i\in[r]}\{z_i/\sqrt{n}\} \geq \kappa$.*

The first assumption is about the eigenvalue spectrum of $\mathbf{X}^\top\mathbf{X}$. It guarantees the gap between two adjacent eigenvalues is not very large, and there is a gap between the largest and the second largest eigenvalue. Note the second part of the assumption is a relaxed version of Assumption 3.7. In our CIFAR-10 1k-subset with samples' mean subtracted, $\lambda_1/\lambda_2 = \chi \approx 3$ (See Appendix D.2.1, Figure 19). The second assumes that all components $z_i = \mathbf{Y}^\top \boldsymbol{v}_i$ are not too small.

Under the two assumptions, we have the following theorem:

**Theorem 3 (Progressive Sharpening).** *Suppose Assumption C.1, C.2 hold. Suppose $\lambda_r = \lambda_{\min}(\mathbf{X}^\top\mathbf{X}) > 0, \lambda_1 = \lambda_{\max}(\mathbf{X}^\top\mathbf{X}) = \Theta(n)$. For any $\epsilon > 0$, if $m = \Omega(\frac{n^2}{\lambda_r^2})$, and $n = \max\{\Omega(\frac{\lambda_r}{\kappa^2}), \Omega(\frac{\lambda_r^2}{\kappa^2}), \Omega(\frac{\lambda_r^2}{\kappa^4\epsilon^2}), \Omega(\frac{\lambda_r^4}{\kappa^4\epsilon^2})\}$, we have the following properties for $t = 1, 2, ..., t_0 - 1$ where $t_0$ is the time when $\|\boldsymbol{D}(t)\|^2 = O(\epsilon^2)$ or $\lambda_{\max}(\boldsymbol{M}^*(t)) > 1/\eta$ for the first time.*

- *(Progressive Sharpening) (Lemma C.6) $\Lambda^*(t+1) - \Lambda^*(t) > 0$.*

- *(Lemma C.3) $\|\boldsymbol{\Gamma}(t)\| = O(1/m)$.*

We first prove Lemma C.3, which implies that $\boldsymbol{D}(t)$ approximately converges independently in each direction of $\mathbf{X}^\top\mathbf{X}$'s non-zero eigenspace. Lemma C.3 also proves the second point $\|\boldsymbol{\Gamma}(t)\| \leq O(1/m)$. (Recall that $\|\boldsymbol{\Gamma}(t)\|$ is defined in (10).)

Then, based on the conclusion in Lemma C.3, Assumption C.1 and C.2, we prove that $\|\boldsymbol{A}(t)\|^2$ grows in Lemma C.4 and Lemma C.5. Specifically, here we prove a sufficient condition of $-\boldsymbol{D}(t)^\top \boldsymbol{F}(t) > 0$.

Finally, we use the dynamics of $\Lambda^*(t)$ and its dependence on $\|\boldsymbol{A}(t)\|^2$ dynamics to prove the sharpness grows (Lemma C.6), thus finishing the proof.

**Remark 1.** In the theorem, $\lambda_{\max}(\boldsymbol{M}^*(t)) > 1/\eta$ is the termination condition for this theorem. Here, $\boldsymbol{M}^*(t)$ is the Gram matrix $\boldsymbol{M}(t)$ plus the second order term $(-\frac{4\eta}{n^2 m}(\boldsymbol{D}(t)^\top \boldsymbol{F}(t))\mathbf{X}^\top \mathbf{X})$. Since the second order term is small, we have $\boldsymbol{M}(t) \approx \boldsymbol{M}^*(t)$. Also, in Section C.3, we prove $\boldsymbol{M}^*(t)$ is almost a linear interpolation of $\boldsymbol{M}(t)$ and $\boldsymbol{M}(t+1)$. Thus, $\lambda_{\max}(\boldsymbol{M}^*(t))$ can be seen as an approximation of the sharpness.

**Remark 2.** From the experiments, one can see that gradient descent is in progressive sharpening phase until the sharpness crosses the threshold $2/\eta$. Right now, our proof only works till $\lambda_{\max}(\boldsymbol{M}^*(t))$ reaches $1/\eta$. It would be interesting to extend our result to $2/\eta$.

**Remark 3.** In this theorem, we require mild over-parameterization ($m = \Omega(n^2)$, assuming $\lambda_r = \Theta(1)$, to prove the direction guarantee of the Gram matrix $\boldsymbol{M}(t)$ and the monotone increment of $\|\boldsymbol{A}(t)\|^2$. Astute readers may find it similar to the NTK regime, where the parameters do not move far from the initialization. However, we stress that our analysis does not necessarily require NTK regime, and can go beyond NTK. We defer the discussion on the difference between our results and the NTK regime to Appendix C.4.

Then we break the proof of theorem into lemmas. The first lemma (Lemma C.3) proves that the Gram matrix $\boldsymbol{M}(t) \approx \frac{2}{mn}(\|\boldsymbol{A}(t)\|^2 + \frac{m}{d})\mathbf{X}^\top \mathbf{X}$ by bounding the movement of $\|\boldsymbol{\Gamma}(t)\|$. In this way, $\boldsymbol{D}(t) \approx -\prod_{j=0}^{t-1}(\boldsymbol{I}_n - \eta \widetilde{\boldsymbol{M}}(j))\boldsymbol{Y}$ can approximately descend in each eigenvector $\boldsymbol{v}_i$ of $\mathbf{X}^\top \mathbf{X}$ independently. Also, by proving $\boldsymbol{M}(t) \approx \frac{2}{mn}(\|\boldsymbol{A}(t)\|^2 + \frac{m}{d})\mathbf{X}^\top \mathbf{X}$, we justify $\|\boldsymbol{A}(t)\|^2$ as an indicator of the sharpness.

**Lemma C.3** (**Direction Guarantee**). *Along GD training trajectory, if $m \geq \frac{112c_1 n^2}{\lambda_r^2}$, we have the following properties until $\lambda_{\max}(\boldsymbol{M}^*(t)) > 1/\eta$:*

1. *$\|\boldsymbol{A}(t)\|^2 \geq m/2$ and $\|\boldsymbol{\Gamma}(t)\| \leq R_w := 40/m$.*

2. *$\boldsymbol{D}(t) = -\prod_{j=0}^{t-1}(\boldsymbol{I}_n - \eta \widetilde{\boldsymbol{M}}(j))\boldsymbol{Y} + \boldsymbol{e}(t)$, where $\|\boldsymbol{e}(t)\|_2 \leq \frac{40 n^{3/2}}{\lambda_r m}$*

3. *$\lambda_r(\widetilde{\boldsymbol{M}}(t)) \geq \lambda_r/n$, $\lambda_r(\boldsymbol{M}^*(t)) \geq \lambda_r/n$.*

*Proof.* We prove the theorem by induction.

We first consider the base case. For property 1, $\|\boldsymbol{A}(0)\|^2 = m$ and $\|\boldsymbol{\Gamma}(0)\| = 0$. For property 2, with *symmetric initialization*, $\boldsymbol{F}(0) = 0$, hence $\boldsymbol{D}(0) = -\boldsymbol{Y}$, $\|\boldsymbol{e}(0)\| = 0$. For property 3,

$$\boldsymbol{M}^*(0) = \widetilde{\boldsymbol{M}}(0) = \boldsymbol{M}(0) \succeq \frac{2}{nm}\|\boldsymbol{A}(0)\|^2 \mathbf{X}^\top \mathbf{X}$$

The minimal eigenvalue of $\frac{2}{nm}\|\boldsymbol{A}(0)\|^2 \mathbf{X}^\top \mathbf{X}$ is $2\lambda_r/n > \lambda_r/n$. Thus all the properties hold at iteration $t = 0$.

Suppose for all $k \leq t$, these properties hold. Then we consider the case in iteration $t+1$.

We first show a worst-case upper bound for $\|\boldsymbol{D}(t)\|$.

$$\|\boldsymbol{D}(t)\| = \left\|\prod_{k=0}^{t-1}(\boldsymbol{I} - \eta \boldsymbol{M}^*(k))\boldsymbol{D}(0)\right\|$$

$$\leq \left\|\prod_{k=0}^{t-1}(\boldsymbol{I} - \eta \boldsymbol{M}^*(k))\right\| \|\boldsymbol{D}(0)\|$$

$$\leq (1 - \frac{\eta \lambda_r}{n})^t \|\boldsymbol{Y}\| = \sqrt{n}(1 - \frac{\eta \lambda_r}{n})^t \qquad (D)$$

The second inequality uses property 3 in $k \leq t$. Note $\|\boldsymbol{Y}\| = \sqrt{n}$ in our setting, so the last equality holds. That means $\|\boldsymbol{D}(t)\| \leq \sqrt{n}$ and $\|\boldsymbol{F}(t)\| = \|\boldsymbol{D}(t) + \boldsymbol{Y}\| \leq 2\sqrt{n}$.

Now, we show the error $\boldsymbol{e}(t)$ is bounded. We have

$$\boldsymbol{D}(t+1) = (\boldsymbol{I} - \eta\widetilde{\boldsymbol{M}}(t))\boldsymbol{D}(t) - \eta\boldsymbol{\Gamma}(t)\boldsymbol{D}(t)$$

$$= -\prod_{j=0}^{t}(\boldsymbol{I} - \eta\widetilde{\boldsymbol{M}}(j))\boldsymbol{Y} + (\boldsymbol{I} - \eta\widetilde{\boldsymbol{M}}(t))\boldsymbol{e}(t) - \eta\boldsymbol{\Gamma}(t)\boldsymbol{D}(t)$$

$$= -\prod_{j=0}^{t}(\boldsymbol{I} - \eta\widetilde{\boldsymbol{M}}(j))\boldsymbol{Y} + (\boldsymbol{I} - \eta\boldsymbol{M}^*(t))\boldsymbol{e}(t) - \eta\boldsymbol{\Gamma}(t)\prod_{j=0}^{t-1}(\boldsymbol{I} - \eta\widetilde{\boldsymbol{M}}(j))\boldsymbol{D}(0).$$

Hence we have $\boldsymbol{e}(t+1) = \boldsymbol{e}(t)(\boldsymbol{I} - \eta\boldsymbol{M}^*(t)) - \eta\boldsymbol{\Gamma}(t)\prod_{j=0}^{t-1}(\boldsymbol{I} - \eta\widetilde{\boldsymbol{M}}(j))\boldsymbol{D}(0)$.

Then we can have the following bound by this recursion:

$$\|\boldsymbol{e}(t+1)\|_2 \le \|\boldsymbol{e}(t)\|_2 \|\boldsymbol{I} - \eta\boldsymbol{M}^*(t)\|_2 + \eta\|\boldsymbol{D}(0)\|_2 \left\|\prod_{j=0}^{t-1}(\boldsymbol{I} - \eta\widetilde{\boldsymbol{M}}(j))\right\|_2 \|\boldsymbol{\Gamma}(t)\|_2$$

$$\le \|\boldsymbol{e}(t)\|_2 \left(1 - \frac{\eta\lambda_r}{n}\right) + \eta\sqrt{n}\left(1 - \frac{\eta\lambda_r}{n}\right)^t R_w$$

$$\le \sum_{k=0}^{t} \eta\sqrt{n}\left(1 - \frac{\eta\lambda_r}{n}\right)^t R_w \tag{E1}$$

$$\le \frac{\sqrt{n}}{\frac{1}{n}\lambda_r} \cdot R_w$$

$$\le \frac{40n^{3/2}}{m\lambda_r}$$

Here the third inequality holds because $\|\boldsymbol{e}(0)\| = 0$. Thus the second property holds for $t+1$.

Then we consider the lower bound of $\|\boldsymbol{A}(t)\|^2$.

Consider the dynamics of $\|\boldsymbol{A}(t)\|^2$ in (7) and sum the difference up from 0 to $t$. Recall that we proved $\|\boldsymbol{D}(t)\| \le \sqrt{n}$ by (D) above.

$$\|\boldsymbol{A}(t+1)\|^2 - \|\boldsymbol{A}(0)\|^2 = \frac{4\eta}{n}\sum_{k=0}^{t}(-\boldsymbol{D}(k)^\top \boldsymbol{F}(k)) + \eta^2\sum_{k=0}^{t}\|\frac{\partial\mathcal{L}(k)}{\partial\boldsymbol{A}(k)}\|^2$$

$$\ge -\frac{4\eta}{n}\sum_{k=0}^{t}\boldsymbol{D}(k)^\top(\boldsymbol{D}(k) + \boldsymbol{Y}) \qquad [\|\frac{\partial\mathcal{L}(k)}{\partial\boldsymbol{A}(k)}\|^2 > 0]$$

$$\ge -\frac{4\eta}{n}\sum_{k=0}^{t}(\|\boldsymbol{D}(k)\|^2 + \|\boldsymbol{D}(k)\|\|\boldsymbol{Y}\|) \qquad [\ell_2\text{-norm inequality}]$$

$$\ge -\frac{4\eta}{n}\sum_{k=0}^{t}\|\boldsymbol{D}(k)\|(\sqrt{n} + \sqrt{n}) \qquad [\|\boldsymbol{D}(t)\| \le \sqrt{n}]$$

$$\ge -\frac{8\eta}{n}\sum_{k=0}^{t}n(1 - \frac{\eta\lambda_r}{n})^k \qquad [\text{ by (D)}]$$

$$\ge -\frac{8n}{\lambda_r} \ge -\frac{m}{2} \tag{#}$$

Since $\|\boldsymbol{A}(0)\|^2 = m$, we have the lower bound $\|\boldsymbol{A}(t)\|^2 \ge m/2$.

Next, we lower bound the minimal non-zero eigenvalue (i.e. the $r$-th largest eigenvalue since $\mathbf{X}^\top\mathbf{X}$ is rank $r$) of $\boldsymbol{M}^*(t+1)$ and $\widetilde{\boldsymbol{M}}(t+1)$.

For $\boldsymbol{M}^*(t+1)$ and $\widetilde{\boldsymbol{M}}(t+1)$, since the $\mathbf{X}^\top\boldsymbol{W}(t)^\top\boldsymbol{W}(t)\mathbf{X}$ part is PSD, we have

$$\boldsymbol{M}^*(t+1) \succeq \frac{2}{mn}\|\boldsymbol{A}(t+1)\|^2 \mathbf{X}^\top \mathbf{X} - \frac{4\eta}{n^2 m}(\boldsymbol{D}(t+1)^\top \boldsymbol{F}(t+1))\mathbf{X}^\top \mathbf{X}$$

$$\widetilde{\boldsymbol{M}}(t+1) \succeq \frac{2}{mn}\|\boldsymbol{A}(t+1)\|^2 \mathbf{X}^\top \mathbf{X} - \frac{4\eta}{n^2 m}(\boldsymbol{D}(t+1)^\top \boldsymbol{F}(t+1))\mathbf{X}^\top \mathbf{X}$$

From the inequality in (#), and from (D) $\|\boldsymbol{D}(t)\| \leq \sqrt{n}, \|\boldsymbol{F}(t)\| \leq 2\sqrt{n}$, we know

$$\|\boldsymbol{A}(t+1)\|^2 - \|\boldsymbol{A}(0)\|^2 - \frac{2\eta}{n}(\boldsymbol{D}(t+1)^\top \boldsymbol{F}(t+1)) \geq -\frac{8n}{\lambda_r} - \frac{2\eta}{n}\sqrt{n}\cdot 2\sqrt{n} \geq -m/2$$

$$\boldsymbol{M}^*(t+1) \succeq \frac{2}{mn}\left(\|\boldsymbol{A}(t+1)\|^2 - \frac{2\eta}{n}(\boldsymbol{D}(t+1)^\top \boldsymbol{F}(t+1))\right)\mathbf{X}^\top \mathbf{X} \succeq \frac{1}{n}\mathbf{X}^\top \mathbf{X}$$

Thus $\boldsymbol{M}^*(t+1)$ (and also $\widetilde{\boldsymbol{M}}(t+1)$) has its smallest eigenvalue larger than $\lambda_r/n$. Property 3 holds. To extend this conclusion, we actually have $\lambda_i(\widetilde{\boldsymbol{M}}(t+1)) \geq \lambda_i/n$ for all $i \in [r]$ from the argument above.

Finally we bound $\|\boldsymbol{\Gamma}(t+1)\|$. We first write down the dynamics of $\boldsymbol{\Gamma}(t)$ according to Lemma C.2.

$$
\begin{aligned}
\boldsymbol{\Gamma}(t+1) - \boldsymbol{\Gamma}(t) &= \frac{2}{nm}\left(\mathbf{X}^\top(\boldsymbol{W}(t+1)^\top \boldsymbol{W}(t+1) - \boldsymbol{W}(t)^\top \boldsymbol{W}(t))\mathbf{X}\right) \\
&= \frac{2}{nm}\left(\mathbf{X}^\top((\boldsymbol{W}(t+1)^\top - \boldsymbol{W}(t)^\top)\boldsymbol{W}(t) + \boldsymbol{W}(t)^\top(\boldsymbol{W}(t+1) - \boldsymbol{W}(t)))\mathbf{X}\right. \\
&\quad \left. + \mathbf{X}^\top(\boldsymbol{W}(t+1)^\top - \boldsymbol{W}(t)^\top)(\boldsymbol{W}(t+1) - \boldsymbol{W}(t))\mathbf{X}\right) \quad \text{(Use Equation (6))} \\
&= -\frac{4}{n^2 m}\eta\left(\boldsymbol{F}(t)\boldsymbol{D}(t)^\top \mathbf{X}^\top \mathbf{X} + \mathbf{X}^\top \mathbf{X} \boldsymbol{D}(t)\boldsymbol{F}(t)^\top\right) \\
&\quad + \frac{8\eta^2}{n^3 m^2}\|\boldsymbol{A}(t)\|^2 \cdot \mathbf{X}^\top \mathbf{X} \boldsymbol{D}(t)\boldsymbol{D}(t)^\top \mathbf{X}^\top \mathbf{X}
\end{aligned}
$$

Hence we sum it up and get:

$$
\begin{aligned}
\|\boldsymbol{\Gamma}(t+1) - \boldsymbol{\Gamma}(0)\|_2 &= \left\|\frac{4}{n^2 m}\eta \sum_{k=0}^{t}\left(\boldsymbol{F}(t)\boldsymbol{D}(t)^\top \mathbf{X}^\top \mathbf{X} + \mathbf{X}^\top \mathbf{X} \boldsymbol{D}(t)\boldsymbol{F}(t)^\top\right)\right. \\
&\quad \left. + \frac{8\eta^2}{n^3 m^2}\sum_{k=0}^{t}\|\boldsymbol{A}(t)\|^2 \cdot \mathbf{X}^\top \mathbf{X} \boldsymbol{D}(t)\boldsymbol{D}(t)^\top \mathbf{X}^\top \mathbf{X}\right\|_2 \quad \text{(Triangle Inequality)} \\
&\leq \frac{4}{n^2 m}\eta \left\|\sum_{k=0}^{t}\left(\boldsymbol{D}(t)\boldsymbol{D}(t)^\top \mathbf{X}^\top \mathbf{X} + \mathbf{X}^\top \mathbf{X} \boldsymbol{D}(t)\boldsymbol{D}(t)^\top\right)\right\|_2 &\text{(G1)} \\
&\quad + \frac{4}{n^2 m}\eta \left\|\sum_{k=0}^{t}\left(\boldsymbol{Y}\boldsymbol{D}(t)^\top \mathbf{X}^\top \mathbf{X} + \mathbf{X}^\top \mathbf{X} \boldsymbol{D}(t)\boldsymbol{Y}^\top\right)\right\|_2 &\text{(G2)} \\
&\quad + \frac{8\eta^2}{n^3 m^2}\left\|\sum_{k=0}^{t}\|\boldsymbol{A}(t)\|^2 \cdot \mathbf{X}^\top \mathbf{X} \boldsymbol{D}(t)\boldsymbol{D}(t)^\top \mathbf{X}^\top \mathbf{X}\right\|_2 &\text{(G3)}
\end{aligned}
$$

Then we bound these three terms one by one.

Term (G1): $\frac{4}{n^2 m}\eta \left\|\sum_{k=0}^{t}\left(\boldsymbol{D}(t)\boldsymbol{D}(t)^\top \mathbf{X}^\top \mathbf{X} + \mathbf{X}^\top \mathbf{X} \boldsymbol{D}(t)\boldsymbol{D}(t)^\top\right)\right\|_2$

By symmetry, we just consider the first term $\boldsymbol{D}^\top(t)\boldsymbol{D}(t)\mathbf{X}^\top \mathbf{X}$.

$$\boldsymbol{D}(t)\boldsymbol{D}(t)^\top \mathbf{X}^\top \mathbf{X} = \prod_{j=0}^{t-1}(\boldsymbol{I}_n - \eta\widetilde{\boldsymbol{M}}(j))\boldsymbol{Y}\boldsymbol{Y}^\top \prod_{j=0}^{t-1}(\boldsymbol{I}_n - \eta\widetilde{\boldsymbol{M}}(j))\mathbf{X}^\top \mathbf{X} \tag{A1}$$

$$+ \boldsymbol{e}(t)\boldsymbol{Y}^\top \prod_{j=0}^{t-1}(\boldsymbol{I}_n - \eta\widetilde{\boldsymbol{M}}(j))\mathbf{X}^\top\mathbf{X} \tag{A2}$$

$$+ \prod_{j=0}^{t-1}(\boldsymbol{I}_n - \eta\widetilde{\boldsymbol{M}}(j))\boldsymbol{Y}\boldsymbol{e}(t)^\top\mathbf{X}^\top\mathbf{X} \tag{A3}$$

$$+ \boldsymbol{e}(t)\boldsymbol{e}(t)^\top\mathbf{X}^\top\mathbf{X} \tag{A4}$$

We bound each term in $\ell_2$-norm and then add them up.

Term (A1)

$$\frac{4\eta}{n^2 m}\left\|\sum_{k=0}^{t}\prod_{j=0}^{k-1}(\boldsymbol{I}_n - \eta\widetilde{\boldsymbol{M}}(j))\boldsymbol{Y}\boldsymbol{Y}^\top\prod_{s=0}^{k-1}(\boldsymbol{I}_n - \eta\widetilde{\boldsymbol{M}}(s))\mathbf{X}^\top\mathbf{X}\right\|_2$$

$$=\frac{4\eta}{n^2 m}\left\|\sum_{k=0}^{t}\prod_{j=0}^{k-1}(\boldsymbol{I}_n - \eta\widetilde{\boldsymbol{M}}(j))\sum_{i=1}^{r}z_i\boldsymbol{v}_i\sum_{j'=1}^{r}z_{j'}\boldsymbol{v}_{j'}^\top\prod_{s=0}^{k-1}(\boldsymbol{I}_n - \eta\widetilde{\boldsymbol{M}}(s))\mathbf{X}^\top\mathbf{X}\right\|_2$$

$$=\frac{4\eta}{n^2 m}\left\|\sum_{k=0}^{t}\left(\sum_{i=1}^{r}\prod_{j=0}^{k-1}(1 - \eta\lambda_i(\widetilde{\boldsymbol{M}}(j)))z_i\boldsymbol{v}_i\right)\left(\sum_{j'=1}^{r}\boldsymbol{v}_{j'}^\top z_{j'}\prod_{s=0}^{k-1}(1 - \eta\lambda_{j'}(\widetilde{\boldsymbol{M}}(s)))\lambda_{j'}\right)\right\|_2$$

where $\lambda_j(\widetilde{\boldsymbol{M}}) = \boldsymbol{v}_j^\top\widetilde{\boldsymbol{M}}\boldsymbol{v}_j$ (See (11)). We know since $\|\boldsymbol{A}(t)\|^2 \geq m/2$, $\lambda_j(\widetilde{\boldsymbol{M}}(t)) \geq \frac{1}{n}\lambda_j, \forall t, j$

$$=\frac{4\eta}{n^2 m}\left\|\sum_{i=1}^{r}\sum_{j'=1}^{r}\sum_{k=0}^{t}\left(\prod_{j=0}^{k-1}(1 - \eta\lambda_i(\widetilde{\boldsymbol{M}}(j)))z_i z_{j'}(1 - \eta\lambda_{j'}(\widetilde{\boldsymbol{M}}(j)))\lambda_{j'}\right)\boldsymbol{v}_i\boldsymbol{v}_{j'}^\top\right\|_2$$

$$\leq\frac{4\eta}{n^2 m}\left\|\sum_{i=1}^{r}\sum_{j'=1}^{r}\sum_{k=0}^{t}\left(\prod_{j=0}^{k-1}(1 - \eta\lambda_i(\widetilde{\boldsymbol{M}}(j)))z_i z_{j'}(1 - \eta\lambda_{j'}(\widetilde{\boldsymbol{M}}(j)))\lambda_{j'}\right)\boldsymbol{v}_i\boldsymbol{v}_{j'}^\top\right\|_F$$

$$=\frac{4\eta}{n^2 m}\sqrt{\sum_{i=1}^{r}\sum_{j'=1}^{r}\left(\sum_{k=0}^{t}\prod_{j=0}^{k-1}(1 - \eta\lambda_i(\widetilde{\boldsymbol{M}}(j)))z_i z_{j'}(1 - \eta\lambda_{j'}(\widetilde{\boldsymbol{M}}(j)))\lambda_{j'}\right)^2}$$

$$\leq\frac{4\eta}{n^2 m}\sqrt{\sum_{i=1}^{r}\sum_{j'=1}^{r}\left(\sum_{k=0}^{t}(1 - \eta\frac{1}{n}\lambda_{j'})^k z_i z_{j'}(1 - \eta\frac{1}{n}\lambda_i)^k\lambda_{j'}\right)^2}$$

$$\leq\frac{4}{n^2 m}\sqrt{\sum_{i=1}^{r}\sum_{j'=1}^{r}z_i^2 z_{j'}^2\lambda_{j'}^2\frac{\eta^2}{(\frac{1}{n}\eta(\lambda_i + \lambda_{j'}) - \frac{\eta^2}{n^2}\lambda_i\lambda_{j'})^2}}$$

$$\leq\frac{4}{n^2 m}\sqrt{\|\boldsymbol{Y}\|_2^4 \cdot n^2}$$

$$=\frac{4}{m}$$

The first inequality comes from $\|\cdot\|_2 \leq \|\cdot\|_F$. The last three inequalities require that $\eta\lambda_{\max}(\boldsymbol{M}^*) < 1$.

Combined with its symmetric counterpart in $\mathbf{X}^\top\mathbf{X}\boldsymbol{D}(t)\boldsymbol{D}(t)^\top$, the sum of (A1) is smaller than $\frac{8}{m}$.

Term (A2) and (A3):

$$\frac{4\eta}{n^2 m}\left\|\sum_{k=0}^{t}\boldsymbol{e}(k)\boldsymbol{Y}^\top\prod_{j=0}^{k-1}(\boldsymbol{I}_n - \eta\widetilde{\boldsymbol{M}}(j))\mathbf{X}^\top\mathbf{X} + \prod_{j=0}^{k-1}(\boldsymbol{I}_n - \eta\widetilde{\boldsymbol{M}}(j))\boldsymbol{Y}\boldsymbol{e}(k)^\top\mathbf{X}^\top\mathbf{X}\right\|_2$$

$$\leq \frac{8\eta}{n^2 m} \left\| \sum_{k=0}^{t} \boldsymbol{e}(k) \boldsymbol{Y}^\top \prod_{j=0}^{k-1} (\boldsymbol{I}_n - \eta \widetilde{\boldsymbol{M}}(j)) \mathbf{X}^\top \mathbf{X} \right\|_2 \qquad \text{(Triangle Inequality)}$$

$$\leq \frac{8\eta}{n^2 m} \sum_{k=0}^{t} \|\boldsymbol{e}(k)\|_2 \left\| \boldsymbol{Y}^\top \prod_{j=0}^{k-1} (\boldsymbol{I}_n - \eta \widetilde{\boldsymbol{M}}(j)) \right\|_2 \|\mathbf{X}^\top \mathbf{X}\|_2 \qquad \text{(Cauchy-Schwarz)}$$

$$\leq \frac{8}{n^2 m} \left( \sum_{k=0}^{t} \eta (1 - \frac{\eta}{n} \lambda_r)^k \sqrt{n} \right) \cdot \frac{n^{3/2}}{\lambda_r} R_w \cdot c_1 n \qquad \text{((E1) and Algebra)}$$

$$\leq \frac{8 c_1 n^2}{m \lambda_r^2} R_w$$

Similarly, we have their symmetric counterpart added and get a bound of $\frac{16 c_1 n^2}{m \lambda_r^2} R_w$.

Term (A4):

$$\frac{4\eta}{n^2 m} \left\| \sum_{k=0}^{t} \boldsymbol{e}(k) \boldsymbol{e}(k)^\top \mathbf{X}^\top \mathbf{X} \right\|_2$$

$$\leq \frac{4}{n^2 m} \eta \sum_{k=0}^{t} \|\boldsymbol{e}(k)\|^2 \|\mathbf{X}^\top \mathbf{X}\|_2 \qquad \text{(Cauchy-Schwarz)}$$

$$(\|\boldsymbol{e}(k)\|_2 \leq \|\boldsymbol{D}(k)\| + \| \prod_{j=0}^{k-1} (I - \eta \widetilde{\boldsymbol{M}}(j)) \boldsymbol{Y} \| \leq 2\sqrt{n}(1 - \frac{\eta}{n} \lambda_r)^k )$$

$$\leq \frac{4\eta}{N^2 m} \sum_{k=0}^{t} 2(1 - \frac{\eta}{n} \lambda_r)^k \sqrt{n} \cdot \frac{n^{3/2}}{\lambda_r} R_w \cdot c_1 n \qquad \text{((E1) and Algebra)}$$

$$\leq \frac{8 n^2 c_1}{m \lambda_r^2} R_w$$

Similarly, we can combine its symmetric counterpart and get a bound of $\frac{16 c_1 n^2}{m \lambda_r^2} R_w$.

Add them up, and we can get the bound of the first part.

$$\|(1)\|_2 \leq \frac{8}{m} + \frac{32 c_1 n^2}{m \lambda_r^2} R_w$$

Term (G2):

$$\frac{4}{n^2 m} \eta \left\| \sum_{k=0}^{t} \left( \boldsymbol{Y} \boldsymbol{D}(t)^\top \mathbf{X}^\top \mathbf{X} + \mathbf{X}^\top \mathbf{X} \boldsymbol{D}(t) \boldsymbol{Y}^\top \right) \right\|_2 \qquad \text{(Triangle Inequality)}$$

$$\leq \frac{8}{N^2 m} \eta \left\| \sum_{k=0}^{t} \boldsymbol{Y} \boldsymbol{Y}^\top \prod_{j=0}^{k-1} (\boldsymbol{I}_n - \eta \widetilde{\boldsymbol{M}}(j)) \mathbf{X}^\top \mathbf{X} \right\|_2 \qquad \text{(A5)}$$

$$+ \frac{8}{N^2 m} \left\| \sum_{k=0}^{t} \boldsymbol{Y} \boldsymbol{e}(k)^\top \mathbf{X}^\top \mathbf{X} \right\|_2 \qquad \text{(A6)}$$

We bound the two parts separately.

Term (A5):

$$\frac{8}{N^2 m} \eta \left\| \sum_{k=0}^{t} \boldsymbol{Y} \boldsymbol{Y}^\top \prod_{j=0}^{k-1} (\boldsymbol{I}_n - \eta \widetilde{\boldsymbol{M}}(j)) \mathbf{X}^\top \mathbf{X} \right\|_2 \qquad \text{(Cauchy-Schwarz)}$$

$$\leq \frac{8\eta}{n^2 m} \|\boldsymbol{Y}\boldsymbol{Y}^\top\|_2 \left\| \sum_{k=0}^{t} \sum_{i=1}^{r} (1 - \eta\frac{1}{n}\lambda_i)^k \lambda_i \boldsymbol{v}_i \boldsymbol{v}_i^\top \right\|_2 \leq \frac{8}{m} \qquad \text{(Algebra)}$$

Term (A6):

$$\frac{8}{n^2 m} \left\| \sum_{k=0}^{t} \boldsymbol{Y}\boldsymbol{e}(k)^\top \mathbf{X}^\top \mathbf{X} \right\|_2 \leq \frac{8}{n^2 m} \|\boldsymbol{Y}\|_2 \left\| \sum_{k=0}^{t} \boldsymbol{e}(k) \right\|_2 \cdot c_1 n \qquad \text{(Cauchy-Schwarz)}$$

$$\leq \frac{8}{n^2 m} \sqrt{n}\eta \sum_{k=0}^{t} \eta\sqrt{n}k(1 - \eta\frac{1}{n}\lambda_r)^{k-1} R_w c_1 n$$

$$\text{((E1) and } \|\boldsymbol{Y}\| = \sqrt{n})$$

$$\leq \frac{8c_1}{m}\eta^2 \sum_{k=0}^{t} \sum_{j=k}^{t-1} (1 - \eta\frac{1}{n}\lambda_r)^j R_w \qquad \text{(Abel's lemma)}$$

$$\leq \frac{8c_1}{m}\eta^2 \sum_{k=0}^{t} \frac{n}{\eta\lambda_r}(1 - \eta\frac{1}{n}\lambda_r)^k R_w \qquad \text{(Algebra)}$$

$$\leq \frac{8c_1}{m}\frac{n^2}{\lambda_r^2} R_w = \frac{8c_1 n^2}{m\lambda_r^2} R_w$$

Sum up and we get

$$\|(2)\|_2 \leq \frac{8}{m} + \frac{8c_1 n^2}{m\lambda_r^2} R_w$$

Term (G3):

$$\frac{8\eta^2}{n^3 m^2} \left\| \sum_{k=0}^{t} \|\boldsymbol{A}(t)\|^2 \mathbf{X}^\top \mathbf{X} \boldsymbol{D}(t)\boldsymbol{D}(t)^\top \mathbf{X}^\top \mathbf{X} \right\|_2$$

$$(\text{Since } \mathbf{X}^\top \mathbf{X} \boldsymbol{D}(t)\boldsymbol{D}(t)^\top \mathbf{X}^\top \mathbf{X} \text{ is PSD})$$

$$\leq \frac{4\eta}{n^2 m}\eta\frac{2}{mn}\max_t \left\| \|\boldsymbol{A}(t)\|^2 \mathbf{X}^\top \mathbf{X} \right\|_2 \cdot \left\| \sum_{k=0}^{t} \boldsymbol{D}(t)\boldsymbol{D}(t)^\top \mathbf{X}^\top \mathbf{X} \right\|_2$$

$$(\text{Since } \lambda_{\max}(\boldsymbol{M}^*) < 1/\eta)$$

$$\leq \frac{4\eta}{n^2 m} \left\| \sum_{k=0}^{t} \boldsymbol{D}(t)\boldsymbol{D}(t)^\top \mathbf{X}^\top \mathbf{X} \right\|_2 \leq \frac{4}{m} + \frac{16c_1 n^2}{m\lambda_r^2} R_w$$

The last inequality is the same one as in the term (G1) bound.

Adding the three terms together and using the induction hypothesis, we have

$$\|(G1)\|_2 + \|(G2)\|_2 + \|(G3)\|_2 \leq \frac{40}{m}.$$

Property 1 holds. Therefore the proof is completed. $\qquad \square$

Lemma C.3 tells us that $\boldsymbol{D}(t)$ decreases along GD trajectory in some fixed directions independently depending on $\mathbf{X}^\top \mathbf{X}$. After we have this GD trajectory, we can have the following Lemma C.4 about the dynamics of $\|\boldsymbol{A}(t)\|^2$ under the condition in Lemma C.3. It shows a sufficient condition for $\|\boldsymbol{A}(t)\|^2$ to grow.

**Lemma C.4.** *Under the Assumption C.1 and Assumption C.2, if*

$$n > (70\lambda_r + 25\lambda_r^2)/(196\kappa^2 \min\{a_1 - a_1^2, a_2 - a_2^2\})$$

*and as long as there exist two number $a_1, a_2$ (to be determined) and some $i \in [r]$ at time $t$ s.t.*

$$0 < a_1 \leq \prod_{j=0}^{t-1} (1 - \eta\lambda_i(\widetilde{\boldsymbol{M}}(j))) \leq a_2 < 1 \qquad (14)$$

*we have* $-\boldsymbol{D}(t)^\top \boldsymbol{F}(t) > 0$.

*Proof.* We use property 2 of Lemma C.3 to obtain the following expression:

$$-\boldsymbol{D}(t)^\top \boldsymbol{F}(t) = \boldsymbol{Y}^\top \prod_{j=0}^{t}(\boldsymbol{I}_n - \eta\widetilde{\boldsymbol{M}}(j))(\boldsymbol{I}_n - \prod_{j=0}^{t}(\boldsymbol{I}_n - \eta\widetilde{\boldsymbol{M}}(j)))\boldsymbol{Y}$$

$$+ \boldsymbol{Y}^\top (2\prod_{j=0}^{t}(\boldsymbol{I}_n - \eta\widetilde{\boldsymbol{M}}(j)) - \boldsymbol{I}_n)\boldsymbol{e}(t) - \|\boldsymbol{e}(t)\|^2 \quad \text{(Use (E1) and } m \geq \tfrac{112c_1n^2}{\lambda_r^2})$$

$$\geq \sum_{i=1}^{r} z_i^2 \prod_{j=0}^{t-1}(1 - \eta\lambda_i(\widetilde{\boldsymbol{M}}(j))(1 - \prod_{j=0}^{t-1}(1 - \eta\lambda_i(\widetilde{\boldsymbol{M}}(j)))) - \frac{40n^2}{\lambda_r m} - \frac{1600n^3}{\lambda_r^2 m^2}$$

$$\geq \kappa^2 n \max_i \left\{ \prod_{j=0}^{t-1}(1 - \eta\lambda_i(\widetilde{\boldsymbol{M}}(j))(1 - \prod_{j=0}^{t-1}(1 - \eta\lambda_i(\widetilde{\boldsymbol{M}}(j)))) \right\} - \frac{5\lambda_r}{14} - \frac{25\lambda_r^2}{196}$$

$$\text{(C.4)}$$

Notice that all inequalities hold since $\lambda_i(\boldsymbol{M}^*(t)) < 1/\eta$ for all $i$. In the first inequality, we use the $\|\boldsymbol{e}(t)\|$ bound in (E1), and in the second we just replace $m$ with its lower bound $\tfrac{112c_1n^2}{\lambda_r^2}$. Then we use $n > (70\lambda_r + 25\lambda_r^2)/(196\kappa^2 \min\{a_1 - a_1^2, a_2 - a_2^2\})$ and complete the proof.

$$-\boldsymbol{D}(t)^\top \boldsymbol{F}(t) \geq \kappa^2 n \min\{a_1 - a_1^2, a_2 - a_2^2\} - \frac{5\lambda_r}{14} - \frac{25\lambda_r^2}{196} > 0$$

$\square$

Then, we use Assumption C.1 to prove the next lemma. It tells that condition (14) is satisfied under this assumption in a time interval. That means $-\boldsymbol{D}(t)^\top \boldsymbol{F}(t) > 0$ during this period.

**Lemma C.5.** *Under Assumption C.1, we have condition* (14) *satisfied with* $a_2 = e^{(a_1-1)/\chi}$ *in the time interval* $[t_1, t_2)$. *Here,* $t_1$ *is the iteration that* $\prod_{j=0}^{t-1}(1 - \eta\lambda_1(\widetilde{\boldsymbol{M}}(j))) < a_2$ *for the first time, and* $t_2$ *is the iteration when* $\prod_{j=0}^{t-1}(1 - \eta\lambda_r(\widetilde{\boldsymbol{M}}(j))) < a_1$ *for the first time.*

*Proof.* We prove for all $i \in [r-1]$,

$$\text{If } \prod_{j=0}^{t-1}(1 - \eta\lambda_i(\widetilde{\boldsymbol{M}}(j))) < a_1, \text{then } \prod_{j=0}^{t-1}(1 - \eta\lambda_{i+1}(\widetilde{\boldsymbol{M}}(j))) < e^{(a_1-1)/\chi}. \quad (15)$$

In this way, the only two possibility that all $i \in [r]$ doesn't satisfy the condition (14) is: (1) the $\prod_{j=0}^{t-1}(1 - \eta\lambda_1(\widetilde{\boldsymbol{M}}(j))) > a_2$; (2) $\prod_{j=0}^{t-1}(1 - \eta\lambda_r(\widetilde{\boldsymbol{M}}(j))) < a_1$. Otherwise, there must be some $i$ s.t. $\prod_{j=0}^{t-1}(1 - \eta\lambda_i(\widetilde{\boldsymbol{M}}(j))) \in [a_1, a_2]$. Thus, if condition (15) is satisfied, we have this lemma proved.

Now suppose $\prod_{j=0}^{t-1}(1 - \eta\lambda_i(\widetilde{\boldsymbol{M}}(j))) < a_1$. By Bernoulli's inequality,

$$(1 + x_1)(1 + x_2)...(1 + x_n) \geq 1 + x_1 + x_2 + ... + x_n, \text{if } x_i \geq -1, \forall i \in [n]$$

since $-\eta\lambda_i(\widetilde{\boldsymbol{M}}(j)) > -1$ for all $i$, we have

$$\sum_{j=0}^{t-1} \eta\lambda_i(\widetilde{\boldsymbol{M}}(j)) > 1 - a_1$$

By Jensen inequality, we have

$$\sum_{j=0}^{t-1} \log(1 - \eta\lambda_i(\widetilde{\boldsymbol{M}}(j))/\chi) \leq t \log\left(1 - \frac{\sum_{j=0}^{t-1} \eta\lambda_i(\widetilde{\boldsymbol{M}}(j))}{t\chi}\right)$$

Hence we have the following inequalities:

$$\prod_{j=0}^{t-1}(1 - \eta\lambda_{i+1}(\widetilde{\boldsymbol{M}}(j))) \leq \prod_{j=0}^{t-1}(1 - \eta\lambda_i(\widetilde{\boldsymbol{M}}(j))/\chi)$$

$$\leq \exp\left\{t\log\left(1 - \frac{\sum_{j=0}^{t-1}\eta\lambda_i(\widetilde{\boldsymbol{M}}(j))}{t\chi}\right)\right\}$$

$$\leq (1 + \frac{a_1 - 1}{t\chi})^t$$

$$\leq e^{(a_1-1)/\chi}$$

So we complete the proof. □

Then we pay attention back to sharpness. We have the sharpness's dynamics by Lemma C.2.

$$\Lambda^*(t+1) - \Lambda^*(t) = \boldsymbol{v}_1^\top(\boldsymbol{M}(t+1) - \boldsymbol{M}(t))\boldsymbol{v}_1 \qquad \text{(Definition (13))}$$

$$= -\frac{8\eta\lambda_1}{n^2 m}(\boldsymbol{D}(t)^\top\boldsymbol{F}(t) + (\boldsymbol{D}(t)^\top\boldsymbol{v}_1)(\boldsymbol{F}(t)^\top\boldsymbol{v}_1))$$

$$+ \frac{8\eta^2\lambda_1}{m^2 n^3}(\boldsymbol{D}(t)^\top\mathbf{X}^\top\boldsymbol{W}^\top(t)\boldsymbol{W}(t)\mathbf{X}\boldsymbol{D}(t) + \|\boldsymbol{A}(t)\|^2\lambda_1(\boldsymbol{D}(t)^\top\boldsymbol{v}_1)^2)$$

This equation shows that the dynamics of sharpness is closely related to the dynamics of $\|\boldsymbol{A}(t)\|^2$, i.e. highly dependent on $-\boldsymbol{D}(t)^\top\boldsymbol{F}(t)$. Based on the lemmas above, we can prove progressive sharpening happens almost along the whole training trajectory (Lemma C.6) until the loss $\mathcal{L} = \frac{1}{n}\|\boldsymbol{D}(t)\|^2$ converges to $O(n^{-1})$.

**Lemma C.6.** *Under Assumption C.1 and C.2, if $m > \frac{112c_1 n^2}{\lambda_r^2}$ and*

$$n > \max\{(70\lambda_r + 25\lambda_r^2)/(98\kappa^2\min\{e^{-1/\chi} - e^{-2/\chi}, \eta c_1(1 - \eta c_1)\}), (\frac{70\lambda_r + 25\lambda_r^2}{98\kappa^2\epsilon})^2\}$$

*we have $\Lambda^*(t+1) - \Lambda^*(t) > 0$ until the time t when $\|\boldsymbol{D}(t)\|^2 \leq \epsilon^2 + \frac{5\epsilon\lambda_r}{7\sqrt{n}} + \frac{25\lambda_r^2}{196n}$.*

**Remark.** When $a_1 = \epsilon/\sqrt{n}, a_2 = e^{(a_1-1)/\chi} < e^{-1/\chi}$, the lower bound of $n$ guarantee that

$$\kappa^2 n\min\{a_1 - a_1^2, a_2 - a_2^2, (1 - \eta c_1)\eta c_1\} > \frac{5\lambda_r}{7} - \frac{25\lambda_r^2}{98} \qquad (16)$$

*Proof.* Note that the second order term

$$\frac{8\eta^2\lambda_1}{m^2 n^3}(\boldsymbol{D}(t)^\top\mathbf{X}^\top\boldsymbol{W}(t)^\top\boldsymbol{W}(t)\mathbf{X}\boldsymbol{D}(t) + \|\boldsymbol{A}(t)\|^2\lambda_1(\boldsymbol{D}(t)^\top\boldsymbol{v}_1)^2)$$

is larger than 0. So as long as the first order term

$$-\frac{8\eta\lambda_1}{n^2 m}(\boldsymbol{D}(t)^\top\boldsymbol{F}(t) + (\boldsymbol{D}(t)^\top\boldsymbol{v}_1)(\boldsymbol{F}(t)^\top\boldsymbol{v}_1)) > 0$$

the approximate sharpness will grow.

First, we give a lower bound for the number $-(\boldsymbol{D}(t)^\top\boldsymbol{v}_1)(\boldsymbol{F}(t)^\top\boldsymbol{v}_1)$:

$$-(\boldsymbol{D}(t)^\top\boldsymbol{v}_1)(\boldsymbol{F}(t)^\top\boldsymbol{v}_1) = z_1^2\prod_{j=0}^{t-1}(1 - \eta\lambda_1(\widetilde{\boldsymbol{M}}(j))(1 - \prod_{j=0}^{t-1}(1 - \eta\lambda_1(\widetilde{\boldsymbol{M}}(j)))$$

$$+ z_1(2\prod_{j=0}^{t-1}(1 - \eta\lambda_1(\widetilde{\boldsymbol{M}}(j))) - 1)(\boldsymbol{e}(t)^\top\boldsymbol{v}_1) - (\boldsymbol{e}(t)^\top\boldsymbol{v}_1)^2$$

$$\geq -|z_1|\|\boldsymbol{e}(t)\| - \|\boldsymbol{e}(t)\|^2 \qquad (\lambda_1(\boldsymbol{M}^*(t)) < 1/\eta)$$

$$\geq -\frac{40n^2}{\lambda_r m} - \frac{1600n^3}{\lambda_r^2 m^2} \qquad\qquad \text{(Use (E1))}$$

$$\geq -\frac{5\lambda_r}{14} - \frac{25\lambda_r^2}{196} \qquad\qquad (m \geq \tfrac{112c_1 n^2}{\lambda_r^2})$$

where the first equation holds due to Property 2 of Lemma C.3.

With the lower bound of $-(\boldsymbol{D}(t)^\top \boldsymbol{v}_1)(\boldsymbol{F}(t)^\top \boldsymbol{v}_1)$, we show the dynamics of the first order term by similar technique in the expression (C.4) :

$$-\boldsymbol{D}(t)^\top \boldsymbol{F}(t) - (\boldsymbol{D}(t)^\top \boldsymbol{v}_1)(\boldsymbol{F}(t)^\top \boldsymbol{v}_1) \qquad\qquad \text{(Property 2 of Lemma C.3)}$$

$$= \boldsymbol{Y}^\top \prod_{j=0}^{t}(\boldsymbol{I}_n - \eta\widetilde{\boldsymbol{M}}(j))(\boldsymbol{I}_n - \prod_{j=0}^{t}(\boldsymbol{I}_n - \eta\widetilde{\boldsymbol{M}}(j)))\boldsymbol{Y} + \boldsymbol{Y}^\top(2\prod_{j=0}^{t}(\eta\widetilde{\boldsymbol{M}}(j) - \boldsymbol{I}_n) - \boldsymbol{I}_n)e(t)$$

$$- \|e(t)\|^2 - (\boldsymbol{D}(t)^\top \boldsymbol{v}_1)(\boldsymbol{F}(t)^\top \boldsymbol{v}_1) \qquad\qquad (\text{Use (E1) and } m \geq \tfrac{112c_1 n^2}{\lambda_r^2})$$

$$\geq \kappa^2 n \max_i \left\{ \prod_{j=0}^{t-1}(1 - \eta\lambda_i(\widetilde{\boldsymbol{M}}(j))(1 - \prod_{j=0}^{t-1}(1 - \eta\lambda_i(\widetilde{\boldsymbol{M}}(j))) \right\} - \frac{5\lambda_r}{14} - \frac{25\lambda_r^2}{196}$$

$$+ (-\frac{5\lambda_r}{14} - \frac{25\lambda_r^2}{196}) \qquad\qquad \text{(Property 2 of Lemma C.3)}$$

$$\geq \kappa^2 n \max_i \left\{ \prod_{j=0}^{t-1}(1 - \eta\lambda_i(\widetilde{\boldsymbol{M}}(j))(1 - \prod_{j=0}^{t-1}(1 - \eta\lambda_i(\widetilde{\boldsymbol{M}}(j))) \right\} - \frac{5\lambda_r}{7} - \frac{25\lambda_r^2}{98}$$

$$\geq \kappa^2 n(1 - \eta c_1)\eta c_1 - \frac{5\lambda_r}{7} - \frac{25\lambda_r^2}{98} > 0$$

The last inequality holds due to the lower bound of $n$ (16) assumed in Lemma C.6.

Now, $\prod_{j=0}^{t-1}(1 - \eta\lambda_1(\widetilde{\boldsymbol{M}}(j)))$ begins to decrease each iteration. Before the time when $\prod_{j=0}^{t-1}(1 - \eta\lambda_1(\widetilde{\boldsymbol{M}}(j)))$ becomes smaller than $\frac{\epsilon}{\sqrt{n}}$, $-\boldsymbol{D}(t)^\top \boldsymbol{F}(t) > 0$ always holds because of the lower bound of $n$ (16) assumed in Lemma C.6.

Then, after the time $t_1$ when $\prod_{j=0}^{t-1}(1 - \eta\lambda_1(\widetilde{\boldsymbol{M}}(j)) < \frac{\epsilon}{\sqrt{n}}$ and before the time $t_2$ when $\prod_{j=0}^{t-1}(1 - \eta\lambda_i(\widetilde{\boldsymbol{M}}(j)) < \frac{\epsilon}{\sqrt{n}}$ for all $i$, we enter the time interval $[t_1, t_2)$ where Lemma C.5 begins to hold. We use Lemma C.5 to show that there exists some $i$ to make

$$\prod_{j=0}^{t-1}(1 - \eta\lambda_i(\widetilde{\boldsymbol{M}}(j)) \in [\frac{\epsilon}{\sqrt{n}}, e^{-1/\chi}]$$

Before the time $t_2$, we have this inequality always hold. Thus

$$-\boldsymbol{D}(t)^\top \boldsymbol{F}(t) - (\boldsymbol{D}(t)^\top \boldsymbol{v}_1)(\boldsymbol{F}(t)^\top \boldsymbol{v}_1) > 0$$

holds until the iteration $t_2$. Thus during this period, $\Lambda^*(t)$ keeps increasing.

At this iteration, $\left\| \prod_{j=0}^{t-1}(\boldsymbol{I}_n - \eta\widetilde{\boldsymbol{M}}(j)) \right\| \leq \frac{\epsilon}{\sqrt{n}}$, and we can bound the norm of the residual $\boldsymbol{D}(t)$ with the inequality below. We have

$$\|\boldsymbol{D}(t)\|^2 = \| - \prod_{j=0}^{t-1}(\boldsymbol{I}_n - \eta\widetilde{\boldsymbol{M}}(j))\boldsymbol{Y} + e(t)\|^2 \qquad\qquad \text{(Triangle Inequality)}$$

$$\leq \left\| \prod_{j=0}^{t-1}(\boldsymbol{I}_n - \eta\widetilde{\boldsymbol{M}}(j))\boldsymbol{Y} \right\|^2 + 2\left\| \prod_{j=0}^{t-1}(\boldsymbol{I}_n - \eta\widetilde{\boldsymbol{M}}(j))\boldsymbol{Y} \right\| \|e(t)\| + \|e(t)\|^2$$

$$\qquad\qquad \text{(Cauchy-Schwarz)}$$

$$\leq \left\| \prod_{j=0}^{t-1}(\boldsymbol{I}_n - \eta\widetilde{\boldsymbol{M}}(j)) \right\|^2 \|\boldsymbol{Y}\|^2 + 2\left\| \prod_{j=0}^{t-1}(\boldsymbol{I}_n - \eta\widetilde{\boldsymbol{M}}(j)) \right\| \|\boldsymbol{Y}\| \|e(t)\| + \|e(t)\|^2$$

$$\qquad\qquad (\|\boldsymbol{Y}\| = \sqrt{n})$$

$$\leq n \max_i \left\{ \prod_{j=0}^{t-1} (1 - \eta \lambda_i(\widetilde{\boldsymbol{M}}(j))) \right\}^2 + \frac{80n^2}{\lambda_r m} \max_i \left\{ \prod_{j=0}^{t-1} (1 - \eta \lambda_i(\widetilde{\boldsymbol{M}}(j))) \right\} + \frac{1600n^3}{\lambda_r^2 m^2}$$

$$< \epsilon^2 + \frac{5\epsilon\lambda_r}{7\sqrt{n}} + \frac{25\lambda_r^2}{196n} \qquad\qquad (\left\| \prod_{j=0}^{t-1}(\boldsymbol{I}_n - \eta \widetilde{\boldsymbol{M}}(j)) \right\| \leq \frac{\epsilon}{\sqrt{n}})$$

Thus, before the norm of the residual $\boldsymbol{D}(t)$ decreases to this value, $\Lambda^*(t)$ keeps increasing. $\qquad\square$

### C.3 Edge of Stability (Phase II-IV)

In the edge of stability regime, we focus on the largest eigenvalue $\Lambda$ and its corresponding eigenvector $\boldsymbol{u}$. Since $\boldsymbol{M}(t)$ has a large similarity with $\mathbf{X}^\top \mathbf{X}$ in progressive sharpening phase, we consider the eigenvector $\boldsymbol{v}_1$ corresponding to the largest eigenvalue $\lambda_1$ of $\mathbf{X}^\top \mathbf{X}$.

After $\Lambda \geq 2/\eta$, the proof in Section C.2 does not extend to this phase. However, the bound of $\|\boldsymbol{\Gamma}(t)\|$ (Lemma C.3) still holds up to a constant factor empirically (See Figure 20). Hence, we make this bound an assumption as follows.

**Assumption C.3.** *There exists some constant $c_2 > 0$, such that for any time $t$,*

$$\|\boldsymbol{\Gamma}(t)\| \leq \frac{c_2}{m}$$

Note that in above progressive sharpening stage, the assumption holds by Theorem 3. We propose this assumption to keep the gram matrix from deviating too far from the original trajectory even in other phases. The verification of this assumption can be found in Appendix D.2.2.

**Corollary C.2.** *Recall that $\boldsymbol{v}_1$ is the largest eigenvector of $\mathbf{X}^\top \mathbf{X}$ and $\boldsymbol{u}$ is the largest eigenvector of $\boldsymbol{M}$. There exists a constant $c_4$, such that $\langle \boldsymbol{v}_1, \boldsymbol{u} \rangle^2 \geq 1 - c_4 \sqrt{\frac{1}{nm}}$, and $\Lambda(t) \geq \Lambda^*(t) \geq \Lambda(t) - \frac{2c_2}{m}$.*

*Proof.* By difinition of $\boldsymbol{\Gamma}(t)$ in 10, $\boldsymbol{M} = \gamma(t)\mathbf{X}^\top\mathbf{X} + \boldsymbol{\Gamma}(t)$, here $\gamma(t) = \frac{2}{mn}(\|\boldsymbol{A}(t)\|^2 + \frac{m}{d}) \geq \frac{2}{nd}$.

Let $\langle \boldsymbol{v}_1, \boldsymbol{u} \rangle = \cos\theta$ and decompose $\boldsymbol{u}$ by $\boldsymbol{u} = \boldsymbol{v}\cos\theta + \boldsymbol{v}_\perp \sin\theta$. Then we have $\Lambda = \boldsymbol{u}^\top \boldsymbol{M}\boldsymbol{u} \leq \lambda_1 \gamma(t)\cos^2\theta + \frac{\lambda_1}{2}\sin^2\theta\gamma(t) + \frac{c_2}{m}$.

Also we have $\Lambda \geq \boldsymbol{v}_1^\top \boldsymbol{M}\boldsymbol{v}_1 \geq \gamma(t)\lambda_1 - \frac{c_2}{m}$. So $\lambda_1\gamma(t)\cos^2\theta + \frac{\lambda_1}{2}\sin^2\theta\gamma(t) + \frac{c_2}{m} \geq \gamma(t)\lambda_1 - \frac{c_2}{m}$, which induces $\sin^2\theta \leq \frac{2c_2}{m\gamma(t)} \cdot \frac{2}{\lambda_1} \leq \frac{2c_2^2 d}{m\lambda_1}$. Because in our setting $\lambda_1 = \Theta(n)$, there exists a constant $c_4$ such that $\cos\theta \geq \sqrt{1 - \frac{c_4^2}{nm}} \geq 1 - c_4\sqrt{\frac{1}{mn}}$.

The inequality $\Lambda(t) \geq \Lambda^*(t)$ is because $\boldsymbol{u}^\top\boldsymbol{M}\boldsymbol{u} \geq \boldsymbol{v}_1^\top\boldsymbol{M}\boldsymbol{v}_1$ by definition of $\boldsymbol{u}$. The other side can be proved as the following:

$$\begin{aligned} \boldsymbol{v}_1^\top \boldsymbol{M}\boldsymbol{v}_1 =& \boldsymbol{v}_1^\top(\boldsymbol{M} - \boldsymbol{\Gamma})\boldsymbol{v}_1 + \boldsymbol{v}_1^\top\boldsymbol{\Gamma}\boldsymbol{v}_1 \\ \geq& \boldsymbol{u}^\top(\boldsymbol{M} - \boldsymbol{\Gamma})\boldsymbol{u} - c_2/m \\ \geq& \boldsymbol{u}^\top\boldsymbol{M}\boldsymbol{u} - 2c_2/m. \end{aligned}$$

$\square$

To prove the main theorem (Theorem 4), we need two more assumptions.

**Assumption C.4.** *There exists some constant $c_3$ such that $|\boldsymbol{D}(t)^\top \boldsymbol{v}_1| > c_3\sqrt{n}/m$ for some $t = t_0$ at the beginning of phase II.*

**Assumption C.5.** *There exists some constant $\beta > 0$, such that*

$$\frac{4}{\eta}(1 - \beta) \geq \Lambda$$

The above assumption is consistent with Assumption 3.4, in which we assume an upper bound of the sharpness. Lewkowycz et al. [19] showed that $4/\eta$ is a upper bound of the sharpness in two-layer linear network with one datapoint, otherwise the training process would diverge. Here we make it an assumption.

**Theorem 4.** *Suppose the smallest nonzero eigenvalue $\lambda_r = \lambda_r(\mathbf{X}^\top \mathbf{X}) > 0, \lambda_1 = \lambda_{\max}(\mathbf{X}^\top \mathbf{X})$. Under Assumption C.3, C.5, C.4, and $\lambda_1(\mathbf{X}^\top \mathbf{X}) \geq 2\lambda_2(\mathbf{X}^\top \mathbf{X})$ in Assumption C.1, there exists constants $C_1, C_2, C_3$, such that if $n > c_1 \lambda_r \eta, m > \max\{\frac{C_2 d^2 n^2}{\lambda_r^2}, C_3\eta\}$, we have*

- *(Lemma C.13) There exists $\rho = O(1)$ which is related to $c_3$ such that if $\Lambda(t_0) > \frac{2}{\eta}(1 + \rho)$ for some $t_0$, there must exist some $t_1 > t_0$ such that $\Lambda(t_1) < \frac{2}{\eta}(1 + \rho)$.*

- *(Lemma C.11) If $\Lambda(t), \Lambda(t + 1) > \frac{2}{\eta}(1 + \rho)$, then there is a constant $c_7 > 0$ (related to $c_3$) such that $|\mathbf{D}(t + 1)^\top \mathbf{v}_1| > |\mathbf{D}(t)^\top \mathbf{v}_1|(1 + c_7)$.*

- *(Lemma C.10) Define $\mathbf{R}(t) := (\mathbf{I} - \mathbf{v}_1 \mathbf{v}_1^\top)\mathbf{D}(t)$, and $\mathbf{R}'(t) := (\mathbf{I} - \eta \mathbf{M}^*(t)(\mathbf{I} - \mathbf{v}_1 \mathbf{v}_1^\top))\mathbf{R}'(t - 1)$. We have $\|\mathbf{R}(t) - \mathbf{R}'(t)\| = O(\frac{\sqrt{n^3}d}{\lambda_r \sqrt{m}})$.*

Next, we prove this theorem in three parts: first we prove the third statement, which gives $\mathbf{R}(t)$ an upper bound (Lemma C.10), then we prove the second statement, in which we use Assumption C.4 to prove that when sharpness is above $2/\eta$, $\mathbf{D}^\top \mathbf{v}_1$ increases geometrically (Lemma C.11), and lastly we prove that the sharpness eventually drops below $2/\eta$ (Lemma C.13), which is the first statement in our theorem.

Now we first give a key equation:

**Lemma C.7.** *The dynamics of the approximation on sharpness is:*

$$
\begin{aligned}
&\Lambda^*(t + 1) - \Lambda^*(t) \\
&= -\frac{8\eta \lambda_1}{mn^2} \left( \mathbf{F}(t)^\top \mathbf{D}(t) + (\mathbf{F}(t)^\top \mathbf{v}_1)(\mathbf{D}(t)^\top \mathbf{v}_1) - \frac{\eta}{2}(\mathbf{D}(t)^\top \mathbf{v}_1)^2 \Lambda^*(t) \right. \\
&\quad \left. -\frac{\eta}{2}\mathbf{R}(t)^\top \mathbf{\Gamma}(t)\mathbf{R}(t) - \eta \mathbf{R}(t)^\top \mathbf{\Gamma}(t) \left( \mathbf{v}_1 \mathbf{v}_1^\top \mathbf{D}(t) \right) - \frac{\eta}{mn} \mathbf{R}(t)^\top \left( \frac{m}{d}\mathbf{X}^\top \mathbf{X} \right) \mathbf{R}(t) \right).
\end{aligned} \tag{17}
$$

The proof of the equation is long and tedious, so we leave that in Lemma E.11.

Next we deal with the gap between $\mathbf{M}(t)$ and $\mathbf{M}^*(t)$. Note that $\mathbf{M}$ is the Gram Matrix, but in gradient descent trajectory, $\mathbf{D}(t + 1) - \mathbf{D}(t) = -\mathbf{M}^*(t)\mathbf{D}(t)$, so $\mathbf{M}^*$ is the one that truly controls $\mathbf{D}$'s dynamics. From the following lemma, we can see that $\mathbf{M}^*(t)$ is a smoothed version of $\mathbf{M}(t)$ and $\mathbf{M}(t + 1)$ plus a small perturbation.

**Lemma C.8.** *If $m \geq \eta c_2(d + 1)$ (recall that $c_2$ is defined in Assumption C.3), then there exists $k_s \in [0, 1)$ and a constant $c_6$ such that $\|\mathbf{M}^*(t) - (1 - k_s)\mathbf{M}(t) - k_s \mathbf{M}(t + 1)\| \leq \frac{c_6}{m}$.*

*Proof.* Recall that $\mathbf{M}^*(t) = \mathbf{M}(t) - \frac{4\eta}{N^2 m}(\mathbf{D}(t)^\top \mathbf{F}(t))\mathbf{X}^\top \mathbf{X}$. Then we consider two different cases.

**Case 1**: $|\mathbf{D}(t)^\top \mathbf{F}(t)| \leq \frac{d+1}{d-2}(1 + \frac{d+1}{d-2})n$.

In this case $\|\mathbf{M}^* - \mathbf{M}\|_2 \leq \frac{4\eta \lambda_1}{mn^2}\frac{d+1}{d-2}(1 + \frac{d+1}{d-2})n = O(\frac{\eta \lambda_1}{mn}) = O(\frac{1}{m})$. Here the last inequality follows from the inequality (5).

**Case 2**: $|\mathbf{D}(t)^\top \mathbf{F}(t)| > \frac{d+1}{d-2}(1 + \frac{d+1}{d-2})n$.

We claim that in this case we have $\mathbf{F}(t)^\top \mathbf{D}(t) > (\frac{2\eta}{mn}\frac{m}{d}\lambda_1 + \frac{c_2 \eta}{m})\|\mathbf{D}(t)\|^2$. If it does not hold, then $R.H.S. \geq \mathbf{F}(t)^\top \mathbf{D}(t) \geq \|\mathbf{D}(t)\|^2 - \|\mathbf{D}(t)\|\|\mathbf{Y}\|$. Hence we can have

$$
\begin{aligned}
\|\mathbf{Y}\| &\geq \|\mathbf{D}(t)\|(1 - \frac{2\eta}{mn}\frac{m}{d}\lambda_1 - \frac{c_2 \eta}{m}) \\
&= \|\mathbf{D}(t)\|(1 - \frac{2\eta \lambda_1}{nd} - \frac{\eta c_2}{m}) \\
&\geq \|\mathbf{D}(t)\|(1 - \frac{2}{d + 1} - \frac{1}{d + 1})
\end{aligned}
$$

The last inequality uses the restriction of $m$ in this lemma and the inequality (5). Hence we have $|\boldsymbol{F}(t)^\top \boldsymbol{D}(t)| \leq \|\boldsymbol{D}(t)\|(\|\boldsymbol{D}(t)\| + \|Y\|) \leq \frac{d+1}{d-2}(1 + \frac{d+1}{d-2})n$, which leads to a contradiction.

Hence now $\boldsymbol{F}(t)^\top \boldsymbol{D}(t) > (\frac{2\eta}{mn} \frac{m}{d} \lambda_1 + \frac{c_2\eta}{m}) \|\boldsymbol{D}(t)\|^2$ holds. Since $\mathbf{X}^\top \boldsymbol{W}^\top \boldsymbol{W} \mathbf{X} = \frac{m}{d} \mathbf{X}^\top \mathbf{X} + \boldsymbol{\Gamma}(t)$ and $\|\mathbf{X}^\top \mathbf{X}\| = \lambda_1, \|\boldsymbol{\Gamma}(t)\| \leq \frac{c_2}{m}$, we can have:

$$\boldsymbol{F}(t)^\top \boldsymbol{D}(t) > \frac{2\eta}{mn} \boldsymbol{D}(t)^\top \mathbf{X}^\top \boldsymbol{W}^\top \boldsymbol{W} \mathbf{X} \boldsymbol{D}(t).$$

Now let $k_s := \frac{\boldsymbol{F}(t)^\top \boldsymbol{D}(t)}{2\boldsymbol{F}(t)^\top \boldsymbol{D}(t) - \frac{2\eta}{mn} \boldsymbol{D}(t)^\top \mathbf{X}^\top \boldsymbol{W}^\top \boldsymbol{W} \mathbf{X} \boldsymbol{D}(t)}$, by the inequality above we have $k_s < 1$.

By the equation

$$\boldsymbol{M}(t+1) - \boldsymbol{M}(t) = \frac{2}{mn} \left( \|\boldsymbol{A}(t+1)\|^2 - \|\boldsymbol{A}(t)\|^2 \right) \mathbf{X}^\top \mathbf{X} + (\boldsymbol{\Gamma}(t+1) - \boldsymbol{\Gamma}(t))$$

$$\|\boldsymbol{A}(t+1)\|^2 - \|\boldsymbol{A}(t)\|^2 = -\frac{4\eta}{n} \boldsymbol{F}(t)^\top \boldsymbol{D}(t) + \frac{4\eta^2}{n^2} \cdot \frac{1}{m} \boldsymbol{D}(t)^\top \mathbf{X}^\top \boldsymbol{W}^\top \boldsymbol{W} \mathbf{X} \boldsymbol{D}(t)$$

we have $\boldsymbol{M}^* = \boldsymbol{M}(t) + k_s(\boldsymbol{M}(t+1) - \boldsymbol{M}(t) - (\boldsymbol{\Gamma}(t+1) - \boldsymbol{\Gamma}(t))$. Hence $\|\boldsymbol{M}^* - (1 - k_s)\boldsymbol{M}(t) - k_s \boldsymbol{M}(t+1)\| \leq \frac{2c_2}{m}$. Now combining the conclusions in both cases together, we finish the proof. $\square$

Then we consider a corollary of this lemma. Basically, since $\boldsymbol{M}^*(t)$ is a weighted sum of $\boldsymbol{M}(t)$ and $\boldsymbol{M}(t+1)$ adding a small perturbation and $\boldsymbol{M}(t)$ has a decomposition to the $\mathbf{X}^\top \mathbf{X}$ component and a small noise $\boldsymbol{\Gamma}(t)$, $\boldsymbol{M}^*(t)$ can also be decomposed into the $\mathbf{X}^\top \mathbf{X}$ component and a small noise.

**Corollary C.3.** *$\boldsymbol{M}^*$ can be decomposed to $\boldsymbol{M}^*(t) = \gamma^*(t)\mathbf{X}^\top \mathbf{X} + \boldsymbol{\Gamma}^*(t)$, where $\|\boldsymbol{\Gamma}^*(t)\| < \frac{c_2+c_6}{m}$ and for any eigenvector $\boldsymbol{u}$ of $\mathbf{X}^\top \mathbf{X}$ except $\boldsymbol{v}_1$, and if let $\tau = \frac{2\lambda_r}{nd}$,*

$$\tau \leq \boldsymbol{u}^\top (\boldsymbol{M}^*(t) - \boldsymbol{\Gamma}^*(t))\boldsymbol{u} \leq \frac{2}{\eta} - \tau$$

*Proof.* By Lemma C.8, if we denote $\boldsymbol{\Gamma}'(t) = \boldsymbol{M}^*(t) - k_s \boldsymbol{M}(t+1) - (1-k_s)\boldsymbol{M}(t)$, then $\|\boldsymbol{\Gamma}'(t)\| \leq \frac{c_6}{m}$.

Hence, we can see

$$\boldsymbol{M}^*(t) = k_s \boldsymbol{M}(t+1) + (1-k_s)\boldsymbol{M}(t) + \boldsymbol{\Gamma}'$$

$$= \frac{2}{mn}(k_s \|\boldsymbol{A}(t+1)\|^2 + (1-k_s)\|\boldsymbol{A}(t)\|^2 + \frac{2m}{d})\mathbf{X}^\top \mathbf{X}$$

$$+ k_s \boldsymbol{\Gamma}(t+1) + (1-k_s)\boldsymbol{\Gamma}(t) + \boldsymbol{\Gamma}'(t)$$

Denote $k_s \boldsymbol{\Gamma}(t+1) + (1-k_s)\boldsymbol{\Gamma}(t) + \boldsymbol{\Gamma}'(t)$ by $\boldsymbol{\Gamma}^*(t)$. By Assumption C.3 and Lemma C.8, $\|\boldsymbol{\Gamma}^*(t)\| < \frac{c_2+c_6}{m}$.

Note that by Assumption C.1, $\lambda_i(\mathbf{X}^\top \mathbf{X}) \leq \frac{1}{2}\lambda_1$ for any $i > 1$. Then, we can see that

$$\boldsymbol{v}_1^\top (\boldsymbol{M}(t) - \boldsymbol{\Gamma}(t))\boldsymbol{v}_1 = \frac{2\lambda_1}{mn}(\|\boldsymbol{A}(t)\|^2 + \frac{m}{d}) \geq \frac{4\lambda_i}{mn}(\|\boldsymbol{A}(t)\|^2 + \frac{m}{d})$$

Hence, we can see

$$\frac{2\lambda_i}{mn}(\|\boldsymbol{A}(t)\|^2 + \frac{m}{d}) \leq \frac{1}{2}\boldsymbol{v}_1^\top (\boldsymbol{M}(t) - \boldsymbol{\Gamma}(t))\boldsymbol{v}_1$$

$$\leq \frac{1}{2}(\Lambda + \frac{c_2}{m})$$

$$\leq \frac{2}{\eta}(1 - \beta) + \frac{c_2}{2m}.$$

Here the last inequality holds due to Assumption C.5.

Now the inequality above shows that for any eigenvector $\boldsymbol{u}$ of $\mathbf{X}^\top \mathbf{X}$ except $\boldsymbol{v}_1$,

$$\boldsymbol{u}^\top (\boldsymbol{M}^*(t) - \boldsymbol{\Gamma}^*(t))\boldsymbol{u} \leq \frac{2}{\eta} - (\frac{2\beta}{\eta} - \frac{c_2}{2m}).$$

On the other side, $\frac{2\lambda_i}{mn}(\|\boldsymbol{A}(t)\|^2 + \frac{m}{d}) \geq \frac{2\lambda_r}{nd}$.

Take $\tau = \min\{\frac{2\lambda_r}{nd}, \frac{2\beta}{\eta} - \frac{c_2}{2m}\}$. Now because $m = \Omega(\eta)$ and $\beta$ is some constant, we can see $\tau = \min\{\frac{2\lambda_r}{nd}, \frac{2\beta}{\eta} - \frac{c_2}{2m}\} = \frac{2\lambda_r}{nd}$. $\qquad\square$

Before using this corollary to derive the dynamics of $\boldsymbol{R}(t)$ (and thus gives an upper bound of $\|\boldsymbol{R}(t)\|$), we need an upper bound for $\|\boldsymbol{D}(t)\|$.

**Lemma C.9.** *For any constant* $c_5 < \min\{2\beta, \frac{d}{d+1}\}$, *if* $m > \frac{3\eta c_2}{\frac{2d}{d+1} - 2c_5}$ *and* $m > \frac{\eta c_2}{2\beta - c_5}$, *there exists a constant* $c_3$ *such that* $\|\boldsymbol{D}(t)\| \leq c_3 \sqrt{nm}$.

*Proof.* First we analyze the right hand side of equation (17). We can get

R.H.S.
$$= -\frac{8\eta\lambda_1}{mn^2} \left( \boldsymbol{F}(t)^\top \boldsymbol{D}(t) + (\boldsymbol{F}(t)^\top \boldsymbol{v}_1)(\boldsymbol{D}(t)^\top \boldsymbol{v}_1) - \frac{\eta}{2}(\boldsymbol{D}(t)^\top \boldsymbol{v}_1)^2 \lambda_1 \right.$$
$$\left. -\frac{\eta}{2}\boldsymbol{R}(t)^\top \boldsymbol{\Gamma}(t)\boldsymbol{R}(t) - \eta\boldsymbol{R}(t)^\top \boldsymbol{\Gamma}(t)\left(\boldsymbol{v}_1\boldsymbol{v}_1^\top \boldsymbol{D}(t)\right) - \frac{\eta}{mn}\boldsymbol{R}(t)^\top \left(\frac{m}{d}\mathbf{X}^\top \mathbf{X}\right)\boldsymbol{R}(t) \right)$$
$$\leq -\frac{8\eta\lambda_1}{mn^2} \left( \|\boldsymbol{D}(t)\|^2 + (\boldsymbol{D}(t)^\top \boldsymbol{v}_1)^2 - \frac{\eta}{2}(\boldsymbol{D}(t)^\top \boldsymbol{v}_1)^2 \lambda_1 - \frac{\eta}{2}\|\boldsymbol{R}(t)\|^2 \|\boldsymbol{\Gamma}(t)\|_2 \right.$$
$$\left. -\eta\|\boldsymbol{R}(t)\| \cdot \|\boldsymbol{D}(t)\| \cdot \|\boldsymbol{\Gamma}(t)\|_2 - \frac{\eta\lambda_1}{nd}\|\boldsymbol{R}(t)\|^2 + \boldsymbol{Y}^\top \boldsymbol{D}(t) + \left(\boldsymbol{Y}^\top \boldsymbol{v}_1\right)\left(\boldsymbol{D}(t)^\top \boldsymbol{v}_1\right) \right)$$
$$\leq -\frac{8\eta\lambda_1}{mn^2} \left( 2\beta(\boldsymbol{D}(t)^\top \boldsymbol{v}_1)^2 + \|\boldsymbol{R}(t)\|^2 - \frac{c_2\eta}{2m}\|\boldsymbol{R}(t)\|^2 \right.$$
$$\left. -\frac{c_2\eta}{m}\|\boldsymbol{R}(t)\|\|\boldsymbol{D}(t)\| - \frac{1}{d+1}\|\boldsymbol{R}(t)\|^2 - \|\boldsymbol{Y}\| \cdot \|\boldsymbol{D}(t)\| - \|\boldsymbol{Y}\|(\boldsymbol{D}(t)^\top \boldsymbol{v}_1) \right)$$
$$\leq -\frac{8\eta\lambda_1}{mn^2} \left( 2\beta(\boldsymbol{D}(t)^\top \boldsymbol{v}_1)^2 + \left(\frac{d}{d+1} - \frac{\eta c_2 n}{2m}\right)\|\boldsymbol{R}(t)\|^2 - \frac{c_2\eta}{m}\|\boldsymbol{D}(t)\|^2 - 2\|\boldsymbol{Y}\| \cdot \|\boldsymbol{D}(t)\| \right)$$
$$\leq -\frac{8\eta\lambda_1}{mn^2} \left( c_5\|\boldsymbol{D}(t)\|^2 - 2\|\boldsymbol{Y}\| \cdot \|\boldsymbol{D}(t)\| \right)$$

Here the third inequality follows from inequality (5) which is $\frac{\eta\lambda_1}{nd} < \frac{1}{d+1}$, and the last inequality holds because $2\beta - \frac{c_2\eta}{m} > c_5$ and $\frac{d}{d+1} - \frac{3\eta c_2}{2m} > c_5$ by the restriction of $m$ in this lemma.

On the other hand, the left hand side of equation (17) is $\Lambda^*(t+1) - \Lambda^*(t) \geq 0 - \Lambda(t) > -\frac{4}{\eta}$. Here the first inequality follows because of $\Lambda^*(t) \geq 0$ and Corollary C.2, and the second inequality holds because of Assumption C.5.

Hence we have $\frac{4}{\eta} > \frac{8}{n^2}\frac{\eta\lambda_1}{m}(c_5\|\boldsymbol{D}(t)\|^2 - 2\|\boldsymbol{Y}\|\|\boldsymbol{D}(t)\|)$. Now if $\|\boldsymbol{D}(t)\| > \frac{2\|\boldsymbol{Y}\|}{c_5} + n\sqrt{m}\frac{1}{\eta\sqrt{2\lambda_1}}$, the inequality cannot hold. Note that $\|\boldsymbol{Y}\| = \sqrt{n}$ and $\lambda_1 \in \Theta(n)$, and also from inequality (5), $\eta < \frac{\lambda_1}{n} = O(1)$, we finish the proof. $\qquad\square$

Now we can give a lemma on $\boldsymbol{R}(t) := (\boldsymbol{I} - \boldsymbol{v}_1\boldsymbol{v}_1^\top)\boldsymbol{D}(t)$. The proof idea is similar to Lemma 3.5 in Section 3.

**Lemma C.10.** *Define* $\boldsymbol{R}'(t) := (\boldsymbol{I} - \eta\boldsymbol{M}^*(t)(\boldsymbol{I} - \boldsymbol{v}_1\boldsymbol{v}_1^\top))\boldsymbol{R}'(t-1)$. *We have* $\|\boldsymbol{R}(t) - \boldsymbol{R}'(t)\| = O(\frac{\sqrt{n^3 d}}{\lambda_r\sqrt{m}})$.

*Proof.* We consider the update rule for $\boldsymbol{R}(t)$, whose proof is in Lemma E.12:

$$\boldsymbol{R}(t+1) = (I - \eta\boldsymbol{M}^*(t))\boldsymbol{R}(t) + \eta\left(\boldsymbol{v}_1\boldsymbol{v}_1^\top \boldsymbol{\Gamma}(t) - \boldsymbol{\Gamma}(t)\boldsymbol{v}_1\boldsymbol{v}_1^\top\right)\boldsymbol{D}(t) \qquad (18)$$

Hence if we denote $\boldsymbol{e}_1(t) = \boldsymbol{R}(t+1) - (\boldsymbol{I} - \eta\boldsymbol{M}^*(t)(\boldsymbol{I} - \boldsymbol{v}_1\boldsymbol{v}_1^\top))\boldsymbol{R}(t)$, then we have

$$e_1(t) = -\eta M^* v_1 v_1^\top R(t) + \eta \left( v_1 v_1^\top \Gamma(t) - \Gamma(t) v_1 v_1^\top \right) D(t) = \eta \left( v_1 v_1^\top \Gamma(t) - \Gamma(t) v_1 v_1^\top \right) D(t).$$

Using the upper bound of $\|D(t)\|$ in Lemma C.9, and Assumption C.3 we have

$$\|e_1(t)\| \le 2\eta \|\Gamma(t)\| \|D(t)\| \le 2\eta \frac{c_2}{m} c_3 \sqrt{nm} = 2c_2 c_3 \eta \sqrt{\frac{n}{m}}$$

.

Now we consider the sequence $R'(t)$.

First by Corollary C.3, we can see that the eigenvalues of $(M^*(t) - \Gamma^*(t))(I - v_1 v_1^\top)$ are all the eigenvalues of $M^*(t) - \Gamma^*(t)$ except the largest one, hence are in $(\tau, 2/\eta - \tau)$. Hence because $m = \Omega(n^2)$, if we let $\frac{\lambda_r}{nd} < \frac{c_2 + c_6}{m}$, then by Corollary C.3 we have all eigenvalues of $M^*(t)(I - v_1 v_1^\top)$ are in $(\tau', 2/\eta - \tau')$ where $\tau' = \frac{\lambda_r}{nd}$ . Hence $\|I - \eta M(t)(I - v_1(t) v_1(t)^\top)\| \le 1 - \eta \tau'$.

By the calculations above we can get

$$
\begin{aligned}
\|R(t+1) - R'(t+1)\| &= \|(I - \eta M^*(t)(I - v_1 v_1^\top))(R(t) - R'(t)) + e_1(t)\| \\
&\le \|(I - \eta M(t)(I - v_1 v_1^\top))(R(t) - R'(t))\| + \|e_1(t)\| \\
&\le \|I - \eta M(t)(I - v_1 v_1^\top)\| \|R(t) - R'(t)\| + \|e_1(t)\| \\
&\le \|R(t) - R'(t)\|(1 - \eta \tau') + 2c_2 c_3 \eta \sqrt{\frac{n}{m}}.
\end{aligned}
$$

Thus if we denote $\|R(t) - R'(t)\| - \frac{2c_2 c_3 \eta \sqrt{\frac{n}{m}}}{\eta \tau'}$ by $p(t)$, and replace $\|R(t) - R'(t)\|$ in the inequality above by $p(t) + \frac{2c_2 c_3 \eta \sqrt{\frac{n}{m}}}{\eta \tau'}$, we can get $|p(t+1)| \le |p(t)|(1 - \eta \tau')$. Hence, we can have $|p(t)| < |p(0)| = \frac{2c_2 c_3 \eta \sqrt{\frac{n}{m}}}{\eta \tau'}$ for any time $t$. Therefore, we can obtain $\|R(t) - R'(t)\| < \frac{2c_2 c_3 \eta \sqrt{\frac{n}{m}}}{\eta \tau'} + |p(0)| = \frac{4c_2 c_3 \sqrt{\frac{n}{m}}}{\tau'} = O(d\sqrt{\frac{n^3}{m}}/\lambda_r)$.

$\square$

Now based on Theorem C.10, we can give an upper bound of $\|R(t)\|$. Because $\|R'(t)\|$ is always decreasing and $\|R'(0)\| = \sqrt{n}$, hence if $m = \Omega(n^2 d^2 / \lambda_r^2)$, we can have $O(d\sqrt{\frac{n^3}{m}}/\lambda_r) = O(\sqrt{n})$. Hence

$$\|R(t)\| \le \|R'(t)\| + O(\sqrt{n}) = O(\sqrt{n}). \tag{19}$$

Let $\|R(t)\| \le c_7 \sqrt{n}$.

Next we can use Assumption C.4 to prove that $D(t)^\top v_1$ increases geometrically when $\Lambda > 2/\eta$, which then causes the drop of the sharpness.

**Lemma C.11.** *Let* $\rho^* = \frac{(c_2 + c_6) c_7}{2 c_3}$, $\rho = \rho^* + \frac{\eta}{2} \frac{5 c_2 + c_6}{m}$, $\epsilon_1 = 2\eta(\rho^* - \frac{(c_2 + c_6) c_7}{2 c_3})$. *If* $\Lambda(t), \Lambda(t+1) > \frac{2}{\eta}(1 + \rho)$, *we have* $|D(t+1)^\top v_1| > (1 + \epsilon_1) D(t)^\top v_1$.

*Proof.*
$$
\begin{aligned}
D(t+1)^\top v_1 &= D(t)^\top \left( I - \eta M^*(t) \right) v_1 \\
&= D(t)^\top \left( I - \eta \gamma^*(t) \mathbf{X}^\top \mathbf{X} \right) v_1 - \eta D(t)^\top \Gamma^*(t) v_1 \\
&= \left( 1 - \eta \gamma^*(t) \lambda_1 \right) D(t)^\top v_1 - \eta D(t)^\top \Gamma^*(t) v_1
\end{aligned}
$$

First by Lemma C.8 and Corollary C.2:
$$
\begin{aligned}
\gamma^*(t) \lambda_1 &= v_1^\top \left( k_s (M(t+1) - \Gamma(t+1)) + (1 - k_s)(M(t) - \Gamma(t)) \right) v_1 \\
&\ge k_s \Lambda^*(t+1) + (1 - k_s) \Lambda^*(t) - \frac{2c_2}{m} \\
&\ge \frac{2(1 + \rho)}{\eta} - \frac{4c_2}{m}
\end{aligned}
$$

Also we have $\left|\boldsymbol{D}(t)^\top \boldsymbol{\Gamma}^*(t)\boldsymbol{v}_1\right| \le \frac{c_2+c_6}{m}\|\boldsymbol{D}(t)\| \le \frac{c_2+c_6}{m}(|\boldsymbol{D}(t)^\top \boldsymbol{v}_1| + \|\boldsymbol{R}(t)\|)$.

Hence

$$
\begin{aligned}
|\boldsymbol{D}(t+1)^\top \boldsymbol{v}_1| &\ge |1 - \eta\gamma^*(t)\lambda_1||\boldsymbol{D}(t)^\top \boldsymbol{v}_1| - \eta|\boldsymbol{D}(t)^\top \boldsymbol{\Gamma}^*(t)\boldsymbol{v}_1| \\
&\ge |-1 - 2\rho + \frac{4c_2\eta}{m} + \frac{(c_2+c_6)\eta}{m}| \\
&\ge (1 + 2\eta\rho^*)|\boldsymbol{D}(t)^\top \boldsymbol{v}_1| - \eta\frac{c_2+c_6}{m}c_7\sqrt{n} \\
&\ge (1 + \epsilon_1)|\boldsymbol{D}(t)^\top \boldsymbol{v}_1|
\end{aligned}
$$

Here the third inequality holds by the definition of $\rho^*$. $\qquad\square$

Now we state a lemma which proves that if $\boldsymbol{D}(t)^\top \boldsymbol{v}_1$ is large enough, then the sharpness will decrease in the next iteration.

**Lemma C.12.** *Assume $m = \Omega(\eta)$. There exists constant $c_8, c_9 > 0$ such that if $\boldsymbol{D}(t)^\top \boldsymbol{v}_1 > c_8\sqrt{n}$, then $\boldsymbol{v}_1^\top \boldsymbol{M}(t+1)\boldsymbol{v}_1 - \boldsymbol{v}_1^\top \boldsymbol{M}(t)\boldsymbol{v}_1 < -c_9/m$.*

*Proof.* Recall Equation (17). Also we have $\|\boldsymbol{R}(t)\| \le c_7\sqrt{n}$ by (19). Hence

$$
\begin{aligned}
&\Lambda^*(t+1) - \Lambda^*(t) \\
&= -\frac{8\eta\lambda_1}{mn^2}\Big(\boldsymbol{F}(t)^\top \boldsymbol{D}(t) + (\boldsymbol{F}(t)^\top \boldsymbol{v}_1)(\boldsymbol{D}(t)^\top \boldsymbol{v}_1) - \frac{\eta}{2}(\boldsymbol{D}(t)^\top \boldsymbol{v}_1)^2\Lambda^*(t) \\
&\quad -\frac{\eta}{2}\boldsymbol{R}(t)^\top \boldsymbol{\Gamma}(t)\boldsymbol{R}(t) - \eta\boldsymbol{R}(t)^\top \boldsymbol{\Gamma}(t)\left(\boldsymbol{v}_1\boldsymbol{v}_1^\top \boldsymbol{D}(t)\right) - \boldsymbol{R}(t)^\top \left(\frac{m}{d}\mathbf{X}^\top \mathbf{X}\right)\boldsymbol{R}(t)\frac{\eta}{mn}\Big) \\
&\le -\frac{8\eta\lambda 1}{mn^2}\Big(2\beta(\boldsymbol{D}(t)^\top \boldsymbol{v}_1)^2 + \|\boldsymbol{R}(t)\|^2 - \frac{\eta c_2}{2m}\|\boldsymbol{R}(t)\|^2 \\
&\quad -\frac{c_2\eta}{m}\|\boldsymbol{R}(t)\| \cdot \|\boldsymbol{D}(t)\| - \frac{\eta\lambda_1}{dn}\|\boldsymbol{R}(t)\|^2 - 2\|\boldsymbol{Y}\| \cdot \|\boldsymbol{D}(t)\|\Big) \\
&\le -\frac{8\eta\lambda_1}{mn^2}\Big(2\beta(\boldsymbol{D}(t)^\top \boldsymbol{v}_1)^2 + \|\boldsymbol{R}(t)\|^2(1 - \frac{c_2\eta}{2m}) \\
&\quad -\frac{c_2\eta}{m}\|\boldsymbol{R}(t)\| \cdot (\|\boldsymbol{R}(t)\| + |\boldsymbol{D}(t)^\top \boldsymbol{v}_1|) - \frac{\eta\lambda_1}{dn}\|\boldsymbol{R}(t)\|^2 - 2\|\boldsymbol{Y}\| \cdot (\|\boldsymbol{R}(t)\| + |\boldsymbol{D}(t)^\top \boldsymbol{v}_1|)\Big) \\
&\le -\frac{8\eta\lambda_1}{mn^2}\Big(2\beta\left(\boldsymbol{D}(t)^\top \boldsymbol{v}_1\right)^2 - (\frac{\eta c_7 c_2\sqrt{n}}{m} + 2\sqrt{n})\cdot |\boldsymbol{D}(t)^\top \boldsymbol{v}_1| \\
&\quad +c_7^2 n(1 - \frac{c_2\eta}{2m} - \frac{c_2\eta}{m} - \frac{\eta\lambda_1}{dn}) - 2c_7 n\Big) \\
&= -\frac{8\eta\lambda_1}{mn^2}\left(2\beta(\boldsymbol{D}(t)^\top \boldsymbol{v}_1)^2 - c_{10}\left(\boldsymbol{D}(t)^\top \boldsymbol{v}_1\right) + c_{11}\right)
\end{aligned}
$$

Here $c_{10} := \eta c_7\sqrt{n} \cdot \frac{c_2}{m} + 2\sqrt{n} = \Theta(\sqrt{n})$, $c_{11} := c_7^2 n\left(1 - \frac{c_2\eta}{2m} - \frac{c_2\eta}{m} - \frac{\eta\lambda_1}{dn}\right) - 2c_7 n = Nn)$, and the first inequality holds because of Assumption C.5, Assumption C.3 and $\|\boldsymbol{D}(t)\|^2 = (\boldsymbol{D}(t)^\top \boldsymbol{v}_1)^2 + \|\boldsymbol{R}(t)\|^2$, the second inequality holds because of $\|\boldsymbol{D}(t)\| \le |\boldsymbol{D}(t)^\top \boldsymbol{v}_1| + \|\boldsymbol{R}(t)\|$, and the third inequality follows from the upper bound (19) of $\|\boldsymbol{R}(t)\|$.

Note that it is a quadratic function of $\boldsymbol{D}(t)^\top \boldsymbol{v}_1$. Hence if $\boldsymbol{D}(t)^\top \boldsymbol{v}_1 > \frac{|c_{10}|}{2\beta} + \sqrt{\frac{|c_{11}|}{2\beta}}$, we have

$$
\Lambda^*(t+1) - \Lambda^*(t) < -\frac{8\eta\lambda_1}{n^2 m}\cdot \frac{|c_{10}|\sqrt{|c_{11}|}}{2\beta} = \Theta(\frac{\eta}{m})
$$

Hence we finish the proof. $\qquad\square$

Now we can prove the final part (the first conclusion) of Theorem 4.

**Lemma C.13.** *Let $\rho$ be the one defined in Theorem C.11. If $\Lambda(t_0) > \frac{2}{\eta}(1 + \rho)$ for some $t_0$, there exists some $t_1 > t_0$ such that $\Lambda(t_1) < \frac{2}{\eta}(1 + \rho)$.*

*Proof.* Otherwise, for any $t > t_0$, $\Lambda(t) > \frac{2}{\eta}(1 + \rho)$. Then by Theorem C.11, $\boldsymbol{D}(t)^\top \boldsymbol{v}_1$ increases geometrically, hence there exists $t_2 > t_0$, such that $\boldsymbol{D}(t)^\top \boldsymbol{v}_1 > c_8 \sqrt{n}$. Now by Lemma C.12, each iteration $\Lambda^*(t)$ will decrease by at least a fixed amount. Hence there must exist a time $t_3 > t_0$ such that $\Lambda^*(t_3) < \frac{2}{\eta} - 2c_2/m$. Then by Corollary C.2, we get a contradiction. $\qquad\square$

## C.4 Our Results and the NTK Regime

In this subsection, we explain why our results (Theorem 3, Theorem 4) are sufficiently different from the quadratic setting (e.g., linear regression) or the recent convergence analysis in NTK setting.

A key requirement in the convergence analysis in the NTK regime is that the learning rate is very small and the GD trajectory almost tracks the gradient flow, hence converges to the global minimum. However, we consider typical learning rate used in practice, which can be much larger. In particular, $\eta > 2/\Lambda$ can happen in our setting, which causes instability (i.e., such as the growth of loss in Lemma C.11) along the training trajectory. Such instability cannot be captured by any existing convergence analysis in NTK regime at all. Hence, all existing NTK convergence results do not directly apply here.

Equally importantly, we find that even when $\boldsymbol{W}(t)$ changes slightly (several orders of magnitude smaller than its initialization), PS and EOS still happen with a not so small learning rate $\eta$. To support our claim, we include the experimental results in Appendix D.2.3. In Figure 22, we can see that the initialization $\boldsymbol{W}(0)$ is much larger than the change of $\boldsymbol{W}(t)$ and the norm of $\boldsymbol{W}(t)$ grows larger when $m$ becomes larger. However, we still observe that PS and EOS occur in this setting. Hence, the setting we study in this paper and our results are intrinsically different from the quadratic setting (in which case EOS cannot happen).

Last, in our proofs in Section C.2 and C.3, our current bound requires that $m = \Omega(n^2)$ and we also assume $\lambda_r = \Omega(1)$. This may create an impression that we need a very wide network which operates in the NTK regime. However, we remark that if our analysis can be tightened to $m = O(n)$, one can formally prove that $\|\boldsymbol{A}(t)\|$ can actually change significantly ($\|\boldsymbol{A}(t)\|^2 - \|\boldsymbol{A}(0)\|^2$) has the same scale as the initialization $\|\boldsymbol{A}(0)\|^2$), resulting a significant change of sharpness as well, hence beyond the NTK regime. For example, in the proof of EOS (Section C.3), we prove a loose $O(\sqrt{mn})$ upper bound of $\|\boldsymbol{D}(t)\|$ (Lemma C.9). However by Lemma C.12, when $\|\boldsymbol{D}(t)\|$ reaches $O(\sqrt{n})$, the sharpness starts dropping quickly. So if a better upper bound of $\|\boldsymbol{D}(t)\| = O(\sqrt{n})$ can be proven (this is true empirically for all of our experiments), the width $m$ can be set to $\Theta(n)$, and this suffices to implies a significant change of $\|\boldsymbol{A}(t)\|^2$. We leave these improvements as future directions.

# D Verification for Assumptions

In this section, we first justify the assumptions we made in Section 3 and Section 4 empirically. Then we present the experiment described in Section 5.

## D.1 Assumptions in Section 3

In this subsection, we conduct experiments to verify the assumptions in Section 3. The detailed experiment settings can be found in Appendix A.

### D.1.1 $\mathrm{M}_A$ has small eigenvalues

In Section 3.2, we mentioned that the largest eigenvalue of $\boldsymbol{M_A} := (\frac{\partial \boldsymbol{F}}{\partial \boldsymbol{A}})(\frac{\partial \boldsymbol{F}}{\partial \boldsymbol{A}})^\top$ is much smaller than the sharpness. We verify this assumption under different settings in Figure 13, including a fully-connected linear network, a fully-connected network with tanh activation and a convolutional one. Observe that $\|\boldsymbol{M_A}\|$ (the blue curve) is very close to 0 and hardly increases during the training process along the whole trajectory.

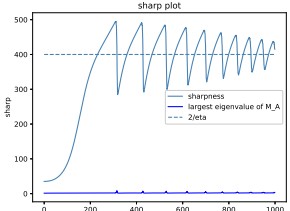
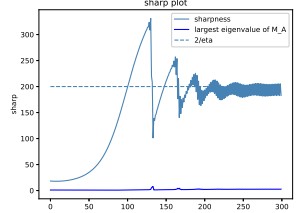
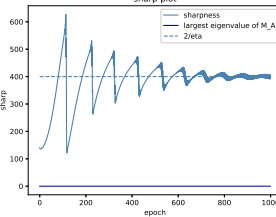

(a) Fully-connected Linear Network  (b) Fully-connected tanh Network  (c) Convolutional tanh Network

Figure 13: In this figure, we show that $\|M_A\|$ is much smaller than the sharpness. Note that $\|M_A\|$ (the blue curve) is very close to 0 and hardly increases during the training process along the whole trajectory. The sharpness (the steel-blue curve) strictly dominates $\|M_A\|$ for all time. The detailed experiment settings can be found in Appendix A.

### D.1.2    Assumption 3.7

In this assumption, we assume that there is a large gap between the largest and the second largest eigenvalue, and thus the second largest eigenvalue is always below $1/\eta$. We verify the outlier assumption by calculating the largest and the second largest eigenvalue of $M(t)$. In Sagun et al. [28, 29], the sharpness is much larger than the largest eigenvalue in the bulk (the $(K+1)$-th largest eigenvalue of $M$ where $K$ is the number of classes). In our binary setting $K = 1$. In Figure 16, we show that the largest eigenvalue indeed dominates the second one, and the second one never reaches $1/\eta$, which verifies Assumption 3.7.

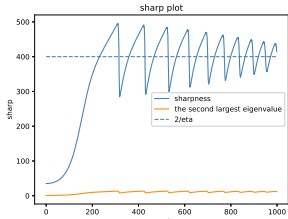
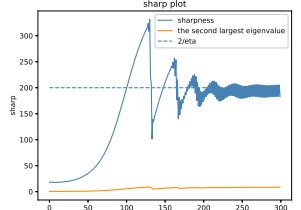
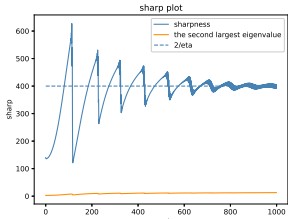

(a) Fully-connected linear network  (b) Fully-connected tanh network  (c) Convolutional   network   with ELU activation

Figure 14: The second largest eigenvalue (the orange curve) of $M$ is much smaller than the sharpness (the steel-blue curve). Also, the second largest eigenvalue never reaches $2/\eta$.

### D.1.3    Assumption 3.2 (First Order Approximation of GD)

In Assumption 3.2, we assume the gradient descent trajectory is close to the first order approximation. To verify that the first order term is indeed dominant along the trajectory, for both the residual $D(t)$ and the output layer norm $\|A(t)\|^2$, we plot the norms of the actual GD update, the first order approximation order approximation and the higher order terms of the update rule in Figure 15. Observe that in the progressive sharpening phase, the first order approximation is almost the same as the actual gradient update; while in the EOS phase, the first order approximation is still close to the actual gradient most of the time. We can see that the norm of the higher order terms spikes occasionally, but when this happens the first order term spikes much higher.

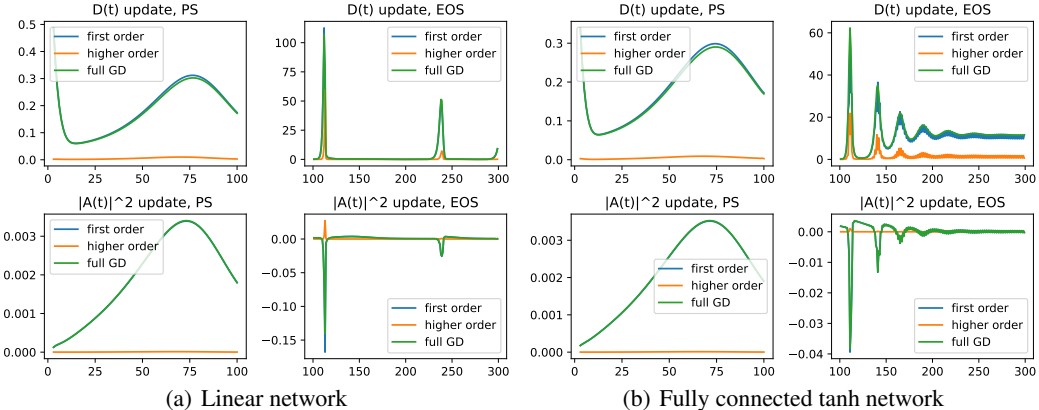

(a) Linear network          (b) Fully connected tanh network

Figure 15: This figure shows the norm of the full gradient $\|\boldsymbol{D}(t+1) - \boldsymbol{D}(t)\|$, the first order approximation $\|\eta \boldsymbol{M}(t)\boldsymbol{D}(t)\|$ and the higher order terms $\|\boldsymbol{D}(t+1) - \boldsymbol{D}(t) + \eta \boldsymbol{M}(t)\boldsymbol{D}(t)\|$ in the training process. Observe that in both the $\|\boldsymbol{A}(t)\|^2$ and $\boldsymbol{D}(t)$ dynamics, the first order approximation (the blue curve) almost overlaps with the actual update (the green curve) in the PS phase. While in the EOS phase, though the norm of the higher order terms (the orange curve) spikes, the first order approximation is still close to the actual update.

### D.1.4    Assumption 3.3 (Gradient flow for the PS phase)

In Assumption 3.3, we assume that in the progressive sharpening phase, the gradient descent trajectory is close to the gradient flow trajectory. We refer to Appendix J in Cohen et al. [6] for more experiments about this claim. Here for readers' convenience, we duplicate their Figure 29 in Appendix J as Figure 16.

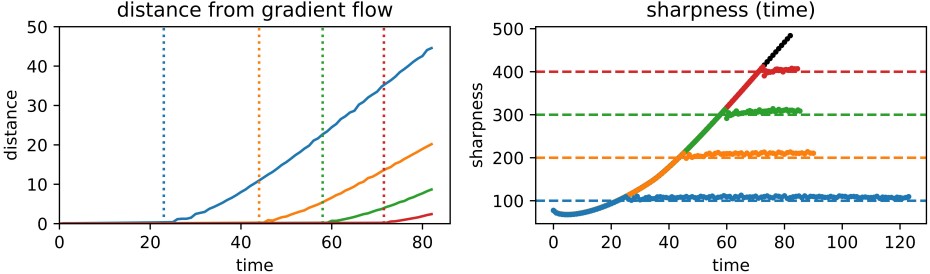

Figure 16: **Left**: the picture shows the $\ell_2$ distance between the gradient flow trajectory at $t$ iteration and the gradient descent trajectory at $t/\eta$ iteration. The vertical dotted line is the time when the sharpness reaches $2/\eta$. **Right: the plot of sharpness where (iteration $\times$ learning rate) is on the x-axis.**

### D.1.5    Assumption 3.5 (iii) (principal directions moves slowly)

In Figure 17 we verify Assumption 3.5 (iii) and as well as the discussion after Assumption 3.5 (i). In general models, we find that the eigenvectors corresponding to small eigenvalues may change drastically. But for the largest eigenvector, it indeed changes slowly from the initialized direction. In Figure 17, we can see that over a long training time the similarity of $\boldsymbol{v}_1(t)$ with its initialization is still larger than at least 0.98. Mulayoff et al. [24] also proved that near the minima, the top eigenvectors of the Hessian matrices tend to align. That is, the directions of these top eigenvectors are approximately parallel. This fact also corroborates our assumption near the minima.

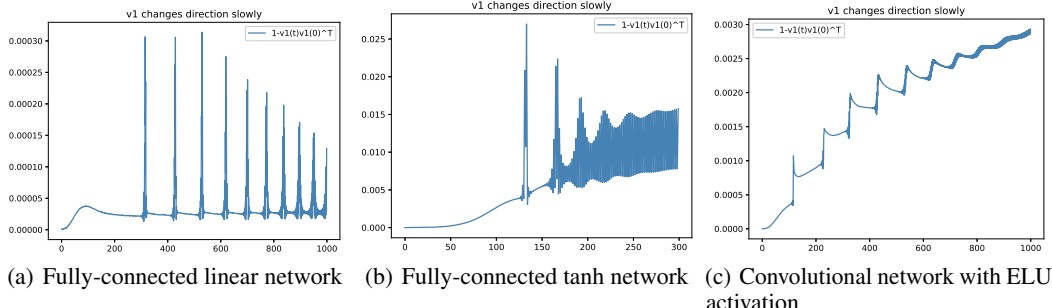

(a) Fully-connected linear network    (b) Fully-connected tanh network    (c) Convolutional network with ELU activation

Figure 17: In this figure, we show that under various architectures, the first eigenvector of $M$ changes slowly throughout the training process, even when Edge of Stability phenomenon occurs.

## D.2 Assumptions in Section 4

In this subsection, we present the assumptions in Section 4, and conduct experiments to verify them. We add a scale coefficient $\frac{1}{\sqrt{m}}$ in the linear network to be consistent with the settings of theoretical analysis.

### D.2.1 Assumption 4.1

In Assumption 4.1, we assume the ratio $\lambda_i/\lambda_{i+1}$ between the two adjacent eigenvalues of $\mathbf{X}^\top\mathbf{X}$ is bounded by a small constant. In the 1000-example subset of CIFAR-10, we verify this assumption by experiments. We plot the eigenvalues and their ratio in Figure 18. It shows that Assumption 4.1 holds and the constant $\chi \approx 38$, since almost all the ratios are close to 1, and the largest ratio is $\lambda_1/\lambda_2 \approx 38$.

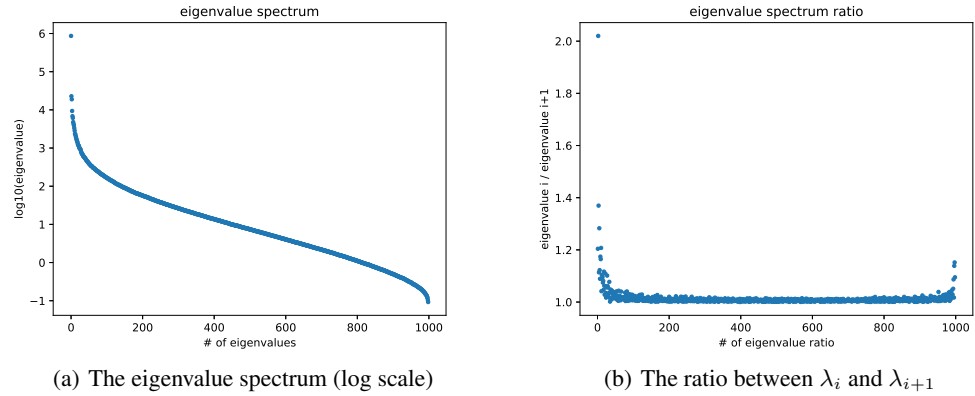

(a) The eigenvalue spectrum (log scale)    (b) The ratio between $\lambda_i$ and $\lambda_{i+1}$

Figure 18: (a): The full eigenvalue spectrum of of $\mathbf{X}^\top\mathbf{X}$. Observe that the maximal ratio $\max\{\lambda_i/\lambda_{i+1}\} = \lambda_1/\lambda_2$. (b): The ratio between two adjacent eigenvalues $\lambda_i/\lambda_{i+1}$, $i \geq 2$. We exclude the largest eigenvalue to make the figure clearer.

If we further consider a mean-subtracted version of this subset of CIFAR-10 (See Figure 19), we can reduce the ratio to $\chi \approx 3$.

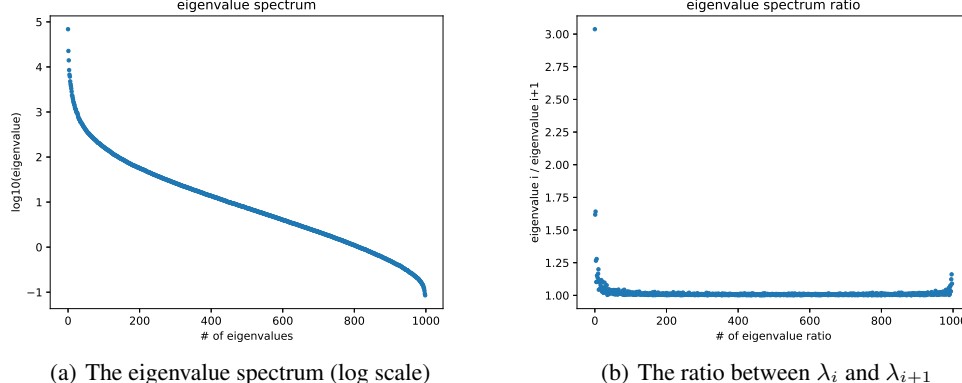

(a) The eigenvalue spectrum (log scale)    (b) The ratio between $\lambda_i$ and $\lambda_{i+1}$

Figure 19: (a): The full eigenvalue spectrum of of $\mathbf{X}^\top \mathbf{X}$ when samples' mean is subtracted. (b): the ratio between two adjacent eigenvalues $\lambda_i/\lambda_{i+1}$. In Figure (b), observe that the maximal ratio $\max\{\lambda_i/\lambda_{i+1}\} = \lambda_1/\lambda_2 \approx 3$. **Remark.** Note we should exclude the minimal eigenvalue if the mean of examples is subtracted, because when we subtract the mean from the dataset, it cannot be full rank. The minimal eigenvalue of $\mathbf{X}^\top \mathbf{X}$ is 0, thus it is unnecessary to check it for our assumption.

### D.2.2 Assumption 4.3

In Appendix C.2, by Theorem 3 we prove that $\|\mathbf{\Gamma}(t)\|$ is bounded by $R_w = O(1/m)$ in the progressive sharpening phase. When gradient descent enters EOS, the proof does not hold and we make this bound into an assumption (Assumption 4.3). We empirically verify that the bound can only increase by a constant factor (despite some spikes in EOS). See Figure 20. In this figure $\|\mathbf{\Gamma}(t)\| \leq \frac{24}{m}$ with $m = 40, 80, 160, 200$.

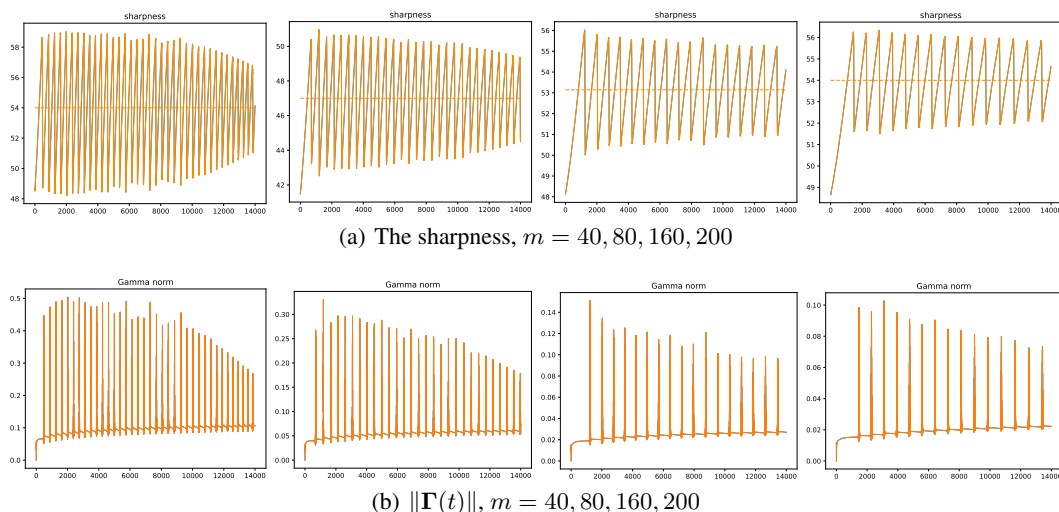

(a) The sharpness, $m = 40, 80, 160, 200$

(b) $\|\mathbf{\Gamma}(t)\|$, $m = 40, 80, 160, 200$

Figure 20: In this figure, we show that even when GD enters EOS, the bound of $\|\mathbf{\Gamma}(t)\|$ is still $\Theta(1/m)$ along the whole trajectory. The corresponding constant $c_2 \approx 24$.

Actually there are two interesting empirical facts related to $\|\mathbf{\Gamma}(t)\|$. One is the noticeable fact that $\|\mathbf{\Gamma}(t)\|$ spikes, the other is that the values of $\|\mathbf{\Gamma}(t)\|$ are overall quite small (despite the fact that $\mathbf{W}$ changes non-trivially in our experiments) and decreases as $m$ becomes larger. Our assumption tries to model the second fact (see Figure 20, $\|\mathbf{\Gamma}(t)\|$ (despite the spikes) decreases as the width $m$ grows, and the largest $\|\mathbf{\Gamma}(t)\|$ is almost $24/m$ in all these experiments). However, we admit that our results do not reflect the first fact (the spikes of $\|\mathbf{\Gamma}(t)\|$ ), and it is an interesting fact that is

worth investigating. We have strong intuition that $\|\mathbf{\Gamma}(t)\|$ only grows by at most a constant factor, but currently do not have a formal proof yet. Nevertheless, the spiking behavior does not directly contradict our assumption.

**Remark:** Note that Assumption 4.3 is not equivalent to a small movement of $\boldsymbol{W}(t)$. Actually, in the experiments above ($m = 40, 80, 160, 200$) the movement of $\boldsymbol{W}(t)$ is quite significant compared to the norm of $\boldsymbol{W}(0)$ at initialization. See Figure 21.

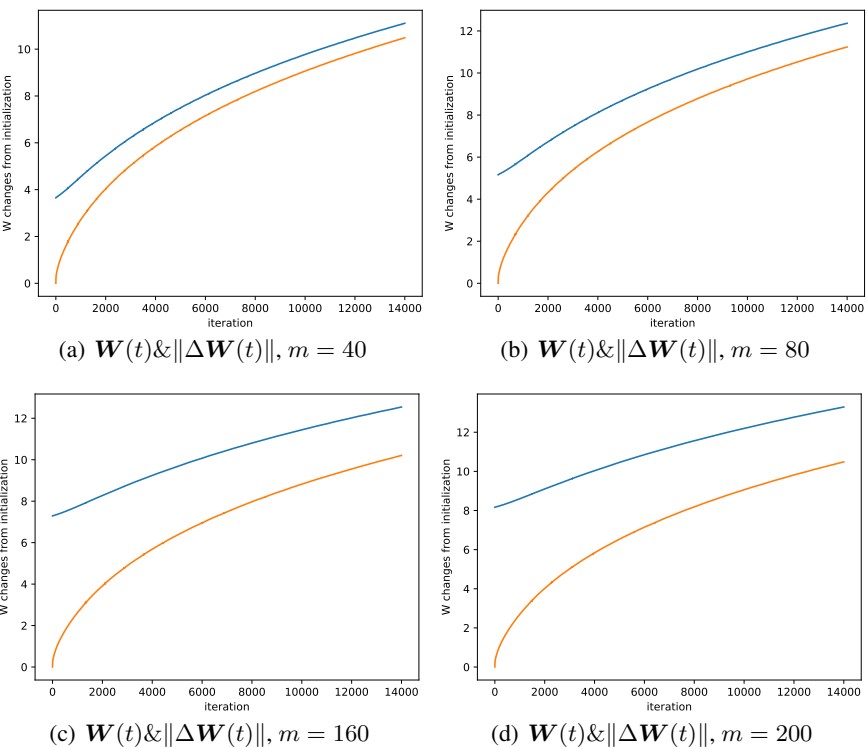

(a) $\boldsymbol{W}(t)\&\|\Delta\boldsymbol{W}(t)\|, m = 40$        (b) $\boldsymbol{W}(t)\&\|\Delta\boldsymbol{W}(t)\|, m = 80$

(c) $\boldsymbol{W}(t)\&\|\Delta\boldsymbol{W}(t)\|, m = 160$        (d) $\boldsymbol{W}(t)\&\|\Delta\boldsymbol{W}(t)\|, m = 200$

Figure 21: In this figure, we show that even the $\mathbf{\Gamma}(t)$ is small and bounded, the movement $\|\Delta\boldsymbol{W}(t)\| := \|\boldsymbol{W}(t) - \boldsymbol{W}(0)\|$ (the dark orange curve) is close to $\|\boldsymbol{W}(t)\|$ (the blue curve), implying that $\boldsymbol{W}(t)$ moves considerably.

### D.2.3 Comparison with the NTK regime: the non-quadratic property

Here we illustrate why our setting and results are sufficiently different from the quadratic setting (e.g., linear regression) or the recent convergence analysis in NTK setting. In particular, we show that even when the movement of $\boldsymbol{W}(t)$ is comparably negligible compared to the initialization $\boldsymbol{W}(0)$, EOS can still happen. Here we take a larger initialization of $\boldsymbol{W}(0)$, which is ten times of the standard initialization in order to dwarf the movement of $\boldsymbol{W}(t)$. The widths are $m = 1000, 2000, 4000, 8000$. We can see that the initialization $\boldsymbol{W}(0)$ is much larger than the change of $\boldsymbol{W}(t)$ and the norm of $\boldsymbol{W}(t)$ grows larger when $m$ becomes larger. See Figure 22. Detailed comparison is in Appendix C.4.

### D.3 Experiments in Section 5

In this subsection we show some empirical results that may reveal some interesting relation between the inner layers and the sharpness, which is not yet reflected in our theory. Our experimental results show that all layers seem to work together to influence the sharpness, and contribute to the progressive sharpening and edge of stability phenomena. In our experiment, while the sharpness is still calculated by the gradient of all parameters, we freeze some of the layers in the training process. The experimental results show that the more layers we freeze, the slower progressive sharpening happens and the weaker the oscillation of the sharpness is (See Figure 23, Figure 24). This indicates

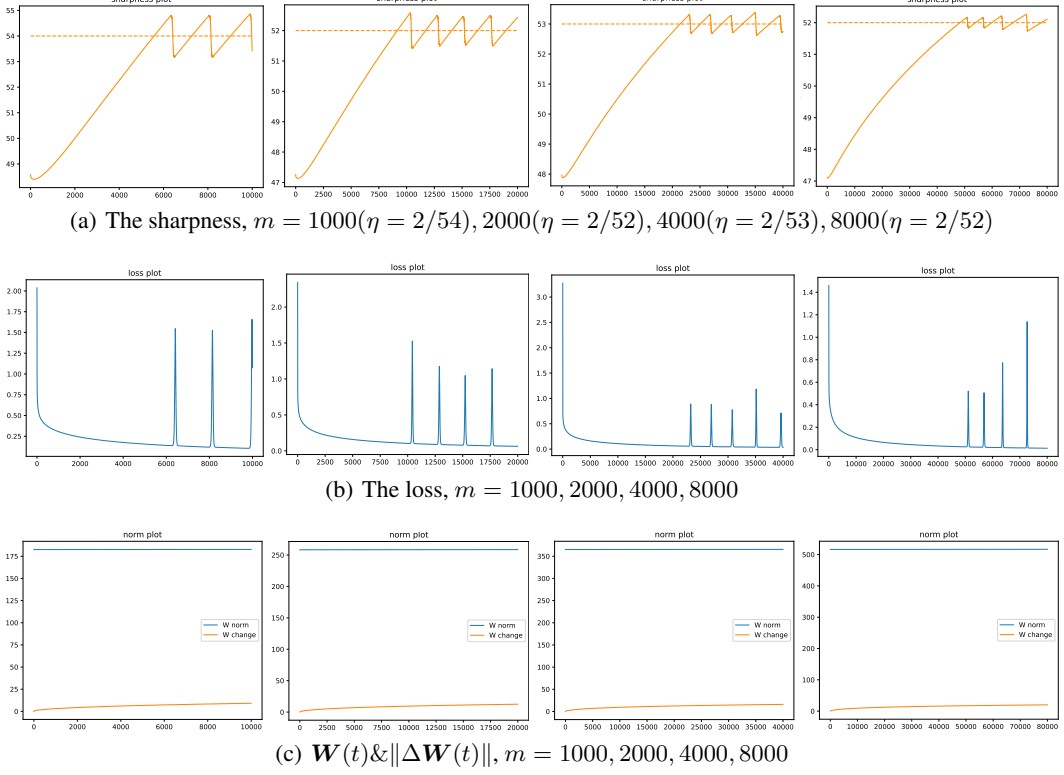

(a) The sharpness, $m = 1000(\eta = 2/54), 2000(\eta = 2/52), 4000(\eta = 2/53), 8000(\eta = 2/52)$

(b) The loss, $m = 1000, 2000, 4000, 8000$

(c) $\boldsymbol{W}(t) \& \|\Delta\boldsymbol{W}(t)\|$, $m = 1000, 2000, 4000, 8000$

Figure 22: In this figure, we show that even when $\boldsymbol{W}(t)$ stays close to its initialization, EOS can still happen. In particular, Figure (c) shows the $\|\boldsymbol{W}(t)\|$ (the blue curve) and $\|\boldsymbol{W}(t) - \boldsymbol{W}(0)\|$ (the orange curve). Here for $m = 1000, 2000, 4000, 8000$, observe that the EOS still happens when $\|\boldsymbol{W}(t)\|/\|\boldsymbol{W}(t) - \boldsymbol{W}(0)\| > 10$. That indicates the intrinsic non-quadratic property of neural networks in the EOS regime.

that layers other than the output layer has nontrivial influence on the sharpness, but since different layers seem to work in the same direction, further justifying our assumption that $\|\boldsymbol{A}\|$ is positively related with the sharpness.

**Fully-connected tanh Network:**

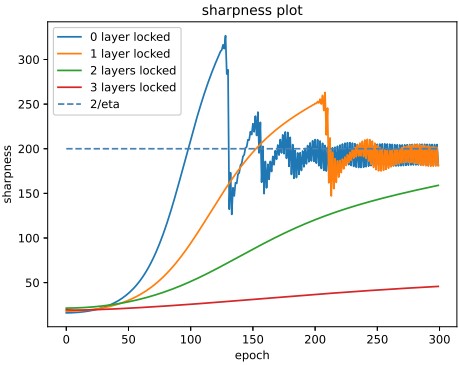

Figure 23: We conduct the experiment on a 5-layer tanh fully-connected network. Four lines in the plot show the sharpness for four independent training processes but with 0,1,2,3 outer layers locked. All other hyper-parameters are the same. The result shows that all layers have cooperative effect on the sharpness.

**Fully-connected Linear Network:**

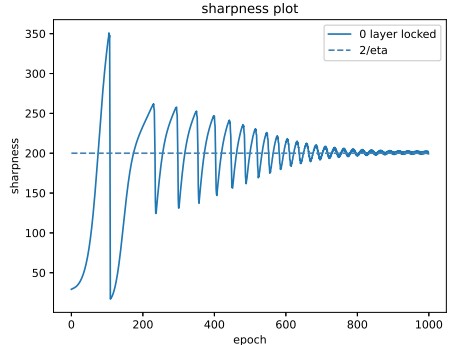

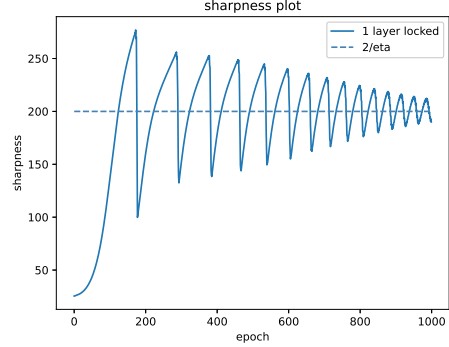

(a) Fully-connected linear network 0 layer frozen

(b) Fully-connected linear network 1 layer frozen

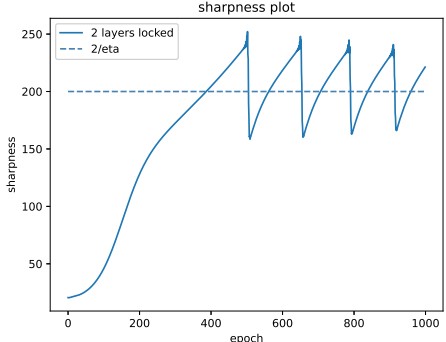

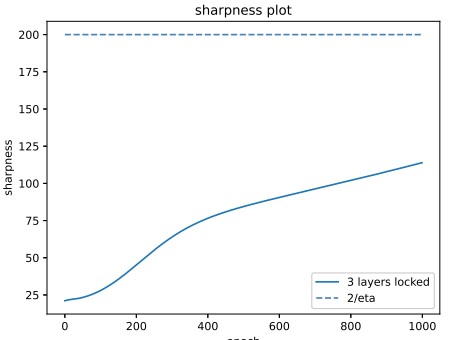

(c) Fully-connected linear network 2 layers frozen

(d) Fully-connected linear network 3 layers frozen

Figure 24: The sharpness after 0,1,2,3 outer layers are locked. Details refer to Figure 23.

# E  Missing proofs

## E.1  Proofs in Section 3

In this section, we provide the missing proofs in Section 3.

**Lemma E.1.** *(Lemma 3.1) For all $t$ in Phase I, under Assumption 3.5 (i) and 3.3, it holds that $\boldsymbol{D}(t)^\top \boldsymbol{F}(t) < 0$.*

*Proof.* Consider the inner product $\boldsymbol{D}(t)^\top \boldsymbol{v}_i$. Recall $\boldsymbol{v}_i$ is the $i$-th eigenvector of $\boldsymbol{M}$. By Assumption 3.5(i), $\boldsymbol{v}_i$ does not change with $t$. By Assumption 3.3, we have

$$\frac{\mathrm{d}\boldsymbol{D}(t)^\top \boldsymbol{v}_i}{\mathrm{d}t} = -\boldsymbol{D}(t)^\top \boldsymbol{M}(t)\boldsymbol{v}_i = -\lambda_i(t)\boldsymbol{D}(t)^\top \boldsymbol{v}_i$$

where $\lambda_i(t) \geq 0$ is the corresponding eigenvalue of $\boldsymbol{v}_i$.

Since $\boldsymbol{F}(0) = 0$, $\boldsymbol{D}(0) = \boldsymbol{F}(0) - \boldsymbol{Y} = -\boldsymbol{Y}$. Then the differential equation has the solution

$$\boldsymbol{D}(t)^\top \boldsymbol{v}_i = e^{-\int_0^t \lambda_i(t)\mathrm{d}t} \cdot \boldsymbol{D}(0)^\top \boldsymbol{v}_i = -e^{-\int_0^t \lambda_i(t)\mathrm{d}t} \cdot \boldsymbol{Y}^\top \boldsymbol{v}_i$$

Plug it into the expression $\boldsymbol{D}(t)^\top \boldsymbol{F}(t) = \boldsymbol{D}(t)^\top (\boldsymbol{D}(t) + \boldsymbol{Y})$ and we have:

$$\boldsymbol{D}(t)^\top \boldsymbol{F}(t) = \sum_{i=1}^n \boldsymbol{D}(t)^\top \boldsymbol{v}_i (\boldsymbol{D}(t)^\top \boldsymbol{v}_i + \boldsymbol{Y}^\top \boldsymbol{v}_i) = -\sum_{i=1}^n (\boldsymbol{Y}^\top \boldsymbol{v}_i)^2 e^{-\int_0^t \lambda_i(t)\mathrm{d}t}(1 - e^{-\int_0^t \lambda_i(t)\mathrm{d}t})$$

since $(\boldsymbol{Y}^\top \boldsymbol{v}_i)^2 e^{-\int_0^t \lambda_i(t)\mathrm{d}t}(1 - e^{-\int_0^t \lambda_i(t)\mathrm{d}t}) > 0$ for all $i \in [n]$, $\boldsymbol{D}(t)^\top \boldsymbol{F}(t) < 0$. $\qquad\square$

**Lemma E.2.** *(Lemma 3.2) Suppose Assumption 3.5 (iii) and 3.6 hold during this phase (with constants $\epsilon_2 > 0$ and $c > 1$). If $\Lambda(t) = (2 + \tau)/\eta$ and $\tau > \frac{1}{1-\epsilon_2-1/c} - 1$, then $\boldsymbol{D}(t)^\top \boldsymbol{v}_1(t)$ increases geometrically with factor $(1 + \tau)(1 - \epsilon_2 - 1/c) > 1$ for $t \geq t_0$ in this phase.*

*Proof.* First by (3), we have $\boldsymbol{D}(t+1)^\top \boldsymbol{v}_1(t) = \boldsymbol{D}(t)^\top (\boldsymbol{I} - \eta \boldsymbol{M}(t)) \boldsymbol{v}_1(t) = (1 - \eta \Lambda(t)) \boldsymbol{D}(t)^\top \boldsymbol{v}_1(t)$. Then we have

$$
\begin{aligned}
|\boldsymbol{D}(t+1)^\top \boldsymbol{v}_1(t+1)| &\geq |\boldsymbol{D}(t+1)^\top \boldsymbol{v}_1(t)(1 - \epsilon_2)| - \epsilon_2 \|\boldsymbol{D}(t+1)\| \\
&\geq |\boldsymbol{D}(t)^\top \boldsymbol{v}_1(t)|(1 - \epsilon_2)(\eta \Lambda(t) - 1) - (\eta \Lambda(t) - 1)\epsilon_2 \|\boldsymbol{D}(t)\| \\
&\geq |\boldsymbol{D}(t)^\top \boldsymbol{v}_1(t)|(1 - \epsilon_2)(\eta \Lambda(t) - 1) - (\eta \Lambda(t) - 1)|\boldsymbol{D}(t)^\top \boldsymbol{v}_1(t)|/c \\
&= |\boldsymbol{D}(t)^\top \boldsymbol{v}_1(t)|(\eta \Lambda(t) - 1)(1 - \epsilon_2 - 1/c) \\
&= |\boldsymbol{D}(t)^\top \boldsymbol{v}_1(t)|(1 + \tau)(1 - \epsilon_2 - 1/c)
\end{aligned}
$$

where first inequality holds due to Assumption 3.5 (iii) and the third follows from Assumption 3.6. $\square$

**Proposition E.1.** *(Proposition 3.1) If $\|\boldsymbol{D}(t)\| > \|\boldsymbol{Y}\|$, then $\boldsymbol{D}(t)^\top \boldsymbol{F}(t) > 0$.*

*Proof.* This proposition can be proved by simply noticing that $\boldsymbol{D}(t)^\top \boldsymbol{F}(t) = \boldsymbol{D}(t)^\top (\boldsymbol{D}(t) - \boldsymbol{Y}(t)) \geq \|\boldsymbol{D}\|^2 - \|\boldsymbol{D}\|\|\boldsymbol{Y}\| > 0$ when $\|\boldsymbol{D}(t)\| \geq \|\boldsymbol{Y}\|$. $\square$

**Proposition E.2.** *(Proposition 3.2) Under Assumption 3.2, if $\|\boldsymbol{D}(t)\| > \|\boldsymbol{Y}\|$, then $\|\boldsymbol{A}(t+1)\|^2 - \|\boldsymbol{A}(t)\|^2 < -\frac{4\eta}{n}(\|\boldsymbol{D}(t)\| - \|\boldsymbol{Y}\|)^2$.*

*Proof.* Use Assumption 3.2 and notice that $\boldsymbol{D}(t)^\top \boldsymbol{F}(t) \geq \|\boldsymbol{D}(t)\|^2 - \|\boldsymbol{D}(t)\|\|\boldsymbol{Y}\| > (\|\boldsymbol{D}(t)\| - \|\boldsymbol{Y}\|)^2$. $\square$

**Lemma E.3.** *(Lemma 3.3) If $\Lambda(t) < 2/\eta$, then for any vector $\boldsymbol{u} \in \mathbb{R}^n$, $\|\boldsymbol{u}^\top (\boldsymbol{I} - \eta \boldsymbol{M}(t))\| \leq (1 - \eta \alpha)\|\boldsymbol{u}\|$, where $\alpha = \min\{2/\eta - \Lambda(t), \lambda_{\min}(\boldsymbol{M}(t))\}$. In particular, replacing $\boldsymbol{u}$ with $\boldsymbol{D}(t)$, we can see $\|\boldsymbol{D}(t+1)\| \leq (1 - \eta \alpha)^2 \|\boldsymbol{D}(t)\|$.*

*Proof.* For any vector $\boldsymbol{u} \in \mathbb{R}^n$,

$$
\begin{aligned}
\|\boldsymbol{u}^\top (\boldsymbol{I} - \eta \boldsymbol{M}(t))\|^2 &= \sum_{i=1}^n \|(\boldsymbol{u}^\top \boldsymbol{v}_i(t)) \boldsymbol{v}_i(t)^\top (\boldsymbol{I} - \eta \boldsymbol{M}(t))\|^2 \\
&= \sum_{i=1}^n (1 - \eta \lambda_i)^2 (\boldsymbol{u}^\top \boldsymbol{v}_i(t))^2 \leq (1 - \eta \alpha)^2 \sum_{i=1}^n (\boldsymbol{u}^\top \boldsymbol{v}_i(t))^2 = (1 - \eta \alpha)^2 \|\boldsymbol{u}\|^2.
\end{aligned}
$$

Then the second statement is due to Assumption 3.2. $\square$

**Lemma E.4.** *(Lemma 3.4) Suppose Assumption 3.5 (iii) holds. $\boldsymbol{R}(t)$ satisfies the following:*

$$
\boldsymbol{R}(t+1) = (\boldsymbol{I} - \eta \boldsymbol{M}(t)) \boldsymbol{R}(t) + \boldsymbol{e}_1(t), \quad where \quad \|\boldsymbol{e}_1(t)\| \leq 6\sqrt{\epsilon_2} \|\boldsymbol{D}(t)\| (B_\Lambda - 1)
$$

*Proof.* We consider the update rule for $\boldsymbol{R}(t)$:

$$
\begin{aligned}
\boldsymbol{R}(t+1) &= (\boldsymbol{I} - \boldsymbol{v}_1(t+1)\boldsymbol{v}_1(t+1)^\top)(\boldsymbol{I} - \eta \boldsymbol{M}(t)) \boldsymbol{D}(t) \\
&= (\boldsymbol{I} - \eta \boldsymbol{M}(t)) \boldsymbol{R}(t) - (\boldsymbol{v}_1(t+1)\boldsymbol{v}_1(t+1)^\top)(\boldsymbol{I} - \eta \boldsymbol{M}(t))(\boldsymbol{I} - \boldsymbol{v}_1(t)\boldsymbol{v}_1(t)^\top) \boldsymbol{D}(t) \\
&\quad + (\boldsymbol{I} - \boldsymbol{v}_1(t+1)\boldsymbol{v}_1(t+1)^\top)(\boldsymbol{I} - \eta \boldsymbol{M}(t))(\boldsymbol{v}_1(t)\boldsymbol{v}_1(t)^\top) \boldsymbol{D}(t) \\
&= (\boldsymbol{I} - \eta \boldsymbol{M}(t)) \boldsymbol{R}(t) - (1 - \eta \Lambda(t))(\boldsymbol{v}_1(t+1)\boldsymbol{v}_1(t+1)^\top)(\boldsymbol{I} - \boldsymbol{v}_1(t)\boldsymbol{v}_1(t)^\top) \boldsymbol{D}(t) \\
&\quad + (1 - \eta \Lambda(t))(\boldsymbol{I} - \boldsymbol{v}_1(t+1)\boldsymbol{v}_1(t+1)^\top)(\boldsymbol{v}_1(t)\boldsymbol{v}_1(t)^\top) \boldsymbol{D}(t)
\end{aligned}
$$

Now if we decompose $\boldsymbol{v}_1(t+1) = (1 - \delta(t))\boldsymbol{v}_1(t) + \sqrt{1 - (1-\delta)^2} \boldsymbol{v}_\perp(t)$, where $\langle \boldsymbol{v}_1(t), \boldsymbol{v}_\perp \rangle = 0$. Then we have

$$
\|(\boldsymbol{I} - \boldsymbol{v}_1(t+1)\boldsymbol{v}_1(t+1)^\top) \boldsymbol{v}_1(t) \boldsymbol{v}_1(t)^\top\|
$$

$$= \|\boldsymbol{v}_1(t) - (\boldsymbol{v}_1(t)^\top \boldsymbol{v}_1(t+1))\boldsymbol{v}_1(t+1)\|$$
$$= \|(1 - (1 - \delta(t))^2)\boldsymbol{v}_1(t) + \sqrt{1 - (1 - \delta(t)^2)}\boldsymbol{v}_\perp(t)\|$$
$$\leq \sqrt{(2\delta(t) - \delta(t)^2)(1 - \delta(t)) + (2\delta(t) - \delta(t)^2)^2}$$
$$\leq \sqrt{2\delta(t) + 4\delta(t)^2}$$
$$\leq 3\sqrt{\delta(t)}$$
$$\leq 3\sqrt{\epsilon_2}.$$

Similar result can be obtained for the term $(\boldsymbol{v}_1(t+1)\boldsymbol{v}_1(t+1)^\top)(\boldsymbol{I} - \boldsymbol{v}_1(t)\boldsymbol{v}_1(t)^\top)$.

Therefore, $\|(\eta\Lambda(t) - 1)(\boldsymbol{I} - \boldsymbol{v}_1(t+1)\boldsymbol{v}_1(t+1)^\top)(\boldsymbol{v}_1(t)\boldsymbol{v}_1(t)^\top)\boldsymbol{D}(t)\| \leq 6\sqrt{\epsilon_2}(B_\Lambda - 1)B_D.$ $\quad\square$

**Lemma E.5.** *(Lemma 3.5) Define an auxiliary sequence $\boldsymbol{R}'(t)$ by $\boldsymbol{R}'(0) = \boldsymbol{R}(0)$, and $\boldsymbol{R}'(t+1) = (\boldsymbol{I} - \eta\boldsymbol{M}(t)(\boldsymbol{I} - \boldsymbol{v}_1(t)\boldsymbol{v}_1(t)^\top))\boldsymbol{R}'(t)$. If Assumption 3.4, Assumption 3.5 (iii), Assumption 3.7 hold, and for any time $t$ there exists a quantity $\lambda_r > 0$, such that the smallest eigenvalue of $\boldsymbol{M}(t)$, i.e. $\lambda_{\min}(\boldsymbol{M}(t)) > \lambda_r$, then there exists a constant $c_r > 0$ such that $\|\boldsymbol{R}(t) - \boldsymbol{R}'(t)\| \leq c_r \frac{B_D(B_\Lambda - 1)\sqrt{\epsilon_2}}{\eta\lambda_r}$.*

*Proof.* First we can see that the eigenvalues of $\boldsymbol{M}(t)(\boldsymbol{I} - \boldsymbol{v}_1(t)\boldsymbol{v}_1(t)^\top)$ are all the eigenvalues of $\boldsymbol{M}$ except the largest one. Hence with Assumption 3.7, all eigenvalues of $\boldsymbol{M}(t)(\boldsymbol{I} - \boldsymbol{v}_1(t)\boldsymbol{v}_1(t)^\top)$ are in $(\lambda_r, 1/\eta)$. Thus $\|\boldsymbol{I} - \eta\boldsymbol{M}(t)(\boldsymbol{I} - \boldsymbol{v}_1(t)\boldsymbol{v}_1(t)^\top)\| \leq 1 - \eta\lambda_r.$

Hence by Assumption 3.4 we can get
$$\|\boldsymbol{R}(t+1) - \boldsymbol{R}'(t+1)\|$$
$$= \|(\boldsymbol{I} - \eta\boldsymbol{M}(t))\boldsymbol{R}(t) - (\boldsymbol{I} - \eta\boldsymbol{M}(t)(\boldsymbol{I} - \boldsymbol{v}_1(t)\boldsymbol{v}_1(t)^\top))\boldsymbol{R}'(t) + \boldsymbol{e}_1(t)\|$$
$$= \|(\boldsymbol{I} - \eta\boldsymbol{M}(t)(\boldsymbol{I} - \boldsymbol{v}_1(t)\boldsymbol{v}_1(t)^\top))(\boldsymbol{R}(t) - \boldsymbol{R}'(t)) + \boldsymbol{e}_1(t)\|$$
$$\leq \|(\boldsymbol{I} - \eta\boldsymbol{M}(t)(\boldsymbol{I} - \boldsymbol{v}_1(t)\boldsymbol{v}_1(t)^\top))(\boldsymbol{R}(t) - \boldsymbol{R}'(t))\| + \|\boldsymbol{e}_1(t)\|$$
$$\leq \|\boldsymbol{I} - \eta\boldsymbol{M}(t)(\boldsymbol{I} - \boldsymbol{v}_1(t)\boldsymbol{v}_1(t)^\top)\|\|\boldsymbol{R}(t) - \boldsymbol{R}'(t)\| + \|\boldsymbol{e}_1(t)\|$$
$$\leq \|\boldsymbol{R}(t) - \boldsymbol{R}'(t)\|(1 - \eta\lambda_r) + 3B_D(B_\Lambda - 1)\sqrt{\epsilon_2}.$$

Thus if we denote $\|\boldsymbol{R}(t) - \boldsymbol{R}'(t)\| - \frac{3B_D(B_\Lambda - 1)\sqrt{\epsilon_2}}{\eta\lambda_r}$ by $p(t)$, and replace $\|\boldsymbol{R}(t) - \boldsymbol{R}'(t)\|$ in the inequality above by $p(t) + \frac{3B_D(B_\Lambda - 1)\sqrt{\epsilon_2}}{\eta\lambda_r}$, we can get $|p(t+1)| \leq |p(t)|(1 - \eta\lambda_r)$. Hence, we can see that $|p(t)| < |p(0)| = \frac{3B_D(B_\Lambda - 1)\sqrt{\epsilon_2}}{\eta\lambda_r}$ for any time $t$. Therefore, we obtain that
$$\|\boldsymbol{R}(t) - \boldsymbol{R}'(t)\| < \frac{3B_D(B_\Lambda - 1)\sqrt{\epsilon_2}}{\eta\lambda_r} + |p(0)| < \frac{6B_D(B_\Lambda - 1)\sqrt{\epsilon_2}}{\eta\lambda_r}.$$

Taking $c_r = 6$, we finish the proof. $\quad\square$

**Lemma E.6.** *For all $t$ in Phase I, under Assumption 3.5 (i) and Assumption 3.7, it holds that $\boldsymbol{R}'(t)^\top(\boldsymbol{R}'(t) + \boldsymbol{Y}) < 0$ where $\boldsymbol{R}'(t)$ is defined in Lemma 3.5.*

*Proof.* Consider the inner product $\boldsymbol{R}'(t)^\top \boldsymbol{v}_i$. Recall $\boldsymbol{v}_i$ is the $i$-th eigenvector of $\boldsymbol{M}$. By Assumption 3.5 (i), $\boldsymbol{v}_i$ does not change with $t$.

By Assumption 3.7, $\lambda_i(t) < 1/\eta$ for $i > 1$, where $\lambda_i(t) \geq 0$ is the corresponding eigenvalue of $\boldsymbol{v}_i$.

By definition of $\boldsymbol{R}'(t)$ and Assumption 3.5 (i), $\boldsymbol{R}'(t+1) = (\boldsymbol{I} - \eta\boldsymbol{M}(t)(\boldsymbol{I} - \boldsymbol{v}_1\boldsymbol{v}_1^\top))\boldsymbol{R}'(t)$, hence $\boldsymbol{R}'(t+1)^\top \boldsymbol{v}_i = (1 - \eta\lambda_i(t))\boldsymbol{R}'(t)^\top \boldsymbol{v}_i$ for $i > 1$ and $\boldsymbol{R}'(t)^\top \boldsymbol{v}_1 = 0$. Hence for any $i > 1, t > 0$, $\frac{\boldsymbol{R}'(t)^\top \boldsymbol{v}_i}{\boldsymbol{R}'(0)^\top \boldsymbol{v}_i} = \prod_{j=0}^{t-1}(1 - \eta\lambda_i(j)) \in (0, 1)$.

Since $\boldsymbol{R}'(0) = \boldsymbol{R}(0) = (\boldsymbol{I} - \boldsymbol{v}_1\boldsymbol{v}_1^\top)\boldsymbol{Y}$, hence $\boldsymbol{R}'(0)^\top \boldsymbol{v}_i = -\boldsymbol{Y}^\top \boldsymbol{v}_i$ for any $i > 1$.

Plug it into the expression $\boldsymbol{R}'(t)^\top(\boldsymbol{R}'(t) + \boldsymbol{Y})$ and we have:
$$\boldsymbol{D}(t)^\top \boldsymbol{F}(t) = \sum_{i=2}^n \boldsymbol{R}'(t)^\top \boldsymbol{v}_i(\boldsymbol{R}'(t)^\top \boldsymbol{v}_i + \boldsymbol{Y}^\top \boldsymbol{v}_i) = \sum_{i=2}^n (\boldsymbol{R}'(t)^\top \boldsymbol{v}_i)^2(1 - \frac{\boldsymbol{R}'(0)^\top \boldsymbol{v}_i}{\boldsymbol{R}'(t)^\top \boldsymbol{v}_i}) < 0$$

$\square$

## E.2 Progressive Sharpening under Weaker Assumptions

We prove Lemma 3.1 in Section 3, i.e. progressive sharpening happens under Assumption 3.5 (i) that the set of eigenvectors of the Gram matrix $\boldsymbol{M}(t)$ is fixed throughout. However, empirically, the eigenvectors of $\boldsymbol{M}(t)$ change non-negligibly (see Figure 26). In this section, we show that it is possible to relax Assumption 3.5 (i) to more realistic assumption on the eigenspace of $\boldsymbol{M}(t)$. To this end, we first present a dynamical system view of the dynamics of $\boldsymbol{D}(t)$, then prove Lemma E.7 which is analogue of Lemma 3.1 but under weaker assumptions.

### E.2.1 A dynamical system view

Recall that we assume that $\boldsymbol{D}(t)$ follows the gradient flow trajectory: $\frac{\mathrm{d}\boldsymbol{D}(t)}{\mathrm{d}t} = -\boldsymbol{M}(t)\boldsymbol{D}(t)$. This is a linear dynamical system with changing coefficients $-\boldsymbol{M}(t)$. In order to understand this dynamics, we first consider the linear dynamical system with constant coefficients $-\boldsymbol{M}$, $\frac{\mathrm{d}\boldsymbol{D}(t)}{\mathrm{d}t} = -\boldsymbol{M}\boldsymbol{D}(t)$, which is much better understood. The corresponding phase portrait is shown in Figure 25(a). Here, $O$ is only fixed point. In this figure, we consider the ball $\mathbb{B} \subset \mathbb{R}^n$ such that the south pole of $\mathbb{B}$ is the origin $O$ and the north pole $N$ of $\mathbb{B}$ is fixed to be $-\boldsymbol{Y}$, which is the initial point $-\boldsymbol{D}(t)$. Recall that $\boldsymbol{F}(t) = \boldsymbol{D}(t) + \boldsymbol{Y}$ and $\boldsymbol{D}(0) = -\boldsymbol{Y}$. We denote the tip of the vector $\boldsymbol{D}(t)$ as $D$. Hence, $\overrightarrow{DO} = \boldsymbol{D}(t)$ and $\overrightarrow{ND} = \boldsymbol{F}(t)$. A useful geometric observation is the following:

**Observation E.1.** $\boldsymbol{D}(t) \in \mathbb{B}$ if and only if $\boldsymbol{D}(t)^\top \boldsymbol{F}(t) = \boldsymbol{D}(t)^\top (\boldsymbol{D}(t) - (-\boldsymbol{Y})) < 0$ *(equivalently, the angle $\angle(NDO) \geq \pi/2$).*

Let $\{\boldsymbol{v}_i(t)\}_i$ be the set of eigenvectors of $\boldsymbol{M}(t)$. By symmetry, we can assume the direction of all $\{\boldsymbol{v}_i(t)\}_i$ are pointing inside $\mathbb{B}$ (or equivalently, $-\boldsymbol{v}_i(t)^\top \boldsymbol{Y} \geq 0$). We also define the hypercube $\mathbb{H}(t)$, which is defined by $\{\boldsymbol{v}_i(t)\}_i$ as the edges and $ON$ as the diagonal (see the dashed rectangle in Figure 25(a). All vertices of $\mathbb{H}(t)$ are on the sphere $\partial\mathbb{B}$. In our proof of Lemma 3.1 (in which we assume the eigenvectors do not change), we have shown the entire trajectory of $\boldsymbol{D}(t)$ is contained in $\mathbb{H}(t)$ which is always contained in $\mathbb{B}$. In particular, we prove that the projection of $\boldsymbol{D}(t)$ onto each $\boldsymbol{v}_i$ decreases monotonically).

Now, we consider the more general case when the eigenvectors $\{\boldsymbol{v}_i(t)\}_i$ may change. In Lemma E.7 presented below, we show that if the eigenvectors change directions relatively slowly (precisely the change rate satisfies (20)), $\boldsymbol{D}(t)$ is still contained in $\mathbb{H}(t)$, hence also in $\mathbb{B}$. Then by Observation E.1 and (3), the norm $\|A\|$ increases (hence the sharpness increases).

We also note that it is impossible to prove $\boldsymbol{D}(t) \in \mathbb{B}$ for all $t > 0$ without any assumption on the change of eigenvector directions. In particular, if the eigenvector changes fast enough so that $\boldsymbol{D}(t)$ is out of the hypercube $\mathbb{H}(t)$, the trajectory may eventually go out of the ball $\mathbb{B}$ (e.g., the region inside the dashed red box in Figure 25(b)). However, it seems that this region is fairly close to the origin $O$, and the trajectory does not go very far away from $\mathbb{B}$ (implying $\boldsymbol{D}(t)^\top \boldsymbol{F}(t)$ is not a large positive number). Hence, the above geometric intuition suggests that it is possible to further relax the assumption (inequality (20)) made in E.7 and prove that A-norm increases (hence sharpness increases) until the sharpness reaches close to $2/\eta$, after which gradient flow is not a good approximation of gradient descent anymore. We leave it as an interesting future direction.

### E.2.2 Progressive sharpening when eigendirections change slowly

Suppose at initialization $\boldsymbol{F}(0) = 0$. By $\boldsymbol{D}(t) = \boldsymbol{F}(t) - Y$, we know $\boldsymbol{D}(0) = -Y$. With loss of generality, we assume $\boldsymbol{D}(0)^\top \boldsymbol{v}_i(0) > 0$ at initialization for all $i$ (Otherwise, we define $\boldsymbol{v}_i(0) = -\boldsymbol{v}_i(0)$ and this condition holds).

**Lemma E.7.** *Suppose Assumption 3.3 holds, and the set of (unit) eigenvectors of $\boldsymbol{M}(t)$ is $\{\boldsymbol{v}_i(t)\}$. Assume at all time $t$ and direction $\boldsymbol{v}_i(t)$, the following condition holds:*

$$\boldsymbol{F}(t)^\top \frac{\mathrm{d}\boldsymbol{v}_i(t)}{\mathrm{d}t} < \lambda_i(t)\boldsymbol{D}(t)^\top \boldsymbol{v}_i(t). \tag{20}$$

*Then we have $\boldsymbol{D}(t)^\top \boldsymbol{F}(t) < 0$ for all $t < t_1$, where $t_1$ is the first time when there exists some $i$, $\boldsymbol{D}(t)^\top \boldsymbol{v}_i(t) \leq 0$.*

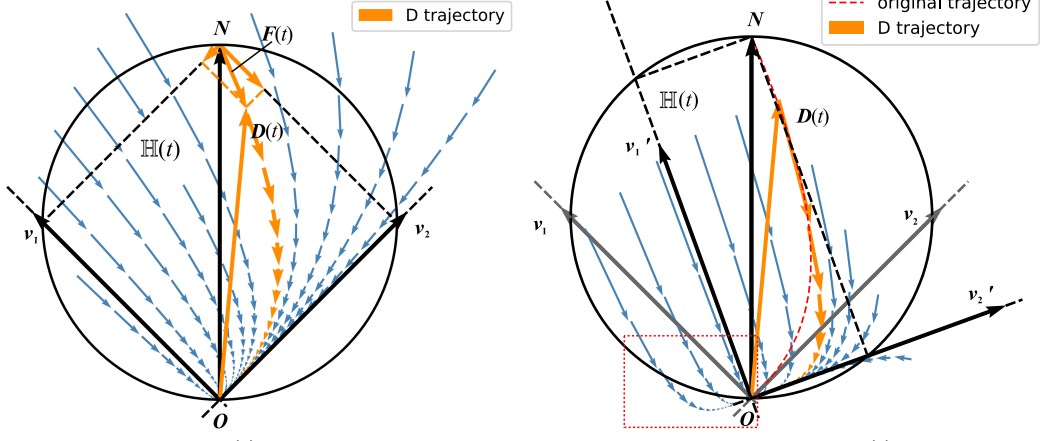

(a) The dynamics of $\boldsymbol{D}(t)$ with constant coefficients.  (b) The eigenvectors of $\boldsymbol{M}(t)$ change

Figure 25: The phase portraits of the dynamics of $\boldsymbol{D}(t)$ (projected to the first two principle directions). We rotate the coordinate system so that south pole of $\mathbb{B}$ is the origin $O$ and the north pole is $-\boldsymbol{Y}$ which is the initial point $\boldsymbol{D}(t)$. $\boldsymbol{v}_1$ and $\boldsymbol{v}_2$ are the first two eigenvectors of $\boldsymbol{M}$. In (b), $\boldsymbol{v}_1$ and $\boldsymbol{v}_2$ change directions to $\boldsymbol{v}_1'$ and $\boldsymbol{v}_2'$. The trajectory may go out of $\mathbb{B}$ from certain regions (the dashed red box).

In the above lemma, we can see that inequality (20) holds if $\boldsymbol{v}_i(t)$ changes relatively slowly. In particular, if $\frac{d\boldsymbol{v}_i(t)}{dt} = 0$ for all $i$, the lemma reduces to Lemma 3.1. Note that in contrast to the fixed eigenvector assumption, we cannot show that $\boldsymbol{D}(t)^\top \boldsymbol{v}_i(t)$ decreases in all directions all the time (since $\boldsymbol{M}(t)$ may change). Instead, we show that $\boldsymbol{D}(t)$ is always in the hypercube $\mathbb{H}(t)$ under the condition of the lemma.

*Proof.* We first notice the decomposition

$$\boldsymbol{D}(t)^\top \boldsymbol{F}(t) = \sum_{i=1}^{r} \boldsymbol{D}(t)^\top \boldsymbol{v}_i(t) \boldsymbol{F}(t)^\top \boldsymbol{v}_i(t).$$

We show for each $\boldsymbol{v}_i(t)$, $\boldsymbol{D}(t)^\top \boldsymbol{v}_i(t) \boldsymbol{F}(t)^\top \boldsymbol{v}_i(t) < 0$ under the conditions of the lemma.

We first consider the dynamics of inner product $\boldsymbol{D}(t)^\top \boldsymbol{v}_i(t)$. According to Assumption 3.3, when $t < t_1$, $\boldsymbol{D}(t)^\top \boldsymbol{v}_i(t) > 0$ due to the continuity. Moreover, by the dynamics of $\boldsymbol{D}(t)$ in Assumption 3.3, we can see

$$\frac{d\boldsymbol{D}(t)^\top \boldsymbol{v}_i(t)}{dt} = -\boldsymbol{D}(t)^\top \boldsymbol{M}(t)\boldsymbol{v}_i(t) + \boldsymbol{D}(t)^\top \frac{d\boldsymbol{v}_i(t)}{dt}$$

$$= -\lambda_i(t)\boldsymbol{D}(t)^\top \boldsymbol{v}_i(t) + \boldsymbol{D}(t)^\top \frac{d\boldsymbol{v}_i(t)}{dt}$$

With this dynamics above and $\boldsymbol{F}(t) = \boldsymbol{D}(t) + \boldsymbol{Y}$, we can get $\boldsymbol{F}$'s dynamics:

$$\frac{d\boldsymbol{F}(t)^\top \boldsymbol{v}_i(t)}{dt} = -\boldsymbol{D}(t)^\top \boldsymbol{M}(t)\boldsymbol{v}_i(t) + \boldsymbol{F}(t)^\top \frac{d\boldsymbol{v}_i(t)}{dt}$$

$$= -\lambda_i(t)\boldsymbol{D}(t)^\top \boldsymbol{v}_i(t) + \boldsymbol{F}(t)^\top \frac{d\boldsymbol{v}_i(t)}{dt} \qquad (D1)$$

With (D1) and $\boldsymbol{F}(t)^\top \frac{d\boldsymbol{v}_i(t)}{dt} \le \lambda_i(t)\boldsymbol{D}(t)^\top \boldsymbol{v}_i(t)$, we know

$$\frac{d\boldsymbol{F}(t)^\top \boldsymbol{v}_i(t)}{dt} = -\lambda_i(t)\boldsymbol{D}(t)^\top \boldsymbol{v}_i(t) + \boldsymbol{F}(t)^\top \frac{d\boldsymbol{v}_i(t)}{dt} < 0.$$

In this way, we know $\boldsymbol{F}(t)^\top \boldsymbol{v}_i(t)$ always decreases for $t < t_1$. Recall that $\boldsymbol{F}(0) = \boldsymbol{D}(0) + \boldsymbol{Y} = 0$, which makes $\boldsymbol{F}(0)^\top \boldsymbol{v}_i(0) = 0$. So $\boldsymbol{F}(t)^\top \boldsymbol{v}_i(t) < 0$ according to the dynamics. Geometrically, this implies $\boldsymbol{D}(t)$ does not cross the facets of $\mathbb{H}(t)$ incident on the north pole $N$ [6] (otherwise $\boldsymbol{F}(t)^\top \boldsymbol{v}_i(t)$

---

[6]The hypercube $\mathbb{H}(t)$ has $2n$ facets, $n$ of them incident on the south pole $O$ and the others the north pole $N$.

needs to change sign). See Figure 25). On the other hand, $\boldsymbol{D}(t)^\top \boldsymbol{v}_i(t) > 0$ for $t < t_1$, by the definition of $t_1$, which is equivalent to say that $\boldsymbol{D}(t)^\top \boldsymbol{v}_i(t)$ does not change sign or $\boldsymbol{D}(t)$ does not cross the facets of $\mathbb{H}(t)$ incident on the south pole $O$.

Therefore we have $\boldsymbol{D}(t)^\top \boldsymbol{v}_i(t)\boldsymbol{F}(t)^\top \boldsymbol{v}_i(t) < 0$ and $\boldsymbol{D}(t) \in \mathbb{H}(t) \subset \mathbb{B}$ for all $t < t_1$, where $t_1$ is the first time when $\boldsymbol{D}(t_1)^\top \boldsymbol{v}_i(t_1) \leq 0$. Summing over all the components $\boldsymbol{D}(t)^\top \boldsymbol{v}_i(t)\boldsymbol{F}(t)^\top \boldsymbol{v}_i(t)$ for all $i$, we prove the lemma. □

### E.2.3 Empirical verification of inequality (20)

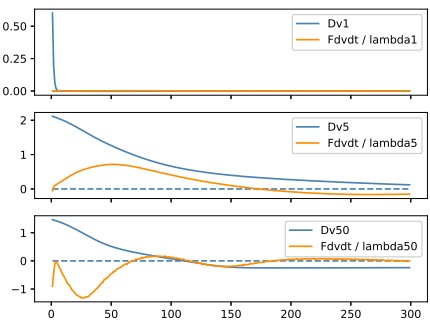

Figure 26: Verification for inequality (20) when $i = 1, 5, 50$.

We verify the condition (20) in Lemma E.7 via experiments. See Figure 26. We can see that (20) holds for $i = 1, 5, 50$ during the first hundred iterations of the training process. However, for very large $i$, the eigenvectors corresponding to very small eigenvalues may change quickly and (20) does not hold. The contributions (i.e., $\boldsymbol{D}(t)^\top \boldsymbol{v}_i(t)\boldsymbol{F}(t)^\top \boldsymbol{v}_i(t)$) from those corresponding to very small eigenvalues tend to cancel each other empirically, hence does not affect the sign of the total sum $\sum_{i=1}^{r} \boldsymbol{D}(t)^\top \boldsymbol{v}_i(t)\boldsymbol{F}(t)^\top \boldsymbol{v}_i(t)$. Hence, it is possible to further relax the condition (20) and we leave it as a future direction.

### E.3 Proofs of equations in Appendix C

The proof of some equations in Appendix C is omitted, and we list them in this subsection.

**Lemma E.8.** *(Lemma C.1) The update rule of the residual vector of $\boldsymbol{D}(t)$:*

$$\boldsymbol{D}(t+1)^\top - \boldsymbol{D}(t)^\top = -\eta \boldsymbol{D}(t)^\top \boldsymbol{M}(t) + \frac{4\eta^2}{n^2 m}\boldsymbol{D}(t)^\top (\boldsymbol{F}(t)^\top \boldsymbol{D}(t))\mathbf{X}^\top \mathbf{X} \qquad (21)$$

*Proof.*

$$
\begin{aligned}
\boldsymbol{D}(t+1)^\top - \boldsymbol{D}(t)^\top &= \frac{1}{\sqrt{m}}(\boldsymbol{A}(t+1)^\top \boldsymbol{W}(t+1) - \boldsymbol{A}(t)^\top \boldsymbol{W}(t))\mathbf{X} \\
&= \frac{1}{\sqrt{m}}\left((\boldsymbol{A}(t+1)^\top - \boldsymbol{A}(t)^\top)\boldsymbol{W}(t+1) + \boldsymbol{A}(t)^\top (\boldsymbol{W}(t+1) - \boldsymbol{W}(t))\right)\mathbf{X} \\
&= \frac{1}{\sqrt{m}}\left((\boldsymbol{A}(t+1)^\top - \boldsymbol{A}(t)^\top)\boldsymbol{W}(t) + \boldsymbol{A}(t)^\top (\boldsymbol{W}(t+1) - \boldsymbol{W}(t))\right)\mathbf{X} \\
&\quad + \frac{1}{\sqrt{m}}(\boldsymbol{A}(t+1)^\top - \boldsymbol{A}(t)^\top)(\boldsymbol{W}(t+1) - \boldsymbol{W}(t))\mathbf{X} \\
&= \frac{2\eta}{nm}\left((-\boldsymbol{W}(t)\mathbf{X}\boldsymbol{D}(t))^\top \boldsymbol{W}(t) - \boldsymbol{A}(t)^\top \boldsymbol{A}(t)\boldsymbol{D}(t)^\top \mathbf{X}^\top\right)\mathbf{X} \\
&\quad + \frac{1}{\sqrt{m}}(\frac{2\eta}{n\sqrt{m}}\boldsymbol{W}(t)\mathbf{X}\boldsymbol{D}(t))^\top \frac{2\eta}{n\sqrt{m}}\boldsymbol{A}(t)\boldsymbol{D}(t)^\top \mathbf{X}^\top \mathbf{X}
\end{aligned}
$$

$$= -\boldsymbol{D}(t)^\top \frac{2\eta}{nm}(\mathbf{X}^\top \boldsymbol{W}(t)^\top \boldsymbol{W}(t)\mathbf{X} + \|\boldsymbol{A}(t)\|^2 \mathbf{X}^\top \mathbf{X})$$

$$+ \frac{4\eta^2}{n^2 m}\boldsymbol{D}(t)^\top(\boldsymbol{F}(t)^\top \boldsymbol{D}(t))\mathbf{X}^\top \mathbf{X}$$

$$= -\eta \boldsymbol{D}(t)^\top \boldsymbol{M}(t) + \frac{4\eta^2}{n^2 m}\boldsymbol{D}(t)^\top(\boldsymbol{F}(t)^\top \boldsymbol{D}(t))\mathbf{X}^\top \mathbf{X}$$

$\square$

**Lemma E.9.** *(Lemma C.2) The update rule of the Gram matrix $\boldsymbol{M}(t)$ is,*

$$\boldsymbol{M}(t+1) - \boldsymbol{M}(t) = -\frac{4\eta}{n^2 m}\left(2(\boldsymbol{F}(t)^\top \boldsymbol{D}(t))\mathbf{X}^\top \mathbf{X} + \boldsymbol{F}(t)\boldsymbol{D}(t)^\top \mathbf{X}^\top \mathbf{X} + \mathbf{X}^\top \mathbf{X}\boldsymbol{D}(t)\boldsymbol{F}(t)^\top\right)$$

$$+ \frac{8\eta^2}{n^3 m^2}(\boldsymbol{D}(t)^\top \mathbf{X}^\top \boldsymbol{W}(t)^\top \boldsymbol{W}(t)\mathbf{X}\boldsymbol{D}(t))\mathbf{X}^\top \mathbf{X}$$

$$+ \frac{8\eta^2}{n^3 m^2}\|\boldsymbol{A}(t)\|^2 \mathbf{X}^\top \mathbf{X}\boldsymbol{D}(t)\boldsymbol{D}(t)^\top \mathbf{X}^\top \mathbf{X}$$

*Proof.* By the update rule of $\boldsymbol{A}(t)$ and $\boldsymbol{W}(t)$ in equation (6) and (7), we have

$$\boldsymbol{M}(t+1) - \boldsymbol{M}(t)$$

$$= \frac{2}{nm}\left((\|\boldsymbol{A}(t+1)\|^2 - \|\boldsymbol{A}(t)\|^2)\mathbf{X}^\top \mathbf{X} + \mathbf{X}^\top(\boldsymbol{W}(t+1)^\top \boldsymbol{W}(t+1) - \boldsymbol{W}(t)^\top \boldsymbol{W}(t))\mathbf{X}\right)$$

(Use equation (7))

$$= \frac{2}{nm}\left(-\frac{4\eta}{n}\boldsymbol{D}(t)^\top \boldsymbol{F}(t) + \eta^2\|\frac{\partial \mathcal{L}}{\partial \boldsymbol{A}}\|^2\right)\mathbf{X}^\top \mathbf{X}$$

$$+ \frac{2}{nm}\left(\mathbf{X}^\top((\boldsymbol{W}(t+1)^\top - \boldsymbol{W}(t)^\top)\boldsymbol{W}(t) + \boldsymbol{W}(t)^\top(\boldsymbol{W}(t+1) - \boldsymbol{W}(t)))\mathbf{X}\right.$$

$$\left. + \mathbf{X}^\top(\boldsymbol{W}(t+1)^\top - \boldsymbol{W}(t)^\top)(\boldsymbol{W}(t+1) - \boldsymbol{W}(t))\mathbf{X}\right) \qquad \text{(Use equation (6))}$$

$$= -\frac{8\eta}{n^2 m}\boldsymbol{D}(t)^\top \boldsymbol{F}(t)\mathbf{X}^\top \mathbf{X} + \frac{8}{n^3 m^2}\eta^2(\boldsymbol{D}(t)^\top \mathbf{X}^\top \boldsymbol{W}(t)^\top \boldsymbol{W}(t)\mathbf{X}\boldsymbol{D}(t))\mathbf{X}^\top \mathbf{X}$$

$$- \frac{4\eta}{n^2 m}\left(\boldsymbol{F}(t)\boldsymbol{D}(t)^\top \mathbf{X}^\top \mathbf{X} + \mathbf{X}^\top \mathbf{X}\boldsymbol{D}(t)\boldsymbol{F}(t)^\top\right)$$

$$+ \frac{8\eta^2}{n^3 m^2}\|\boldsymbol{A}(t)\|^2 \mathbf{X}^\top \mathbf{X}\boldsymbol{D}(t)\boldsymbol{D}(t)^\top \mathbf{X}^\top \mathbf{X}$$

Rearrange the terms and we complete the proof. $\square$

**Lemma E.10.** *For all iteration $t$, if any vector $\boldsymbol{u} \in \mathbb{R}^n$ satisfies $\mathbf{X}^\top \mathbf{X}\boldsymbol{u} = 0$, then*

$$\boldsymbol{Y}^\top \boldsymbol{u} = \boldsymbol{D}(t)^\top \boldsymbol{u} = 0$$

*Proof.* If $\mathbf{X}^\top \mathbf{X}\boldsymbol{u} = 0$, then

$$\boldsymbol{u}^\top \mathbf{X}^\top \mathbf{X}\boldsymbol{u} = 0 \Rightarrow \mathbf{X}\boldsymbol{u} = 0.$$

So we have

$$\boldsymbol{Y}^\top \boldsymbol{u} = -\boldsymbol{D}(0)^\top \boldsymbol{u} = \boldsymbol{A}^{*\top}\boldsymbol{W}^*\mathbf{X}\boldsymbol{u} = 0$$

With the dynamics of $\boldsymbol{D}(t)$, we have

$$\boldsymbol{D}(t+1)^\top \boldsymbol{u} = \boldsymbol{D}(t)^\top(\boldsymbol{I}_n - \eta \boldsymbol{M}^*(t))\boldsymbol{u}$$

While

$$\boldsymbol{M}^*(t)\boldsymbol{u} = \boldsymbol{M}(t)\boldsymbol{u} - \frac{4\eta}{n^2 m}(\boldsymbol{D}(t)^\top \boldsymbol{F}(t))\mathbf{X}^\top \mathbf{X}\boldsymbol{u} = \boldsymbol{M}(t)\boldsymbol{u}$$

$$= \frac{2}{mn}(\|\boldsymbol{A}(t)\|^2 \mathbf{X}^\top \mathbf{X} + \mathbf{X}^\top \boldsymbol{W}^\top(t)\boldsymbol{W}(t)\mathbf{X})\boldsymbol{u} = 0$$

Thus we have

$$\boldsymbol{D}(t+1)^\top \boldsymbol{u} = \boldsymbol{D}(t)^\top (\boldsymbol{I}_n - \eta \boldsymbol{M}^*(t))\boldsymbol{u} = \boldsymbol{D}(t)^\top \boldsymbol{u} = ... = \boldsymbol{D}(0)^\top \boldsymbol{u} = 0$$

$\square$

**Lemma E.11.** *(Lemma C.7) Here we give the proof of the equation* (17)*:*

$$\Lambda^*(t+1) - \Lambda^*(t) = -\frac{8\eta\lambda_1}{mn^2}\left(\boldsymbol{F}(t)^\top \boldsymbol{D}(t) + (\boldsymbol{F}(t)^\top \boldsymbol{v}_1)(\boldsymbol{D}(t)^\top \boldsymbol{v}_1) - \frac{\eta}{2}(\boldsymbol{D}(t)^\top \boldsymbol{v}_1)^2 \Lambda^*(t)\right.$$
$$\left. -\frac{\eta}{2}\boldsymbol{R}(t)^\top \boldsymbol{\Gamma}(t)\boldsymbol{R}(t) - \eta\boldsymbol{R}(t)^\top \boldsymbol{\Gamma}(t)\left(\boldsymbol{v}_1\boldsymbol{v}_1^\top \boldsymbol{D}(t)\right) - \frac{\eta}{mn}\boldsymbol{R}(t)^\top \left(\frac{m}{d}\mathbf{X}^\top \mathbf{X}\right)\boldsymbol{R}(t)\right).$$

*Proof.* First, since $\boldsymbol{W}^\top \boldsymbol{W}$ is initialized as $\frac{m}{d}\mathbf{X}^\top \mathbf{X}$, by definition, we have

$\mathbf{X}^\top \boldsymbol{W}^\top \boldsymbol{W}\mathbf{X} = \frac{mn}{2}\boldsymbol{\Gamma}(t) + \frac{m}{d}\mathbf{X}^\top \mathbf{X}$.

Then

$$\Lambda^*(t+1) - \Lambda^*(t)$$
$$= \boldsymbol{v}_1^\top \boldsymbol{M}(t+1)\boldsymbol{v}_1 - \boldsymbol{v}_1^\top \boldsymbol{M}(t)\boldsymbol{v}_1$$
$$= \boldsymbol{v}_1^\top (\boldsymbol{M}(t+1) - \boldsymbol{M}(t))\boldsymbol{v}_1$$
$$= -\frac{8\eta}{n^2 m}\left(\lambda_1 \boldsymbol{F}(t)^\top \boldsymbol{D}(t) + \lambda_1(\boldsymbol{F}(t)^\top \boldsymbol{v}_1)(\boldsymbol{D}(t)^\top \boldsymbol{v}_1)\right)$$
$$+ \frac{8\eta^2}{n^3 m^2}\left(\lambda_1 \boldsymbol{D}(t)^\top \mathbf{X}^\top \boldsymbol{W}^\top \boldsymbol{W}\mathbf{X}\boldsymbol{D}(t) + \|\boldsymbol{A}\|^2 \lambda_1^2 (\boldsymbol{D}(t)^\top \boldsymbol{v}_1)^2\right)$$

Because

$$\boldsymbol{D}(t)^\top \mathbf{X}^\top \boldsymbol{W}^\top \boldsymbol{W}\mathbf{X}\boldsymbol{D}(t)$$
$$= \boldsymbol{D}(t)^\top \left(\frac{m}{d}\mathbf{X}^\top \mathbf{X} + \frac{mn}{2}\boldsymbol{\Gamma}(t)\right)\boldsymbol{D}(t)$$
$$= \frac{m}{d}\boldsymbol{D}(t)^\top \left(\boldsymbol{v}_1\boldsymbol{v}_1^\top + \boldsymbol{I} - \boldsymbol{v}_1\boldsymbol{v}_1^\top\right)\mathbf{X}^\top \mathbf{X}\left(\boldsymbol{D}(t)^\top \left(\boldsymbol{v}_1\boldsymbol{v}_1^\top + \boldsymbol{I} - \boldsymbol{v}_1\boldsymbol{v}_1^\top\right)\right)^\top$$
$$+ \frac{mn}{2}\boldsymbol{D}(t)^\top \boldsymbol{\Gamma}(t)\boldsymbol{D}(t)$$
$$= \frac{m}{d}\lambda_1(\boldsymbol{D}(t)^\top \boldsymbol{v}_1)^2 + \frac{m}{d}\boldsymbol{R}(t)^\top \mathbf{X}^\top \mathbf{X}\boldsymbol{R}(t) + \frac{mn}{2}\boldsymbol{D}(t)^\top \boldsymbol{\Gamma}(t)\boldsymbol{D}(t)$$

and

$$\lambda_1\left(\frac{m}{d} + \|\boldsymbol{A}\|^2\right) = \frac{mn}{2}\boldsymbol{v}_1^\top (\boldsymbol{M}(t) - \boldsymbol{\Gamma}(t))\boldsymbol{v}_1 = \frac{mn}{2}\Lambda^*(t) - \frac{mn}{2}\boldsymbol{v}_1^\top \boldsymbol{\Gamma}(t)\boldsymbol{v}_1$$

we can see that

$$\Lambda^*(t+1) - \Lambda^*(t)$$
$$= -\frac{8\eta\lambda_1}{n^2 m}\left(\boldsymbol{F}(t)^\top \boldsymbol{D}(t) + (\boldsymbol{F}(t)^\top \boldsymbol{v}_1)(\boldsymbol{D}(t)^\top \boldsymbol{v}_1) - \frac{\eta}{mn}(\boldsymbol{D}(t)^\top \boldsymbol{v}_1)^2 \left(\frac{m}{d} + \|\boldsymbol{A}\|^2\right)\lambda_1.\right.$$
$$\left. -\frac{\eta}{mn}\cdot\frac{m}{d}\boldsymbol{R}(t)^\top \mathbf{X}^\top \mathbf{X}\boldsymbol{R}(t) - \frac{\eta}{2}\boldsymbol{D}(t)^\top \boldsymbol{\Gamma}(t)\boldsymbol{D}(t)\right)$$
$$= -\frac{8\eta\lambda_1}{mn^2}\left(\boldsymbol{F}(t)^\top \boldsymbol{D}(t) + (\boldsymbol{F}(t)^\top \boldsymbol{v}_1)(\boldsymbol{D}(t)^\top \boldsymbol{v}_1) - \frac{\eta}{2}(\boldsymbol{D}(t)^\top \boldsymbol{v}_1)^2 \Lambda^*(t)\right.$$
$$\left. + \frac{\eta}{2}\left(\boldsymbol{v}_1^\top \boldsymbol{\Gamma}(t)\boldsymbol{v}_1 \cdot (\boldsymbol{D}(t)^\top \boldsymbol{v}_1)^2 - \boldsymbol{D}(t)^\top \boldsymbol{\Gamma}(t)\boldsymbol{D}(t)\right) - \boldsymbol{R}(t)^\top \left(\frac{m}{d}\mathbf{X}^\top \mathbf{X}\right)\boldsymbol{R}(t)\right)$$

Note that

$$\boldsymbol{v}_1\boldsymbol{\Gamma}(t)\boldsymbol{v}_1^\top (\boldsymbol{D}(t)^\top \boldsymbol{v}_1)^2 - \boldsymbol{D}(t)^\top \boldsymbol{\Gamma}(t)\boldsymbol{D}(t)$$
$$= \boldsymbol{D}(t)^\top \boldsymbol{v}_1\boldsymbol{v}_1^\top \boldsymbol{\Gamma}(t)\boldsymbol{v}_1\boldsymbol{v}_1^\top \boldsymbol{D}(t) - \left(\boldsymbol{R}(t)^\top + \boldsymbol{D}(t)^\top \boldsymbol{v}_1\boldsymbol{v}_1^\top\right)\boldsymbol{\Gamma}(t)\left(\boldsymbol{R}(t) + \boldsymbol{v}_1\boldsymbol{v}_1^\top \boldsymbol{D}(t)\right)$$
$$= -\boldsymbol{R}(t)^\top \boldsymbol{\Gamma}(t)\boldsymbol{R}(t) - 2\boldsymbol{R}(t)^\top \boldsymbol{\Gamma}(t)\left(\boldsymbol{D}(t)^\top \boldsymbol{v}_1\boldsymbol{v}_1^\top\right)^\top$$

Now we finish the proof. $\square$

**Lemma E.12.** *(Equation 18)* $\boldsymbol{R}(t+1) = (I - \eta \boldsymbol{M}^*(t)) \boldsymbol{R}(t) + \eta \left( \boldsymbol{v}_1 \boldsymbol{v}_1^\top \boldsymbol{\Gamma}(t) - \boldsymbol{\Gamma}(t) \boldsymbol{v}_1 \boldsymbol{v}_1^\top \right) \boldsymbol{D}(t)$

*Proof.*

$$
\begin{aligned}
\boldsymbol{R}(t+1) &= \left( \boldsymbol{I} - \boldsymbol{v}_1 \boldsymbol{v}_1^\top \right) \boldsymbol{D}(t+1) \\
&= \left( \boldsymbol{I} - \boldsymbol{v}_1 \boldsymbol{v}_1^\top \right) \left( \boldsymbol{I} - \eta \boldsymbol{M}^*(t) \right) \boldsymbol{D}(t) \\
&= \left( \boldsymbol{I} - \boldsymbol{v}_1 \boldsymbol{v}_1^\top \right) \left( \boldsymbol{I} - \eta \boldsymbol{M}^*(t) \right) \boldsymbol{v}_1 \boldsymbol{v}_1^\top \boldsymbol{D}(t) \\
&\quad + \left( \boldsymbol{I} - \boldsymbol{v}_1 \boldsymbol{v}_1^\top \right) \left( \boldsymbol{I} - \eta \boldsymbol{M}^*(t) \right) \left( \boldsymbol{I} - \boldsymbol{v}_1 \boldsymbol{v}_1^\top \right) \boldsymbol{D}(t) \\
&= \left( \boldsymbol{I} - \eta \boldsymbol{M}^*(t) \right) \boldsymbol{R}(t) - \boldsymbol{v}_1 \boldsymbol{v}_1^\top \left( \boldsymbol{I} - \eta \boldsymbol{M}^*(t) \right) \left( \boldsymbol{I} - \boldsymbol{v}_1 \boldsymbol{v}_1^\top \right) \boldsymbol{D}(t) \\
&\quad + \left( \boldsymbol{I} - \boldsymbol{v}_1 \boldsymbol{v}_1^\top \right) \left( \boldsymbol{I} - \eta \boldsymbol{M}^*(t) \right) \boldsymbol{v}_1 \boldsymbol{v}_1^\top \boldsymbol{D}(t) \\
&= \left( \boldsymbol{I} - \eta \boldsymbol{M}^*(t) \right) \boldsymbol{R}(t) - \boldsymbol{v}_1 \boldsymbol{v}_1^\top \left( -\eta \boldsymbol{\Gamma}(t) \right) \left( \boldsymbol{I} - \boldsymbol{v}_1 \boldsymbol{v}_1^\top \right) \boldsymbol{D}(t) \\
&\quad + \left( \boldsymbol{I} - \boldsymbol{v}_1 \boldsymbol{v}_1^\top \right) \left( -\eta \boldsymbol{\Gamma}(t) \right) \boldsymbol{v}_1 \boldsymbol{v}_1^\top \boldsymbol{D}(t) \\
&= \left( \boldsymbol{I} - \eta \boldsymbol{M}^*(t) \right) \boldsymbol{R}(t) + \eta \left( \boldsymbol{v}_1 \boldsymbol{v}_1^\top \boldsymbol{\Gamma}(t) - \boldsymbol{\Gamma}(t) \boldsymbol{v}_1 \boldsymbol{v}_1^\top \right) \boldsymbol{D}(t)
\end{aligned}
$$

$\square$

# F   Missing Preliminaries

**Hessian, Fisher information matrix and NTK:** The Hessian of the objective $\mathcal{L}(f(\boldsymbol{\theta}))$ is:

$$
\nabla_{\boldsymbol{\theta}}^2 \mathcal{L}(f(\boldsymbol{\theta})) = \frac{1}{n} \sum_{i=1}^n \nabla_{\boldsymbol{\theta}}^2 \ell(f(\boldsymbol{\theta}, \mathbf{x}_i))
$$

We need the relationship between the Hessian matrix, the Fisher information matrix (FIM) and the neural tangent kernel (NTK). As shown in Papyan [26], Martens [23], Bottou et al. [4], the Hessian can be decomposed into two components:

$$
\nabla_{\boldsymbol{\theta}}^2 \mathcal{L}(f(\boldsymbol{\theta})) = \frac{1}{n} \sum_{i=1}^n \left( \frac{\partial f(\mathbf{x}_i, \boldsymbol{\theta})}{\partial \boldsymbol{\theta}}^\top \left. \frac{\partial^2 \ell(z, y_i)}{\partial z^2} \right|_{f(\mathbf{x}_i)} \frac{\partial f(\mathbf{x}_i, \boldsymbol{\theta})}{\partial \boldsymbol{\theta}} + \left. \frac{\partial \ell(z, y_i)}{\partial z} \right|_{f(\mathbf{x}_i)} \nabla_{\boldsymbol{\theta}}^2 f(\mathbf{x}_i, \boldsymbol{\theta}) \right)
$$

Here $\frac{\partial f(\mathbf{x_i}, \boldsymbol{\theta})}{\partial \boldsymbol{\theta}} \in \mathbb{R}^{1 \times p}$ is the gradient of $f$. Papyan [26] also demonstrate empirically that the first term, known as "Gauss-Newton matrix", G-term or Fisher information matrix (FIM), dominates the second term in terms of the largest eigenvalue. Thus when considering the sharpness, we can analyze the largest eigenvalue of FIM which is a close proxy of the largest eigenvalue of Hessian. In the binary MSE loss case, $\left. \frac{\partial^2 \ell(z, y_i)}{\partial z^2} \right|_{f(\mathbf{x}_i)} = 2$, which implies that

$$
\left\| \nabla_{\boldsymbol{\theta}}^2 \mathcal{L}(f(\boldsymbol{\theta})) \right\|_2 \approx \|G\|_2 := \frac{2}{n} \left\| \sum_{i=1}^n \frac{\partial f(\mathbf{x}_i, \boldsymbol{\theta})}{\partial \boldsymbol{\theta}}^\top \frac{\partial f(\mathbf{x}_i, \boldsymbol{\theta})}{\partial \boldsymbol{\theta}} \right\|_2 = \frac{2}{n} \left\| \frac{\partial \boldsymbol{F}(\boldsymbol{\theta})}{\partial \boldsymbol{\theta}}^\top \frac{\partial \boldsymbol{F}(\boldsymbol{\theta})}{\partial \boldsymbol{\theta}} \right\|_2
$$

where $\frac{\partial \boldsymbol{F}(\boldsymbol{\theta})}{\partial \boldsymbol{\theta}} := \left( \frac{\partial f(\mathbf{x_1}, \boldsymbol{\theta})}{\partial \boldsymbol{\theta}}^\top, ..., \frac{\partial f(\mathbf{x_n}, \boldsymbol{\theta})}{\partial \boldsymbol{\theta}}^\top \right)^\top \in \mathbb{R}^{n \times p}$. Meanwhile, Karakida et al. [16] pointed out the duality between the FIM and a Gram matrix $\boldsymbol{M}$, defined as

$$
\boldsymbol{M} = \frac{2}{n} \frac{\partial \boldsymbol{F}(\boldsymbol{\theta})}{\partial \boldsymbol{\theta}} \frac{\partial \boldsymbol{F}(\boldsymbol{\theta})}{\partial \boldsymbol{\theta}}^\top \tag{22}
$$

It also known as the neural tangent kernel NTK (Karakida et al. [16, 15]), which has been studied extensively in recent years (see e.g., [13],[8],[2],[5]). Note that in this paper, we do not assume the training is in NTK regime, in which the Hessian does not change much during training. It is not hard to see that $\boldsymbol{M}$ and FIM share the same non-zero eigenvalues: if $\boldsymbol{G}\boldsymbol{u} = \lambda \boldsymbol{u}$ for some eigenvector $\boldsymbol{u} \in \mathbb{R}^p$,

$$
\boldsymbol{M} \frac{\partial \boldsymbol{F}(\boldsymbol{\theta})}{\partial \boldsymbol{\theta}} \boldsymbol{u} = \frac{\partial \boldsymbol{F}(\boldsymbol{\theta})}{\partial \boldsymbol{\theta}} \boldsymbol{G} \boldsymbol{u} = \lambda \frac{\partial \boldsymbol{F}(\boldsymbol{\theta})}{\partial \boldsymbol{\theta}} \boldsymbol{u},
$$

in other words, $\lambda$ is also an eigenvalue of $\boldsymbol{M}$.

**Sharpness:** There are various definitions of sharpness in the literature ([20, 30, 9]). In particular, a popular definition of the sharpness is the largest eigenvalue of the Hessian ([30]). Based on the above discussion, in this paper, we adopt the largest eigenvalue of $M$, $\Lambda(\boldsymbol{\theta}) = \lambda_{\max}(M)$ as the definition of the sharpness, which is a close approximation of the largest eigenvalue of the Hessian empirically.

**Gradient Descent:** In this paper, we study the trajectory of gradient descent and gradient flow [7]. We use $\boldsymbol{\theta}(t)$ to denote the parameter at iteration $t$ (or time $t$) and the sharpness at time $t$ as $\Lambda(t) = \Lambda(\boldsymbol{\theta}(t))$. We similarly define $M(t), \boldsymbol{F}(t), \boldsymbol{D}(t), \mathcal{L}(t)$.

Along GD trajectory, the weight vector $\boldsymbol{\theta}(t)$ is updated in the following way:

$$\boldsymbol{\theta}(t+1) = \boldsymbol{\theta}(t) - \eta \frac{\partial \mathcal{L}(\boldsymbol{\theta}(t))}{\partial \boldsymbol{\theta}(t)}$$

When the learning rate $\eta$ is infinitesimal, the GD trajectory above is equivalent to gradient flow trajectory. Here we show the gradient flow dynamics of the residual vector $\boldsymbol{D}(t)$:

$$\frac{\mathrm{d}\boldsymbol{D}(t)}{\mathrm{d}t} = \frac{\partial \boldsymbol{D}(t)}{\partial \boldsymbol{\theta}} \frac{\mathrm{d}\boldsymbol{\theta}(t)}{\mathrm{d}t} = -\frac{\partial \boldsymbol{F}(t)}{\partial \boldsymbol{\theta}} \frac{\partial \mathcal{L}(t)}{\partial \boldsymbol{\theta}} = -\frac{2}{n} \frac{\partial \boldsymbol{F}(t)}{\partial \boldsymbol{\theta}} \frac{\partial \boldsymbol{F}(t)}{\partial \boldsymbol{\theta}}^{\top} \boldsymbol{D}(t) = -M(t)\boldsymbol{D}(t) \quad (23)$$

In light of (23), we have the following approximate update rule of $\boldsymbol{D}(t)$ under gradient descent:

$$\boldsymbol{D}(t+1) - \boldsymbol{D}(t) \approx -\eta M(t)\boldsymbol{D}(t)$$

---

[7] Gradient flow is a good approximation of gradient descent in the beginning of the training. See Cohen et al. [6] for more experiments and Ahn et al. [1] for theoretical justification. We assume this fact during the progressive sharpening phase. See Assumption 3.3.