# OpenReview forum: "Analyzing Sharpness along GD Trajectory: Progressive Sharpening and Edge of Stability"
_NeurIPS.cc/2022/Conference — NeurIPS 2022 Accept_

### Official Review · Reviewer_Nthg · 2022-07-10

**Rating:** 8
**Confidence:** 4
**Soundness:** 4 excellent
**Presentation:** 4 excellent
**Contribution:** 4 excellent

**Summary:**

This paper takes some steps towards theoretically understanding the "Edge of Stability" (EoS) phenomenon.  Two big open questions are: (1) why does the sharpness tend to rise whenever gradient descent is stable, and (2) once gradient descent is destabilized, why does the sharpness go back down / stop rising?

The paper has two parts: (1) a rigorous mathematical analysis of the two-layer linear network case (section 4), and (2) a nonrigorous mathematical analysis of the general neural net case combined with thorough experiments (section 3).

**Rigorous analysis of two-layer linear network**:   For a two-layer linear network, the NTK is the sum of two terms: (a) a term originating from the gradients w.r.t the first layer, and (b) a term originating from the gradients w.r.t the second layer.  Term (a) has a simple form: it is the product of the second-layer weight norm ( a scalar) and the input data Gram matrix.  Sharpness is defined as the maximum eigenvalue of the NTK (rather than the Hessian).

The authors first prove (in Lemma C3) that term (b) of the NTK will barely change during the first phase of training (the initial sharpening phase) -- rigorously, they prove that the delta of this term will be O(1/m) where m is the width of the network.  This implies that the NTK will approximately stay diagonalizable by the eigenvectors of the input data Gram matrix, and also that the sharpness is essentially totally determined by the second-layer weight norm.  Thus, to prove that progressive sharpening occurs, it suffices to prove that the second-layer weight norm increases during gradient descent.  In the update equation of the second-layer weight norm, the first-order term is the negative inner product between the predictions and the residual.  The authors are able to rigorously prove that the higher-order terms don't matter, and thus to prove progressive sharpening it suffices to prove that the inner product between the predictions and residual is always negative.  This is proved in Lemma C5.

The authors next prove that after becoming destabilized, gradient descent will move to a flatter region where the sharpness is below 2 / eta.  To use the authors' own words, the proof is "long and tedious," so I didn't attempt to parse the proof; but I assume that the intuition is similar to that of Proposition 3.2, which is that if the residual ever grows too big, then the last-layer weight norm will automatically decrease.

**Beyond two-layer linear networks** The authors next attempt to show experimentally that the insights from the two-layer linear NN analysis  carry over in a very literal way to the general neural network case.  (Here, I write "next," but in the current draft this section occurs first chronologically.)  In particular, they demonstrate experimentally that in many networks, the dynamics of the sharpness correlate well with those of the last-layer weight norm: when the sharpness rises, so does the last-layer weight norm, and when the sharpness drops, so does the last-layer weight norm.   This seems to be more true in the initial phase of training, and becomes less true towards the end.  To their credit, the authors admit that there are counterexamples to this trend.


**Questions:**

For the two-layer NN:
 - does the degree of sharpening depend on the input dataset?  If we trained on random Gaussian data, would there still be sharpening?
 - does the degree of sharpening depend on the last-layer initialization (which is much larger in scale than the standard initialization)?



**Limitations:**

The paper is honest about its limitations.

**Strengths And Weaknesses:**

- Strength: this paper is the first to prove that some architecture (in this case, a two-layer linear network) undergoes progressive sharpening.  Indeed, this is the first paper to even give any kind of explanation for why progressive sharpening might occur.  (Actually, it was not even previously known that 2-layer linear nets can exhibit progressive sharpening.)

- Strength: I think that this paper is the first to prove that for some architecture (in this case, a two-layer linear network), instability causes the sharpness to decrease.  (The catapult paper includes a handwavy explanation for this phenomenon, but not a literal proof.)

- Weakness: the paper is at its weakest when it attempts to argue that its EoS analysis for two-layer linear networks carries over in a very literal manner to general neural networks.  First, I would point out that the correlation between sharpness and last-layer weight norm is not very robust: in Appendix B.1.2, we see that after the first few cycles of instability, there are a huge number of 'anomaly points' (steps where the change in the sharpness is not positively correlated with the change in last-layer weight norm).  Second, I would point out that "all layers seem to work together to influence the sharpness," as the authors write.

Overall, I would recommend framing this paper very differently.  I suggest centering the two-layer linear network analysis rather than the debatable claims about general neural networks.  If the authors are concerned that this analysis is very tedious, I would recommend just providing the intuition in the main paper (e.g. the fact that the leading term in the change in the sharpness is the inner product between the residual and the predictions) while deferring the complete proofs to the appendix.  Then, _after_ discussing the two-layer linear network analysis, you could mention that some of the patterns might carry over to general networks.  The authors are of course free to take or leave this advice, but I think that this restructuring would make the paper more compelling.

---

> ### Author Response · Authors · 2022-08-02
> **Response**
>
> Thanks a lot for the insightful and encouraging comments.
> We do believe our work provides interesting new insights into PS and EOS
> (despite assumptions and heuristic arguments in various points).
>
> **Regarding the organization of the paper:**
>
> Actually, we did first come up with the theoretical proof of the two-layer network.
> First, as you mentioned, the rigorous proof is very tedious.
> For example, a lot of intricate calculations are used to deal with higher order terms, and bounding those terms does not convey much interesting insights.
> In fact in our experiments (see Figure 14 in the revised version),
> the first order approximation is already a very good approximation.
> More importantly, we found that our main proof ideas surprisingly can capture
> the empirical observations even in most general cases (at least for those deeper and nonlinear NNs in our experiments). So we decided to present the results for the general cases first (at the price of rigor to some extent, but the gain is more generality and less tedious calculations).
>
> >"I would point out that the correlation between sharpness and last-layer weight norm is not very robust":
>
> First, as you have mentioned, this correlation is very true especially in the early phase of the training. To our best of knowledge, existing recent analysis ([3], [22]) on GD trajectory mainly focus on the late training phase near the manifold of minima. So we hope this observation can shed some light on the early phase, which is also very important. Second, we admit that this correlation is not the whole story, and we hope future works can lead to a clearer understanding.
>
> We notice that in some cases, despite $\|A\|^2$ has nearly the same trend as the sharpness,
> while the sharpness oscillates more frequently (i.e., the sharpness curve has higher frequency oscillations. See Figure 7 or Figure 11). In this case, while we believe the change of $\|A\|^2$ is a major driving force of the change of the sharpness, other factors (such as other layers) must be taken into consideration for completely understanding such oscillations.
>
> >"does the degree of sharpening depend on the input dataset? If we trained on random Gaussian data, would there still be sharpening?":
>
> The answers are both yes. The training on random Gaussian data also has Progressive Sharpening, and we include the figure in Appendix (B.1.4).
> Indeed, the degree of sharpening heavily depends on the complexity of the input dataset. For example, Gaussian data, or even MNIST, they are much easier
> to learn than CIFAR, so the convergence is much faster, which leads to a smaller degree of Progressive Sharpening.
>
>
> >"does the degree of sharpening depend on the last-layer initialization (which is much larger in scale than the standard initialization)?":
>
> Yes. In our model, we add a $\sqrt{m}$ before and initialize the last layer $\sqrt{m}$ times larger to simplify our analysis. This causes the gradient of the last layer smaller but the initialization larger. Hence in normal two-layer networks, the last-layer norm changes more significantly compared to its initialization, which leads to an even larger degree of Progressive Sharpening.
>
> Last, thank you again for your valuable advice and questions!

---

> > ### Comment · Reviewer_Nthg · 2022-08-03
> > **additional questions**
> >
> > Thanks!  Here are some additional questions:
> >
> > 1.  What happens at infinite width?  Does sharpening still occur?
> >
> > 2.  If the answer to question 1 is "no," then for the final draft could you provide experiments illustrating how the degree of sharpening depends on the width?
> >
> > 3.  If the answer to question 1 is "yes," then your parameterization must differ from the NTK parameterization -- in what way does it differ?
> >
> > 3.  If your parameterization is different from the NTK parameterization, does it coincide with the "mu parameterization" of Yang and Hu '20   (https://arxiv.org/abs/2011.14522)?
> >
> > 4. You say that little sharpening occurs on random isotropic Gaussian data.  What happens if you considered anisotropic Gaussian data -- say, data with the same covariance spectrum as CIFAR-10.  Would there still be sharpening?  The other factor to consider is the label vector, and whether it aligns with the top eigenvectors of the data covariance, the bottom eigenvectors, or is evenly distributed.  It would be worthwhile for the final draft (no need to do this now) to explore these questions.

---

> > > ### Author Response · Authors · 2022-08-08
> > > **Response**
> > >
> > > Thank you for your further questions! The degree of progressive sharpening is an interesting open problem.
> > >
> > > >"What happens at infinite width? Does sharpening still occur?"
> > >
> > > For neural networks with infinite width, progressive sharpening does not occur. Cohen et al. ([6]) did experiments on two-layer tanh networks with different widths. They found out that the sharpness grows less when using a larger width (Figure 12,13 in [6]), which is consistent with the prediction of NTK theory. Note that they also did experiments using standard parameterization and got similar results (infinite width networks do not have progressive sharpening, see Figure 14 in [6]).
> > >
> > > >anisotropic Gaussian data and the label vector
> > >
> > > We have done further experiments on two-layer fully-connected linear networks with different datasets, and we include the new experiment results in the revised version.
> > >
> > > In our experiments in Appendix B.1.4, we train the network with data points sampled from Gaussian distribution and different label vectors. In particular, the width of the network is 400. The  input $X$ consists of 500 data points sampled from $\mathcal{N}(\mu, \Sigma)$, where the mean $\mu\in \mathbb{R}^{3072}$ and the covariance $\Sigma\in \mathbb{R}^{3072\times 3072}$ are the mean vector and the covariance matrix of 5000 CIFAR-10 data points, respectively. We note that to illustrate the degree of progressive sharpening, we choose a very small learning rate $\eta= 2/20000$
> > > (so that the training converges before EOS can happen).
> > >
> > > In Figure 13(a), the label $Y$ is uniformly sampled from $\text{Unif} \{ -1,1\} ^n$.
> > > Comparing the experimental results with those obtained using the same network trained
> > > with 500 CIFAR-10 data points, we find out that both have a similar degree of progressive sharpening.
> > > In Figure 13(b) and Figure 13(c), we let the label $Y$ be $\sqrt{n}v_1$ and $\sqrt{n}v_{400}$ respectively. The result shows that when the label vector is aligned with the top eigenvector $v_1$, the degree of progressive sharpening is relatively small,
> > > and when the label vector is aligned with a bottom eigenvector, the degree of progressive sharpening is relatively large.
> > >
> > > This phenomenon is consistent with our intuition.
> > > Empirically we found that a dataset that is easier to learn leads to faster convergence rate, which then leads to a smaller degree of progressive sharpening. For example, training standard Gaussian distributed dataset converges faster than that with CIFAR-10 dataset, hence the degree of progressive sharpening of the former is much smaller; as another example
> > > (Figure 13(b) and Figure 13(c)),
> > > if the label vector is aligned with the top eigenvector, the convergence in the first phase (the PS phase) is faster, which leads to a shorter first phase and thus a smaller degree of progressive sharpening compared to the other case.

---

### Official Review · Reviewer_VhEp · 2022-07-11

**Rating:** 3
**Confidence:** 4
**Soundness:** 1 poor
**Presentation:** 1 poor
**Contribution:** 2 fair

**Summary:**

This paper proposes a theoretical explanation for the progressive sharpening and edge of stability phenomena observed in Cohen et al (2021). Specifically, this paper claims that the gradient descent dynamics can be divided into four stages based on the value of the sharpness. For a general neural network, the paper gives a heuristic derivation for the four stages, where the proxy for sharpness is the last layer weight. It also gives a more rigorous derivation for a two-layer linear neural network.

**Questions:**

As mentioned in the weaknesses section, I find a number of the assumptions to be unrealistic, and don’t find the empirical justification sufficient. Can you please elaborate on why these assumptions are reasonable? In particular, why the last layer norm is a good proxy for sharpness, and why one can assume $W(t)$ changes very little for the two-layer linear network?

Furthermore, can you please explain why the argument in section 3 guarantees a return to phase I?

From a clarity perspective, I find the assumptions and heuristic derivations being stated within the main theorems hard to follow. One suggestion is to have an explicit assumption section before the main results in section 3.

**Limitations:**

The paper does admit that the analysis is highly heuristic, which is the main limitation of this work.

**Strengths And Weaknesses:**

Strengths:

The edge of stability phenomenon contradicts much of classical optimization, and as of yet does not have a satisfactory theoretical explanation. Therefore I find this work to be an important research direction. Furthermore, the four stage division of dynamics proposed here appears to be novel.

Weaknesses:

My central issue with this paper is that the derivations are far too heuristic. As a result, I do not find the claims to be theoretically sound or a convincing explanation of the edge of stability phenomenon. Some specific instances are the following (all page references are for the version in the supplementary material):

Section 3:
- Rather than tracking the sharpness, this paper tracks $\|A(t)\|^2$, where $A(t)$ is the last layer weight. I don’t believe that in general the last layer weight is an accurate proxy for sharpness. The justification given here is that the difference in norms is usually the same sign as the change in sharpness; however, the empirical justification in Figure 2 is unconvincing as this correlation is only strong for the two layer linear network. Furthermore, while the analysis in Section 3 shows that $\|A(t)\|^2$ increases or decreases, this does not necessarily imply that the sharpness will decrease below $2/\eta$, which is a central component of the empirical analysis in Cohen et al (2021).
- This analysis does not track the full GD dynamics, but rather a first order Taylor expansion of the gradient. Appropriately dealing with the full dynamics seems far more challenging. This should also be stated explicitly as an assumption.
- Assumption 3.2 is very strong, and changing eigenvectors could have a nontrivial effect on the dynamics.
- I don’t find the argument for why the dynamics eventually return to stage I convincing. This argument is given at the bottom of page 8 + top of page 9 and is quite handwavy / unrigorous.
- Lemma 3.5 claims to explain the non-monotonic loss decrease, by saying $R(t)$, the dynamics with the $v_1$ direction projected out, approximately follows $R’(t)$. However, it seems that the error term in this Lemma can be very large, since $\eta$ is small and $\lambda_r$ can be very small as well.

Section 4:
- I don’t find Assumption 4.3 to be realistic. First, the fact that $\Gamma(t)$ is bounded during the progressive sharpening phase does not mean it will stay bounded during the EOS phase. In fact, Figure 18 clearly shows that $\|\Gamma(t)\|$ spikes during the edge of stability. Another issue is that assumption 4.3 amounts to the weights $W(t)$ changing very little during training, which is essentially equivalent to saying the network is in the lazy training/NTK regime. It is known that neural networks in practice don’t follow lazy training and weights do more far from initialization, and since edge of stability is inherently a non-quadratic phenomenon, I thus find it unreasonable to assume that weights do not move far from initialization. This assumption appears to be key to the claims in section 4, and therefore I do not find the main claims of this section to be convincing.

Furthermore, the supplementary material contains an alternate version of the paper where the main text is much longer (13 pages). I am not sure if this is allowed under the submission guidelines.

---

> ### Author Response · Authors · 2022-08-02
> **Response (For Section 3)**
>
> Thanks for the detailed and insightful comments.
> Since a complete and rigorous explanation of progressive sharpening (PS) and EOS phenomena can be extremely challenging, we adopted several heuristic arguments at various points.
> As indeed our choice of exposition may cause confusion to some extent for those readers who seek for rigorous explanations, in light of the reviewer's critics,
> we have revised our manuscript substantially, by
> reorganizing and explicitly stating our assumptions,
> further relaxing some stringent assumptions,
> and adding several extensive discussions and experimental results.
> We believe our work provides novel and nontrivial insights and techniques for understanding PS and EOS, and hope the reviewer
> can read our responses and reexamine our revision.
>
> Below we summarize our changes and respond to specific questions.
>
> >About |A|
>
> First, we should mention that this observation is very true in the early phase of training (especially during the PS phase and early EOS phase).
>
> Second, in Appendix B.2, we provide detailed discussion about the anomaly points.
> It is true that when the sharpness has high frequency oscillation (but the high frequency part has small magnitude) in late training phase, the change of $|A|^2$ and the sharpness do share very similar larger trend. For example, in Figure 2(b), the
> trend of $|A|$ and the sharpness fit very well even when there are anomaly points (caused by small but high frequency oscillation of the sharpness).
> As we have stated in Appendix B.2, we admit that $|A|$ cannot represent the sharpness to the fullest extent, but we believe the change of $|A|$ is a major driving force of the change of the sharpness.
>
> Third, besides $|A|$, we also discussed the influence of other layers on the sharpness in Appendix D.3. Figure 19 shows that "all layers seem to work together to influence the sharpness", further justifying our assumption that $|A|$ is positively related with the sharpness. Of course, we still do not understand the small but high frequency oscillations of the sharpness during the late training phase, and leave it as an interesting open direction.
>
> >>"this does not necessarily imply that the sharpness will decrease below $2/\eta$".
>
> Proposition 3.2 shows that $|A|$ will decrease significantly as long as the sharpness does not drop below $2/\eta$.
> Although this does not rigorously show the sharpness will drop below $2/\eta$,
> our result is a strong implication.
> Of course a rigorous proof requires a quantitative relation between $|A|$ and the sharpness for general neural networks. We were only able to establish such a relation for two-layer linear networks and prove that the sharpness will decrease below $2(1+\rho)/\eta$ (Lemma C.13).
>
> >First Order Approximation
>
> Yes, handling higher order terms is very challenging. One can see that in our two layer linear case, the higher order terms
> are already very complicated and bounding them can be very technical.
> In our revision, we explicitly make the first order approximation
> as an assumption for Section 3 (the general case). We also verify the assumption in Appendix D.1.3.
>
> >Regarding Assumption 3.2
>
> We admit that Assumption 3.2 is somewhat strong. In fact, the assumption can be relaxed significantly. In our revised version, we provide a dynamical system view of the dynamics of $D(t)^\top F (t)$ and show in Appendix D.4 (Lemma D.1) that if the eigenvectors do not change very fast, we can still guarantee $D(t)^\top F(t) < 0$.
> We also include a discussion that even the condition of Lemma D.1 fails for some eigendirections with small contributions, we can still arrive at the same conclusion, thereby further relaxing the assumption.
>
> > Regarding R and R'
>
> First, despite the fact that $\eta$ and $\lambda_r$ are relatively small variables, we should notice that there is another small variable $\epsilon_2$ appearing in the numerator. In many cases $\epsilon_2$ can be much smaller than the other two terms (for example in the two-layer linear network case), which keeps the whole error term small.
>
> Moreover, the actual value of $|R-R'|$ is quite small in practice and we showed that empirically. As showed in Figure 3(b) or more figures in Appendix B, $|R|$ follows a nearly converging curve except for a tiny bulge when the sharpness just grows over $2/\eta$. Hence, our analysis, especially the decomposition of $D$ into $v_1$-component and $R$ and the analysis of their individual dynamics, provides a good explanation of the non-monotonically decreasing loss.
>
> >Return to Phase I
>
> We apologize for being vague, and we try to make it clearer in the revised version. The basic idea is to decompose $D(t)^\top F(t)$ into two parts. The first part is geometrically decreasing, and then using $R'$ as an approximation of $R$, the second part can be proved negative. Hence we can get a conclusion that either $D(t)^\top F(t)<0$, meaning that the training enters Phase I again, or the loss is almost zero, meaning that the training ends.

---

> > ### Comment · Reviewer_VhEp · 2022-08-08
> > **Thank you for the response**
> >
> > Thank you to the authors for the very detailed response.
> >
> > I appreciate the fact that the revised version of the paper now explicitly states the assumptions made, and that the Appendix contains further discussion / verification of these assumptions. As the revision is quite substantial, I am still in the process of going through the results. Most of my concerns about Section 3 have been addressed, though I still have the following remaining comments:
> >
> > - In regards to analyzing the last layer weight $||A||$ as a proxy for sharpness, I believe that this could be emphasized further in the introduction just to make clear the contribution of this work.
> >
> > - I still do have a concern with the assumption on $||\Gamma(t)||$. Your claim is that Figure 21 shows that $||\Gamma(t)|| = \Theta(1/m)$, even during the EOS phase. However, $||\Gamma(t)||$ spikes a large amount when training is unstable. Assuming that it can be bounded by $\Theta(1/m)$ throughout seems to disregard this spiking behavior in the analysis. To me, it is difficult to believe that this quantity which grows by a factor of 10 during these spikes can actually be considered to be the same order of magnitude throughout training.
> >
> > - Secondly, to comment on the NTK setting, $\Gamma(t)$ seems to control how much $W(t)$ changes, and assuming $\Gamma(t)$ is bounded in the analysis seems to be equivalent to assuming $W(t)$ does not change much. I agree that empirically $W(t)$ does change, but to me this seems to be because $\Gamma(t)$ is large. Therefore I believe assuming $\Gamma(t)$ is $\Theta(1/m)$ prevents $W(t)$ from moving too much, which makes the analysis in Section 4 less significant.

---

> > > ### Author Response · Authors · 2022-08-09
> > > **Response**
> > >
> > > We appreciate your careful reading of our responses, and thanks a lot for your further thoughtful comments. We would like to further clarify our contribution and weakness.
> > >
> > > >"In regards to analyzing the last layer weight $\|A\|$ as a proxy for sharpness, I believe that this could be emphasized further in the introduction just to make clear the contribution of this work."
> > >
> > > According to your suggestion, we have stated this relation more explicitly in the introduction section (see the third paragraph in Section 1.1., the $\|A\|$ approximation of the sharpness)
> > > and made this assumption explicit in the revised version.
> > >
> > > >"the assumption on $\|\Gamma(t)\|$"
> > >
> > > Thanks for the question, and this was indeed one of the assumptions that require much discussion (even among the co-authors of the present submission).
> > > There are two interesting empirical facts related to $\|\Gamma(t)\|$.
> > > One is the noticeable fact that $\|\Gamma(t)\|$ spikes, as the reviewer has also noticed. The other is that the values of $\|\Gamma(t)\|$ are overall
> > > quite small (despite the fact that $W$ changes non-trivially in our experiments) and decreases as $m$ becomes larger. Our assumption tries to
> > > model the second fact
> > > (recall that in Figure 21, $\|\Gamma(t)\|$ (despite the spikes) decreases as the width $m$ grows, and the largest $\|\Gamma(t)\|$ is almost $24/m$ in all these experiments).
> > >
> > > However, we agree with the reviewer that our results do not reflect the first fact (the spikes of $\|\Gamma(t)\|$), and it is an interesting fact that is worth investigating.
> > > We have strong intuition that $\|\Gamma(t)\|$ only grows by at most a constant factor, but currently do not have a formal proof yet.
> > > Nevertheless, the spiking behavior does not directly contradict our assumption.
> > >
> > > >"the NTK setting"
> > >
> > > As we have argued, our setting and results are sufficiently different from the recent convergence analysis in the NTK setting. Even if $\|W(t)\|$ changes little, PS and EOS can still happen (See Appendix D.2.2, and do not forget that the A-norm also contributes to the sharpness), and our results can certainly cover this phenomenon, which cannot be explained by any existing NTK analysis.
> > >
> > > Moreover, $\|\Gamma(t)\|$ is small does not necessarily mean $\|W(t)\|$ changes little.
> > > We state our reasoning in the fifth paragraph in our last response ("Furthermore, ..."). Of course, it would be ideal to formally prove that $W$ changes significantly, and currently we cannot prove it.
> > > But this does not contradict our assumption either.

---

> ### Author Response · Authors · 2022-08-02
> **Response (For Section 4)**
>
> >‘First, the fact that $\Gamma(t)$ is bounded during the progressive sharpening phase does not mean it will stay bounded during the EOS phase.'
>
> Indeed, we do not expect the term $\Gamma(t)$ could be bounded as tight as in
> the progressive sharpening phase, since the instability may cause a notable change in $W(t)$. However, we assume that the bound only grows in a constant multiple of the original bound, and our assumption does not contradict our experiment.
> Though |$\Gamma(t)$| spikes during the edge of stability, it is still a constant multiple of |$\Gamma(t)$| at the end of the progressive sharpening phase, therefore a constant multiple of the bound $40/m$. In fact, in Appendix D.2.2 and before Figure 18 we explained that the corresponding constant $c_2$ in the assumption is approximately 24. Also, we did experiments under the two-layer linear network setting with different $m$. Due the limitation of computation resource, we can verify that the constant here does not change much around this value for $m = 40, 80, 160, 200$.
> We add the four experimental results in the revised version in the same section to further corroborate this point. In sum, our empirical observation supports the assumption.
>
> >‘It is known that neural networks… and since the edge of stability is inherently a non-quadratic phenomenon, I thus find it unreasonable...’
>
> This is an insightful and very critical question.
> Yes, the edge of stability is inherently a non-quadratic phenomenon,
> and now we explain why our setting and results are sufficiently different from
> the quadratic setting (e.g., linear regression) or the recent convergence analysis in
> NTK setting.
>
> A key requirement in the convergence analysis in the NTK regime is that
> the learning rate is very small and the GD trajectory almost tracks the gradient flow,
> hence converges to the global minimum.
> However, we consider typical learning rate used in practice, which can be much larger. In particular, $\eta>2/\Lambda$ can happen in our setting, which causes instability (i.e., such as the growth of loss in Lemma C.11) along the training trajectory.
> Such instability cannot be captured by any existing convergence analysis in NTK regime at all.
> Hence, all existing NTK convergence results do not directly apply here.
>
> Equally importantly, we do find that even when $W(t)$ changes slightly (several orders of magnitude smaller than its initialization), PS and EOS still happen with a not so small learning rate $\eta$. To support our claim, we slightly modified our experiment setting in Appendix D.2.2 (to be specific, we enlarged $W(0)$ by 10 times and used wider networks with $m=1000, 2000, 4000, 8000$). The new experimental results are included in
> Appendix D.2.3 in the revised version). We can see that the initialization $W(0)$ is
> much larger than the change of $W(t)$ and the norm of $W(t)$ grows larger when $m$ becomes larger. However, we still observe that PS and EOS occur in this setting. Hence, the setting we considered and our results are intrinsically not quadratic (in which case EOS cannot happen).
>
> Furthermore, our assumption on |$\Gamma(t)$| does not necessarily imply the sharpness does not change much. In the last subsection in Appendix C, we included a paragraph of discussion. It is our non-tight analysis that makes our setting look like an NTK regime (but it’s not since the large learning rate). For example, if we can tighten our analysis to $m=\Theta(n)$ (currently it is $m=O(n^2)$), both $W(t)$ and $A(t)$ can grow by a constant multiple of its initial norm.
> Meanwhile, evidence shows that even if the assumption on |$\Gamma(t)$| is correct when $m=O(n)$ in our experiment setting in Appendix D.2.2, the norm of the movement of $W(t)$ is significant relative to the initial value $W(0)$.
> We believe the bound $m=O(n^2)$ can be tightened. In any case, our analysis does not fall into the quadratic regime.
>
> >'Furthermore, the supplementary material contains an alternate version of the paper where the main text is much longer (13 pages).'
>
> Since we do not find any page restriction for
> the supplementary material, we chose to moved the proofs in Section 3 to the main text
> in order to provide better reading experiences.

---

### Official Review · Reviewer_z1Jj · 2022-07-11

**Rating:** 7
**Confidence:** 4
**Soundness:** 3 good
**Presentation:** 3 good
**Contribution:** 3 good

**Summary:**

The work aims to understand the phenomena of edge of stability along with progressive sharpening (rising of leading eigenvalues of the Hessian). As a foundation, it provides an interesting observation wrt the correlation between sharpness and the output layer's norm. Then it divides the analysis into four stages where the behavior of the leading eigenvalue is studied. Moreover, it proves the theorems in the setting of two-layer linear neural networks.  Some assumptions are verified with numerical experiments.



**Questions:**

Please see the above concerns.

Typo:

1. Shall the fraction in Eq(3) be $\frac{4}{n}$?

**Limitations:**

No. But it is a pure theoretical work in a standard setting of optimization, so there is no significant need to do so.

**Strengths And Weaknesses:**

Pros:

1. The analysis is clear and convicing generally, with a detailed investigation into the four stages.
2. The observation of the correlation between the sharpness and output layer's norm is quite interesting, which makes it a good start point for the following analysis.

Concerns:

1. How to handle several eigenvalues close to $2/\eta$? This phenomena is observed by Cohen et al., but the assumption 3.6 and 4.1 rule it out.

Generally speaking, I would like to recommend an acceptance score for the complete analysis to resolve the problem of EoS.

---

> ### Author Response · Authors · 2022-08-02
> **Response**
>
> Thank you for your encouraging and constructive comments.
> We have fixed the typos you pointed out.
>
> Question "How to handle several eigenvalues close to $2/\eta$?"
>
> Response:
> In this paper, we focus on the binary classification setting, and empirically
> there is only one eigenvalue (the outlier) that can be close to $2/\eta$. In the multi-class classification setting, previous works (Sagun et al.[29], Papyan [26])
> empirically showed that the $k$-class classification networks have $k$ outliers in the spectrum of their Hessian. Hence, in this setting there might be more than one eigenvalues that are close to $2/\eta$, and this is an important further direction.
>
> Nevertheless, our result for binary classification may have interesting implications on the $k$-class settings. First, S. Fort and S. Ganguli (arXiv:1910.05929) empirically showed that the logit gradients of different classes (i.e, $\frac{\partial f_i(x)}{\partial \theta}$, where $i = 1,2,...,k$, $x$ is one of the training data, and $\theta$ is the parameter of the network) are almost perpendicular to each other. Hence, the $k$-class training can be seen approximately as $k$ independent binary classification problems and the sharpness of the $k$-classification training can be approximated by the maximum of that of these $k$ binary-classification problems. Then, our technique or result may be useful in analyzing the sharpness of these binary-classification problems. In fact, from some experiments on $k$-class classification (e.g., Figure 1 in Cohen et al.), we can see the sharpness hovers above $2/\eta$ for a long time (rather than
> going up and down crossing $2/\eta$ frequently in the binary classification case). We hypothesize that this is because the largest eigenvalues of these $k$ approximately independent binary-classification problems exceed $2/\eta$ alternatively. Here we only scratch the surface and leave a more serious exploration of multi-class classification setting as an important further work.

---

> > ### Comment · Reviewer_Nthg · 2022-08-03
> > **agree with the authors**
> >
> > I agree with the authors that the observation of multiple eigenvalues above $2/\eta$ in Cohen et al '21 is probably related to the fact that that paper was studying multiclass problems.  I also agree that it is sufficient for now to focus on binary classification.

---

### Meta-Review · Area_Chair_LBz1 · 2022-08-26

**Recommendation:** Accept
**Confidence:** Less certain

**Metareview:**

While there is a rather large gap between the reviewers' scores, all the reviewers agreed that the paper is novel and the contributions are significant, especially given the little knowledge about the EoS phenomenon. While one of the reviewers raised important concerns about the appropriateness of the assumptions, I do believe that in a field without a rich enough literature, initial theoretical results with potentially strong assumptions are still valuable. Hence I am recommending an acceptance for the paper.

Please implement all the changes that have been requested by the reviewers. On the other hand please avoid using gender pronouns like "he" when addressing the reviewers.

**Award:**

No

---

### Decision · Program_Chairs · 2022-09-14

Accept